# SAMPLING COMPLEXITY OF TD AND PPO IN RKHS

**Lu Zou[1], Wendi Ren[2], Weizhong Zhang[3], Liang Ding[3], Shuang Li[2]***
[1]School of Management, Shenzhen Polytechnic University
[2]School of Data Science, The Chinese University of Hong Kong, Shenzhen
[3]School of Data Science, Fudan University
`zoulu1990330@gmail.com, wendiren@link.cuhk.edu.cn`
`{weizhongzhang, liang_ding}@fudan.edu.cn, lishuang@cuhk.edu.cn`

## ABSTRACT

We revisit Proximal Policy Optimization (PPO) from a function-space perspective. Our analysis decouples policy evaluation and improvement in a reproducing kernel Hilbert space (RKHS): (i) A kernelized temporal-difference (TD) critic performs efficient RKHS-gradient updates using only one-step state–action transition samples. (ii) a KL-regularized, natural-gradient policy step exponentiates the evaluated action-value, recovering a PPO/TRPO-style proximal update in continuous state-action spaces. We provide non-asymptotic, instance-adaptive guarantees whose rates depend on RKHS entropy, unifying tabular, linear, Sobolev, Gaussian, and Neural Tangent Kernel (NTK) regimes, and we derive a sampling rule for the proximal update that ensures the optimal $k^{-1/2}$ convergence rate for stochastic optimization. Empirically, the theory-aligned schedule improves stability and sample efficiency on common control tasks (e.g., CartPole, Acrobot, and HalfCheetah), while our TD-based critic attains favorable throughput versus a GAE baseline. Altogether, our results place PPO on a firmer theoretical footing beyond finite-dimensional assumptions and clarify when RKHS-proximal updates with kernel-TD critics yield global policy improvement with practical efficiency.

## 1 INTRODUCTION

Policy-gradient and trust-region methods (e.g., natural policy gradient (NPG) (Kakade, 2001), trust-region policy optimization (TRPO) (Schulman et al., 2015), proximal policy optimization (PPO) (Schulman et al., 2017), and actor–critic (AC) methods (Konda & Tsitsiklis, 1999)), when coupled with temporal-difference (TD) critics, are among the most effective tools in modern RL for large, continuous control, thanks to their compatibility with expressive function approximators and stable improvement steps. Yet despite impressive empirical success, our theoretical understanding of global convergence for these algorithms under expressive function approximation remains fragmented across settings. In particular, a central problem is to design an algorithm that (i) performs policy evaluation with controlled statistical error when the action–value function lies in a rich function class, and (ii) couples this evaluation step with a policy improvement update that provably ascends toward the optimal policy.

Existing analyses often establish convergence only in tabular/linear regimes or under strong realizability and concentrability conditions, see, e.g., (Agarwal et al., 2020; Bhandari & Russo, 2024). With nonlinear or nonparametric critics, guarantees frequently rely on idealized, exact value/advantage estimates. Moreover, many policy-improvement bounds treat expectation terms as if computed without sampling noise, leaving the per-iteration data requirements for ensured improvement unspecified.

For TD learning (Sutton, 1988; Maei et al., 2009; Bhandari et al., 2018), linear TD is well understood, with asymptotic convergence and finite-time error bounds. By contrast, the theory under nonlinear approximation is thinner. A key advance is (Cai et al., 2019), which establishes finite-sample convergence of neural TD under overparameterized networks with discrete actions (and continuous states) at a sublinear rate. On the policy-optimization side, the strongest results remain in tabular or linear settings; with expressive function classes, guarantees weaken. Notably, (Liu et al., 2019)

---
*Corresponding author

prove nonasymptotic global convergence of mirror descent policy optimization for two-layer overparameterized neural policies in continuous-state, discrete-action problems under the neural tangent regime.

In this paper, we take a function-space approach and optimize policies in an RKHS, which encompasses tabular/linear models, Sobolev classes, Gaussian kernels, and wide neural networks through their neural tangent kernels. We study policy evaluation and improvement using gradient-based updates: (i) a kernel TD critic–distinct from least-squares TD–that implements an RKHS-gradient iteration acting as an implicit preconditioner and avoids cubic-time matrix inversions, and (ii) a KL-regularized functional proximal step for policy improvement, implementable in continuous action spaces, where the policy is updated by exponentiating the evaluator's value estimate. Our main contributions are:

- We introduce a kernel, gradient-based TD evaluator in an RKHS that acts as an implicit preconditioner and attains geometric convergence without costly matrix inversions. The evaluator leverages $N$-step TD learning for any $N \geq 1$ and provides non-asymptotic TD-error bounds that attain the minimax rate (up to logarithmic factors).

- We design a KL-regularized proximal update implementable in continuous action spaces. We explicitly quantify the per-iteration sample size needed to achieve the intended improvement, addressing a common gap where policy expectations are treated as exact or left unspecified.

- Because kernel gradient descent mirrors the NTK dynamics of wide networks trained by gradient descent, our RKHS analysis directly informs neural critics/actors in the corresponding regimes. Experiments on continuous-control benchmarks exhibit the trends predicted by our theory.

## 2 RELATED STUDIES

TD method (Sutton, 1988) is one of the most commonly used for policy evaluation. The convergence of linear TD has been extensively studied, with finite-time error bounds established in recent works (Bhandari et al., 2018; Lakshminarayanan & Szepesvari, 2018; Srikant & Ying, 2019). In contrast, the behavior of TD with nonlinear function approximation remains less understood. A notable advance is due to Cai et al. (2019), who provided the first finite-sample analysis of neural TD, proving sublinear convergence under an overparameterized network; see also Brandfonbrener & Bruna (2019); Agazzi & Lu (2019) for related results in tabular settings. Policy optimization has also been extensively studied, with algorithms including policy gradient (PG) (Sutton et al., 1999; Baxter & Bartlett, 2000), natural policy gradient (NPG) (Kakade, 2001), trust-region policy optimization (TRPO) (Schulman et al., 2015), proximal policy optimization (PPO) (Schulman et al., 2017), and actor–critic (AC) methods (Konda & Tsitsiklis, 1999). Among these, NPG has been analyzed most thoroughly. Its convergence is well understood in the tabular setting, while function approximation presents additional challenges (Bhandari & Russo, 2024; Cen et al., 2022; Mei et al., 2020). For linear approximation, Agarwal et al. (2020) established global convergence in the tabular setting and restricted class of parametric policies, Agarwal et al. (2021) derived finite-sample rates for unregularized NPG with softmax parameterization. In more general settings, Zhang et al. (2020) obtained only local guarantees, and Cayci et al. (2024) provided sharp nonasymptotic bounds for entropy-regularized NPG. Broader classes of function approximation, including neural networks, have also been considered (Wang et al., 2019; Liu et al., 2019). TRPO and PPO have likewise received theoretical attention: Neu et al. (2017) and Shani et al. (2020) analyzed TRPO, while Liu et al. (2019) and Cai et al. (2020) studied PPO and its variant.

Our work is closely related to kernel methods (Hofmann et al., 2008; Zhou, 2008; Cho & Saul, 2009), which is one of the commonly used approaches in policy learning, e.g., Bagnell & Schneider (2003); Bethke et al. (2008); Grunewalder et al. (2012); Feng et al. (2020); Koppel et al. (2020). Early work e.g., Ormoneit & Sen (2002); Munos & Szepesvári (2008), established consistency results for non-parametric value function approximation. More refined analyses were later provided by Farahmand et al. (2016), who studied the convergence of the Bellman contraction mapping under the non-parametric setting, and by Duan et al. (2024), who proposed a regularized kernel-based LSTD estimator in RKHS for Markov chains independent of actions, deriving a non-asymptotic error bound and establishing a matching minimax lower bound.

## 3 PRELIMINARIES

***Markov Decision Process.*** We consider the MDP $(\mathcal{S}, \mathcal{A}, P, r, \gamma)$, where $\mathcal{S} \subset \mathbb{R}^{d_2}$ and $\mathcal{A} \subset \mathbb{R}^{d_a}$ are compact and convex sets, $P : \mathcal{S} \times \mathcal{A} \to \mathcal{P}(\mathcal{S})$ is the transition kernel for any state-action pair $(s, a) \in \mathcal{S} \times \mathcal{A}$ to the space of distribution on $\mathcal{S}$ denoted as $\mathcal{P}(\mathcal{S})$, $r : \mathcal{S} \times \mathcal{A} \to \mathbb{R}$ is the reward function, and $\gamma \in [0, 1]$ is the discount factor. These yields a Markov chain $(s_0, a_0, s_1, a_1, \cdots)$.

The performance of a policy $\pi : \mathcal{S} \to \mathcal{P}(\mathcal{A})$ is evaluated by the following value function:

$$V^\pi(s) = \mathbb{E}[\sum_{t=0}^\infty \gamma^t r(s_t, a_t) \big| s_0 = s, a_t \sim \pi(\cdot|s_t), s_{t+1} \sim P(\cdot|s_t, a_t)], \tag{1}$$

and the action-value function (Q-function):

$$Q^\pi(s, a) = \mathbb{E}[\sum_{t=0}^\infty \gamma^t r(s_t, a_t) \big| s_0 = s, a_0 = a, a_t \sim \pi(\cdot|s_t), s_{t+1} \sim P(\cdot|s_t, a_t)]. \tag{2}$$

The corresponding advantage function $A^\pi(s, a)$ is define as $A^\pi(s, a) = Q^\pi(s, a) - V^\pi(s)$. In this study, we will focus on learning the Q-function. To this end, we define the Bellman evaluation operator:

$$\mathcal{T}[Q^\pi](s, a) = r(s, a) + \gamma \mathbb{E}_{a' \sim \pi(\cdot|s'), s' \sim P(s'|s,a)}[Q^\pi(s', a')], \tag{3}$$

for which $Q^\pi$ is the fixed point of the operator: $Q^\pi = \mathcal{T}Q$.

We let $\sigma_\mu^\pi(s, a) = \pi(a|s)\mu(s)$ denote the action-state distribution associated to policy $\pi$. Let $\nu^*$ denote the stationary state distribution of the Markov chain given the optimal policy $\pi^*$ and let $\sigma^*(s, a) = \pi^*(a|s)\nu^*(s)$ denote the stationary state-action distribution. When given a policy $\pi$, we first sample $n$ initial samples from a pre-determined initial distribution $\mu_0$. We have the following assumption on $\sigma_0^\pi = \pi\mu_0$.

**Assumption 1.** *Distribution $C \geq \sigma_0^\pi \geq c$ is bounded.*

Assumption 1 is essential for analyzing the generalization error. For instance, even if the empirical loss on the dataset is small, as shown in Theorem 10, an unbounded distribution could produce extreme outliers or unobserved data, making meaningful generalization (Corollary 11) impossible.

***RKHS.*** Define $\omega = (s, a) \in \mathcal{S} \times \mathcal{A}$. We assume the Q-function $Q$ associated with a policy $\pi$ lies in $\mathcal{H}(\mathcal{S} \times \mathcal{A})$ where $\mathcal{H}$ is a RKHS induced by a symmetric positive definite kernel $K : (\mathcal{S} \times \mathcal{A}) \times (\mathcal{S} \times \mathcal{A}) \to \mathbb{R}$. Using kernel $K$, we define the linear space of functions on $\mathcal{S} \times \mathcal{A}$ as follows:

$$\hat{\mathcal{H}} = \{\sum_{i=1}^n b_i K(\omega_i, \cdot), b_i \in \mathbb{R}, \omega_i \in \mathcal{S} \times \mathcal{A}\}$$

and this space is equipped with the bilinear form

$$\langle \sum_{i=1}^n b_i K(\omega_i, \cdot), \sum_{i=1}^n c_i K(\omega_i, \cdot) \rangle_K = \sum_{i,j=1}^n b_i K(\omega_i, \omega_j) c_j.$$

Without loss of generality, we assume that $\max_\omega K(\omega, \omega) < \infty$ is bounded. The RKHS $\mathcal{H}(\mathcal{S} \times \mathcal{A})$ induced by $K$ is defined as the closure of $\hat{\mathcal{H}}$ under the inner product $\langle \cdot, \cdot \rangle_K$ and $\mathcal{H}(\mathcal{S} \times \mathcal{A})$ is equipped with norm $\| \cdot \|_{\mathcal{H}(\mathcal{S} \times \mathcal{A})}$ induced by $\langle \cdot, \cdot \rangle_K$.

Many functional spaces can be represented within the RKHS framework. For example, deep neural networks, Sobolev spaces, the Euclidean space, and a discrete set can all be formulated as RKHSs. For further details, please refer to Adams & Fournier (2003); Wendland (2004); Jacot et al. (2018).

We first impose the following assumption on the MDP $(\mathcal{S}, \mathcal{A}, P, r, \gamma)$

**Assumption 2.** *There exists $R > 0$ such that $\max\{\|r\|_\mathcal{H}, \max_{s \in \mathcal{S}} \|P(s|\cdot, \cdot)\|_\mathcal{H}\} \leq R$.*

In policy optimization, the RKHS norm $\|Q^\pi\|_\mathcal{H}$, which plays a central role, is difficult to determine, as even generating a single observation of $Q^\pi$ can be expensive. Assumption 2 provides a useful alternative by requiring conditions only on the reward $r$, the transition kernel $P(s' \mid s, a)$, and the policy $\pi$, which are typically much easier to verify than condition directly on the Q-function.

**Lemma 3.** *There exists positive constant $C$ that only depends on $R$, such that $\|Q^\pi\|_{\mathcal{H}} \leq C\|\pi(\cdot|\cdot)\|_{\mathcal{H}}$.*

***Covering Number.*** The following measures of RKHS complexity are fundamental to our analysis:

**Definition 4** (Covering number & entropy)**.** *For a given $\delta > 0$, the covering number of a RKHS $\mathcal{H}$ under $L_\infty$ norm, denoted by $\mathcal{N}(\delta, \|\cdot\|_{L_\infty}, \mathcal{H})$, is defined by the smallest integer $M$ sucht that there exists centers $\{f_m\}_{m=1}^M \subseteq \mathcal{H}$ for which $\forall f \in \mathcal{H}, \exists m: \|f_m - f\|_{L_\infty} \leq \delta$. The entropy of $\mathcal{H}$ is the log of its covering number: $H(\delta, \|\cdot\|_{L_\infty}, \mathcal{H}) = \log \mathcal{N}(\delta, \|\cdot\|_{L_\infty}, \mathcal{H})$.*

**Assumption 5.** *The entropy of a unit ball in the RKHS $\mathcal{H}$: $\mathcal{B} = \{f \in \mathcal{H} : \|f\|_{\mathcal{H}} \leq 1\}$ satisfies*

$$H(\delta, \|\cdot\|_{L_\infty}, \mathcal{B}) \leq C\delta^{-2\beta}|\log \delta|^{2\kappa} \tag{4}$$

*for some $C > 0$, $\beta \in [0, 1)$, and $\kappa \geq 0$.*

Assumption 5 covers a broad class of RKHSs, including those previously mentioned, as well as Sobolev spaces with low intrinsic dimension (Ding et al., 2024; Hamm & Steinwart, 2021) and RKHSs induced by deep neural networks (Anthony & Bartlett, 2009).

## 4 POLICY EVALUATION

For policy evaluation, we generalize the TD learning for parameters in a finite-dimensional space to functions in an infinitely-dimensional RKHS. We focus on one-step TD learning and its analysis in the main paper. The kernelized $N$-step TD is presented in Appendix J as it can be viewed as a one-step TD with a discount factor $\gamma^N$, a modified transition kernel $P(s_N \mid s_0, a_0)$, and a modified reward $\bar{r}(s_0, a_0) = \sum_{t=0}^{N-1} \gamma^t r(s_t, a_t)$. Given this equivalence, the convergence analysis for $N$-step TD is readily obtained simply by following the proofs for its one-step counterpart.

### 4.1 OPTIMIZATION FORMULATION

In this paper, we study the problem of estimating the Q function by samples from the Markov chain. In every update of policy $\pi$, we sample i.i.d. $\{s_0^{(i)} \sim \mu_0\}_{i=1}^n$ from some chosen distribution $\mu_0$ and generate the subsequent state and actions $a_0^{(i)}$ and $(s_1^{(i)}, a_1^{(i)})$ following the Markov chain as follows:

$$s_0^{(i)} \sim \mu_0, \quad a_0^{(i)} \sim \pi(\cdot|s_0^{(i)}), \quad s_1^{(i)} \sim P(\cdot|s_0^{(i)}, a_0^{(i)}), \quad a_1^{(i)} \sim \pi(\cdot|s_1^{(i)}). \tag{5}$$

The initial distribution $\mu_0$ can be specified using prior knowledge or obtained by following the chain trajectory and applying MCMC to sample i.i.d. quadruplets $(s_0, a_0, s_1, a_1)$. The resulting distribution is only required to satisfy Assumption 1 together with an additional assumption:

**Assumption 6.** *There exists a constant $c$ s.t. $\int P(s'|s, a)\pi(a|s)\mu_0(s)dsda \leq c\mu_0(s')$ and $c\gamma < 1$.*

**Remark 7.** *Assumption 6 ensures that the one-step transition is not too far from the initial distribution; otherwise, convergence slows. For example, if $\mu$ is the stationary distribution of the MDP, then $c = 1$. The constant $1 - c\gamma$ in the convergence rate reflects sampling complexity, with smaller values indicating higher sample requirements. In the general $N$-step TD setting, Assumption 6 can be relaxed to Assumption 6B in Appendix J. Under this relaxation, the constant $c\gamma$ in our convergence theorems can be replaced by $c\gamma^N$, resulting in faster convergence rates at the cost of requiring additional sampling steps. Please refer to Appendix J for details.*

Given dataset $(\omega_0^{(i)}, \omega_1^{(i)})_{i=1}^n$ with $\omega_j^{(i)} = (s_j^{(i)}, a_j^{(i)})$, $j = 0, 1$ for any given policy $\pi$, we aim to learn the Q-function $Q^\pi$ associated with the Bellman equation 3 by solving the following fixed-point Kernel Ridge Regression (KRR) over the whole RKHS $\mathcal{H}$:

$$\hat{Q}^\pi = \min_{f \in \mathcal{H}} \frac{1}{n} \sum_{i=1}^n \left(f(\omega_0^{(i)}) - r(\omega_0^{(i)}) - \gamma \hat{Q}^\pi(\omega_1^{(i)})\right)^2 + \lambda\|f\|_{\mathcal{H}}^2. \tag{6}$$

The functional minimization problem in equation 6 is implicit. Nevertheless, by the representer theorem, it can be shown to admit a closed-form solution:

**Proposition 8.** *The estimator $\hat{Q}^\pi$ has closed-form solution as follows*

$$\hat{Q}^\pi = K(\cdot, \boldsymbol{\omega}_0)\mathbf{b}^\pi \tag{7}$$

*where $\boldsymbol{\omega}_0 = [\omega_0^{(1)}, \cdots, \omega_0^{(n)}]^\top \in \mathbb{R}^n$, $K(\cdot, \boldsymbol{\omega}_0) = [K(\cdot, \omega_0^{(1)}), \cdots, [K(\cdot, \omega_0^{(n)})]$, and*

$$\mathbf{b}^\pi = [\mathbf{K} + \lambda n \mathbf{I} - \gamma \mathbf{C}]^{-1} \mathbf{r}, \quad \mathbf{K}_{i,j} = K(\omega_0^{(i)}, \omega_0^{(j)}),$$

$$\mathbf{C}_{i,j} = K(\omega_1^{(i)}, \omega_0^{(j)}), \quad \mathbf{r} = [r(\omega_0^{(1)}), \cdots, r(\omega_0^{(n)})]^\top.$$

KRR equation 6 allows us to extend the current standard TD algorithm for tabular learning to the kernel TD framework.

## 4.2 Kernel Temporal-Difference Learning

From Proposition 8, we know that the solution $\hat{Q}^\pi \in \hat{\mathcal{H}}$, so the KRR equation 6 can be rewritten as

$$\mathbf{b}^\pi = \min_{\mathbf{b} \in \mathbb{R}^n} \frac{1}{n} \sum_{i=1}^n \left( K(\omega_0^{(i)}, \boldsymbol{\omega}_0) \mathbf{b} - r(\omega_0^{(i)}) - \gamma K(\omega_1^{(i)}, \boldsymbol{\omega}_0) \mathbf{b}^\pi \right)^2 + \lambda \mathbf{b}^\top \mathbf{K} \mathbf{b}. \tag{8}$$

Instead of directly solving equation 8 for the vector $\mathbf{b}^\pi$ by gradient-based method on $\mathbb{R}^n$, we treat equation 6 as a functional optimization on the RKHS $\mathcal{H}$. Inspired by Kernel Gradient Descent (Ding et al., 2024; Lin & Zhou, 2018; Raskutti et al., 2014), we propose the following updating rule in the RKHS:

$$f_{t+1} = (1 - \alpha_t) f_t - \eta_t \sum_{i=1}^n \left( f_t(\omega_0^{(i)}) - r(\omega_0^{(i)}) - \gamma f_t(\omega_1^{(i)}) \right) K(\omega_0^{(i)}, \cdot), \tag{9}$$

where $\alpha_t$ acts as the weight decay to prevent over-fitting and improve generalization (Hu et al., 2021), and $\eta_t$ is the step-size for the learning rate. It can be noticed that equation 9 can be converted to a form similar to semi-gradient TD(0) (Sutton et al., 1998) if we represented it by $\mathbf{b}_t = \mathbf{K}^{-1} f_t(\boldsymbol{\omega}_0)$:

$$\mathbf{b}_{t+1} = (1 - \alpha_t) \mathbf{b}_t - \eta_t \left( f_t(\boldsymbol{\omega}_0) - \mathbf{r} - \gamma f_t(\boldsymbol{\omega}_1) \right). \tag{10}$$

The difference between equation 10 and semi-gradient TD(0) lies in the fact equation 10 uses a functional gradient in the infinite-dimensional RKHS, while semi-gradient TD(0) uses a gradient of some parameterized functions in a finite-dimensional space.

A natural question is why we don't directly solve the $n$-dimensional equation 8. The reason is that the update rule equation 10 is more efficient because it uses the RKHS inner product $\langle \cdot, \cdot \rangle_\mathcal{H}$ instead of the $l^2$ inner product in $\mathbb{R}^n$. This change in inner product modifies the gradient and Hessian, effectively serving as a preconditioner that can improve the performance of the algorithm (Neuberger, 2009).

From updating rule equation 10, we can derive the convergence pattern of $f_t$ to the target $\hat{Q}^\pi$ for correctly-selected constant weight decay $\alpha_t = \alpha$ and step size $\eta_t = \eta$:

$$\begin{aligned} \mathbf{b}_{t+1} - \mathbf{b}^\pi &= [\mathbf{I} - (\alpha \mathbf{I} + \eta \mathbf{K} - \eta\gamma\mathbf{C})] \left[ \mathbf{b}_t - \eta \left( \alpha \mathbf{I} + \eta \mathbf{K} - \eta\gamma\mathbf{C} \right)^{-1} \mathbf{r} \right] \\ &= [\mathbf{I} - (\alpha \mathbf{I} + \eta \mathbf{K} - \eta\gamma\mathbf{C})] [\mathbf{b}_t - \mathbf{b}^\pi] \\ &= [((1 - \alpha)\mathbf{I} - \eta\mathbf{K} + \eta\gamma\mathbf{C}]^{t+1} [\mathbf{b}_0 - \mathbf{b}^\pi] \end{aligned} \tag{11}$$

If the eigenvalues of $[(1 - \alpha)\mathbf{I} - \eta\mathbf{K} + \eta\gamma\mathbf{C}]$ are small, then $f_t$ converges to $\hat{Q}^\pi$ exponentially fast.

## 4.3 Convergence Analysis of Kernel TD

We first show an error decomposition for the estimator equation 6. Define the difference function $D^\pi = \hat{Q}^\pi - Q^\pi$ and the Bellman residual:

$$\varepsilon_i = r(\omega_0^{(i)}) + \gamma Q^\pi(\omega_1^{(i)}) - Q^\pi(\omega_0^{(i)}). \tag{12}$$

We have the following error decomposition

**Proposition 9** (Statistical-Approximation Error Decomposition).

$$\frac{1}{n} \sum_{i=1}^n \left( \mathcal{D}^\pi(\omega_0^{(i)})^2 - \gamma \mathcal{D}^\pi(\omega_0^{(i)}) \mathcal{D}^\pi(\omega_1^{(i)}) \right) = \frac{1}{n} \sum_{i=1}^n \varepsilon_i \mathcal{D}^\pi(\omega_0^{(i)}) - \lambda \langle \mathcal{D}^\pi, \hat{Q}^\pi \rangle_\mathcal{H}. \tag{13}$$

From the error decomposition on the right-hand side of equation 13, we then can use empirical process (Geer, 2000) to derive the following convergence rate of $\hat{Q}^\pi$ on training data

**Theorem 10.** *Suppose Assumptions 2, 5, and 6 hold. Let $\eta = C_1/n$, $\alpha = \eta\lambda n$, and iteration number $t \geq C_2 \log n \|\mathbf{b}_0 - \mathbf{b}^\pi\|$ for some universal constants $C_1, C_2 > 0$ and $\lambda = \mathcal{O}((1 - c\gamma)^{\frac{\beta}{2+2\beta}} n^{-\frac{1}{2+2\beta}} |\log n|^{\frac{\kappa}{1+\beta}})$, then the kernel TD estimator $f_t$ satisfies*

$$\sqrt{\frac{1}{n}\sum_{i=1}^{n}\left|f_t(\omega_0^{(i)}) - Q^\pi(\omega_0^{(i)})\right|^2} \leq \mathcal{O}_p\left((1-c\gamma)^{-\frac{2+\beta}{2+2\beta}} n^{-\frac{1}{2+2\beta}} |\log n|^{\frac{\kappa}{1+\beta}}\right)\|Q^\pi\|_{\mathcal{H}}, \quad (14)$$

$$\|f_t\|_{\mathcal{H}} \leq \mathcal{O}_p(1)\|Q^\pi\|_{\mathcal{H}}.$$

Theorems 10 yield convergence only on the dataset, not generalization error. Establishing generalization requires additional structural information on the RKHS, and for the following cases, kernelized TD attains the optimal convergence rates for non-parametric estimators (Nemirovski, 2000).

**Corollary 11.** *Suppose Assumptions 1, 2, 5, and 6 hold, then*

***Tabular:*** *: If $\mathcal{S} \times \mathcal{A}$ are uncorrelated discrete set $\{\omega_j : j = 1, \cdots, n^\nu\}$ for some constant $\nu \in (0,1)$, then $K = \delta_{s=s'}\delta_{a=a'}$ and the Kernel TD estimator satisfies*

$$\|f_t - Q^\pi\|_{L_2(\sigma_0^\pi)} \leq \mathcal{O}_p\left((1-c\gamma)^{-1}n^{-\frac{1}{2}}|\log n|^{1/2} + \frac{1}{n^{(1+\nu)/4}}\right)\|Q^\pi\|_{\mathcal{H}}; \quad (15)$$

***Sobolev:*** *If $K = K_\mathcal{S}\delta_{a=a'}$ where $K_\mathcal{S}$ is a Sobolev type kernel with smoothness $m$ and intrinsic dimension $d$ and $\mathcal{A} = \{a\}_{a=1}^A$, then the Kernel TD estimator satisfies*

$$\|f_t - Q^\pi\|_{L_2(\sigma_0^\pi)} \leq \mathcal{O}_p\left((1-c\gamma)^{-\frac{2m+d/2}{2m+d}} n^{-\frac{m}{2m+d}}\right)\|Q^\pi\|_{\mathcal{H}}. \quad (16)$$

***NTK:*** *If $K = N\delta_{a=a'}$ where $N$ is the NTK $N(s,s')$ of a two-layer neural network on a d-sphere $\mathbb{S}^{d-1}$ and $\mathcal{A} = \{a\}_{a=1}^A$, then the Kernel TD estimator satisfies*

$$\|f_t - Q^\pi\|_{L_2(\sigma_0^\pi)} \leq \mathcal{O}_p\left((1-c\gamma)^{-\frac{3d+1}{4d}} n^{-\frac{d+1}{4d}}\right)\|Q^\pi\|_{\mathcal{H}}. \quad (17)$$

***Gaussian:*** *If $\mathcal{H}$ is Gaussian RKHS, i.e., $K = e^{-\|\omega-\omega'\|^2}$ with $\mathcal{S} \times \mathcal{A} = [0,1]^d$ hypercube, then*

$$\|f_t - Q^\pi\|_{L_2(\sigma_0^\pi)} \leq \mathcal{O}_p\left((1-c\gamma)^{-1}n^{-\frac{1}{2}}|\log n|^{(d+1)/2}\right)\|Q^\pi\|_{\mathcal{H}}. \quad (18)$$

## 5 POLICY IMPROVEMENT

In this section, we focus on the following policy update rule:

$$\pi^{k+1} = \underset{\int_\mathcal{A} \pi(a|\cdot)da=1, \pi\geq 0}{\arg\max} \mathbb{E}_n\left[\Delta_k \int_\mathcal{A} f^{(k)}(s,a)\pi(a|s)da - KL\left(\pi(\cdot|s)\|\pi^k(\cdot|s)\right)\right] \quad (19)$$

where $\mathbb{E}_n[f(s)] = \frac{1}{n}\sum_{i=1}^{n} f(s_0^{(i)})$ is the empirical expectation, $KL(p\|q)$ denote the KL divergence, $f^{(k)}$ is the kernel TD estimator of $Q^{(k)} := Q^{\pi^k}$ trained for $T$ iterations, and $\Delta_k$ is the step size.

Equation 19 differs from the original PPO (Schulman et al., 2017), which uses $KL\left(\pi^k(\cdot|s) \| \pi(\cdot|s)\right)$ instead of $KL\left(\pi(\cdot|s) \| \pi^k(\cdot|s)\right)$. Equation 19 has a closed-form solution equivalent to *Natural Policy Gradient (NPG)* so we refer to it as *explicit PPO* or NPG. Analysis for the original PPO is provided in Appendix K. Its solution takes an implicit form analogous to implicit stochastic gradient descent (SGD) (Toulis et al., 2021), and we therefore refer to it as *implicit PPO*.

### 5.1 NATURAL POLICY GRADIENT IN RKHS

For sequence of RKHS function $\{f^{(j)} \in \mathcal{H}\}_{j=0}^k$, the NPG update rule is given by:

$$\pi^{k+1} \propto \pi^k \exp\{\Delta_k f^{(k)}\} \propto \exp\{\sum_{j=0}^{k} \Delta_j f^{(j)}\} \quad (20)$$

**Lemma 12.** *Suppose $\pi^k \propto \exp\{F\}$ for some function $F$ and Assumption 1 holds, then equation 20 is the solution to equation 19 on $\{s_0^{(i)}\} \times \mathcal{A}$.*

The update rule equation 20 generalizes existing NPG methods, including those for tabular RL (Liu et al., 2024), linear RL (Cen et al., 2022), and two-layer neural networks (Liu et al., 2019), to general RKHS functions. Our update rule equation 19 allows $\pi^k$ to be a continuous function of $(s, a)$ and replaces the population expectation with an empirical expectation that is adaptive to the data.

By combining the NPG update rule equation 20 with the kernel TD update rule equation 9, we introduce an efficient algorithm that generalizes NPG to infinite-dimensional RKHS as Algorithm 1.

---

**Algorithm 1** NPG in RKHS

---

1: **Require**: MDP $(\mathcal{S}, \mathcal{A}, P, r, \gamma)$, RKHS $\mathcal{H}$
2: Initialize $\pi^0 \propto \exp\{f^{(0)}\}$ for some $f^{(0)} \in \hat{\mathcal{H}}$, set $F = \Delta_0 f^{(0)}$
3: **for** $k = 0, 1, , 2, \ldots$ **do**
4:      Select an initial sampling distribution $\mu_0^k$ and number of samples $n \leftarrow n^{(k)}$
5:      Generate $\{[s_0^{(i)}, a_0^{(i)}, s_1^{(i)}, a_1^{(i)}] \sim \mu_0^k(s_0)\pi^k(a_0|s_0)P(s_1|s_0, a_0)\pi^k(a_1|s_1)\}_{i=1}^n$
6:      Set $T \leftarrow T^{(k)}, \alpha \leftarrow \alpha^{(k)}, \eta \leftarrow \eta^{(k)}$
7:      Initialize $f^{(k)}$
8:      **for** $t = 1, \cdots, T$ **do**
9:          Update $f^{(k)} \leftarrow (1 - \alpha)f^{(k)} - \eta \sum_{i=1}^n \left( f^{(k)}(\omega_0^{(i)}) - r(\omega_0^{(i)}) - \gamma f^{(k)}(\omega_1^{(i)}) \right) K(\omega_0^{(i)}, \cdot)$
10:      **end for**
11:      $F \leftarrow F + \Delta_k f^{(k)}, \pi^k \propto \exp\{F\}$
12: **end for**

---

From the NTK perspective, Algorithm 1 can be viewed as the evolution of a deep neural network described by kernel $K$. Consequently, it remains valid if we replace the RKHS functions $f^{(k)}$ with a deep neural net $f_{\boldsymbol{\theta}^{(k)}}$ parameterized by $\boldsymbol{\theta}^{(k)}$ and the policy proportional to $\exp(f_{\boldsymbol{\theta}^{(k)}})$.

## 5.2 GLOBAL CONVERGENCE OF NPG

We first define the expected total reward to measure the optimality of a policy $\pi$

$$\mathcal{R}[\pi] = \mathbb{E}_{S \sim \nu^*}[V^\pi(S)] = \mathbb{E}_{S \sim \nu^*}[\langle Q^\pi(S, \cdot), \pi(\cdot|S)\rangle_{\mathcal{A}}]. \tag{21}$$

We can have a fundamental inequality for the equation 21 in RKHS. This inequality, which slightly modifies the mirror descent analysis in Nesterov (2013); Nemirovsky & Yudin (1985) for infinite-dimensional spaces, is central to our analysis.

**Theorem 13.** *In Algorithm 1,*

$$\inf_k \left( \mathcal{R}[\pi^*] - \mathcal{R}[\pi^k] \right)$$
$$\leq \frac{\left( \sum_k 2\Delta_k \|f^{(k)} - Q^{(k)}\|_{L_\infty} \right) + \left( \sum_k \Delta_k^2 (1 - \gamma)^{-1} \|r\|_{L_\infty} \right) + \mathbb{E}_{S \sim \nu^*} KL \left( \pi^*(\cdot|S) || \pi^0(\cdot|S) \right)}{\sum_k \Delta_k}. \tag{22}$$

From Theorem 13, we observe that achieving the optimal $\mathcal{O}(k^{-1/2})$ convergence rate for stochastic optimization requires selecting suitable parameters in Algorithm 1 to ensure the policy evaluation error is well-controlled under the $L_\infty$ norm. This leads to the following corollary.

**Corollary 14.** *Let $\{\pi^k\}_{k=1}^{k^*}$ be the policies induced by Algorithm 1. Set $\Delta_k = 1/\sqrt{k}$. For settings listed in Corollary 11, set $n^{(k)}$ and $\lambda^{(k)}$ according to Table 1. Set $\alpha^{(k)}, \eta^{(k)}$, and $T^{(k)}$ according to Theorem 10 with $\lambda = \lambda^{(k)}$. Then under the same conditions as Corollary 11, we have*

$$\inf_{1 \leq k \leq k^*} \left( \mathcal{R}[\pi^*] - \mathcal{R}[\pi^k] \right) \leq \mathcal{O}_p(\frac{1}{\sqrt{k^*}}). \tag{23}$$

Table 1: Parameters selection in Algorithm 1 for four cases in Corollary 11.

| Setting | $n^{(k)}$ | $\lambda^{(k)}$ |
|---|---|---|
| Tabular | $\mathcal{O}\left( \dfrac{\|\pi^k\|_{\mathcal{H}}^2 k}{(1-c\gamma)^2} \log \dfrac{\|\pi^k\|_{\mathcal{H}} k}{1-c\gamma} + (\sqrt{k}\|\pi^k\|_{\mathcal{H}})^{\frac{4}{1+\nu}} \right)$ | $\mathcal{O}\left( \dfrac{1-c\gamma}{\|\pi^k\|_{\mathcal{H}} \sqrt{k}} \right)$ |
| Sobolev | $\mathcal{O}\left( \dfrac{\|\pi^k\|_{\mathcal{H}}^{\frac{2(2m+d)}{2m-d}} k^{\frac{2m+d}{2m-d}}}{(1-c\gamma)^{\frac{2m+d/2}{m}}} \right)$ | $\mathcal{O}\left( \dfrac{1-c\gamma}{\|\pi^k\|_{\mathcal{H}}^{\frac{2m}{2m-d}} k^{\frac{m}{2m-d}}} \right)$ |
| NTK | $\mathcal{O}\left( \dfrac{\|\pi^k\|_{\mathcal{H}}^{2d} k^d}{(1-c\gamma)^{\frac{3d+1}{d+1}}} \right)$ | $\mathcal{O}\left( \dfrac{1-c\gamma}{\|\pi^k\|_{\mathcal{H}}^{\frac{d+1}{2}} k^{\frac{d+1}{4}}} \right)$ |
| Gaussian | $\mathcal{O}\left( \dfrac{\|\pi^k\|_{\mathcal{H}}^{\frac{2}{1-\epsilon}} k^{\frac{1}{1-\epsilon}}}{(1-c\gamma)^2} \log \dfrac{\|\pi^k\|_{\mathcal{H}} k}{1-c\gamma} \right)$ | $\mathcal{O}\left( \dfrac{(1-c\gamma)}{\|\pi\|_{\mathcal{H}}^{\frac{1}{1-\epsilon}} \sqrt{k}^{\frac{1}{1-\epsilon}}} \right), \ \forall \epsilon \in (0,1)$ |

Corollary 14 shows that the required sampling number increases with both the step size $k$ and the policy complexity, measured by the RKHS norm $\|\pi^k\|_{\mathcal{H}}$. As the policy nears the optimum, the performance gap narrows, requiring more accurate evaluation and hence more samples. As NPG progresses, the policy may also become highly concentrated, approaching a delta distribution; in this case, the RKHS norm diverges, and the sampling requirement grows accordingly.

The RKHS norm is a well-defined and meaningful measure of policy complexity. For example, in the Sobolev and NTK cases, the policy is from composing a softmax with an RKHS function, and remains in the RKHS since the softmax is analytic. Although the RKHS norm is hard to estimate in the NTK case—where the policy corresponds to a deep neural network—other attributes such as network stability and architecture can serve as practical proxies for the RKHS norm complexity.

## 6 EXPERIMENTS

We conduct numerical experiments to empirically validate our theoretical analysis of Q-function estimation within the NPG framework outlined in Algorithm 1. The experiments are designed to investigate the convergence properties of a deep neural network-based implementation and sample efficiency. Our code is available at `https://github.com/conniemessi/TD-and-PPO-in-RKHS`.

### 6.1 EXPERIMENTAL SETUP

**Environments.** We evaluate our method on standard benchmarks: (1) `CartPole-v1` (Barto et al., 2012), a classic low-dimensional control task (4-dim continuous states and 2-dim discrete actions); (2) `Acrobot-v1` (Sutton et al., 1998), a more challenging underactuated robotics problem (6-dim continuous states and 3-dim discrete actions); (3) `HalfCheetah-v5`, a high-dimensional environment in Multi-Joint dynamics with Contact (MuJoCo) with 17-dim continuous states and 6-dim continuous actions. The details of these three environments are in the Appendix L.1.

**Model Architecture.** We instantiate the non-parametric function $f^{(k)}$ from Algorithm 1 with a neural network $f_{\theta^{(k)}}$, where the policy is given by $\pi_\theta \propto \text{Softmax}(f_\theta)$. The network is a simple Multilayer Perceptron (MLP) with two hidden layers. This model is implemented within an Actor-Critic paradigm where the MLP backbone serves as the Critic ($f_\theta$), outputting Q-values, and the addition of a Softmax layer forms the Actor ($\pi_\theta$). The Actor and Critic share all parameters, enabling efficient end-to-end training.

**Algorithm Configuration.** Algorithm 1 outlines a two-stage process: optimizing the Q-function for $T$ epochs, followed by a single policy update. In practice, a finite number of updates (e.g., $T = 4$) is insufficient for the Critic to learn well. Consequently, updating the Actor based on an imperfect Critic leads to inefficient and unstable learning. To address this, we adopt a joint optimization strategy, a standard practice in algorithms like PPO. We combine the Critic's TD Error loss (Line 9) and the Actor's NPG objective (Line 12) into a unified loss function. This objective is then minimized over $T$ epochs for each batch of collected data.

Each outer iteration $k$ proceeds as follows: we execute the current policy to collect $n$ samples, after which we run $T$ optimization epochs over this batch to update the shared actor–critic parameters. Experiments are run for 1000 episodes and averaged over 10 random seeds; we report the mean and standard deviation across seeds. Hyperparameters are shown in Appendix L.4.

## 6.2 CONVERGENCE ANALYSIS

We investigate the empirical impact of the step size schedule, $\Delta_k = k^{-\alpha}$, by testing three theoretically motivated exponents: $\alpha \in \{0.1/0.2, 0.5, 1.5\}$.

**Results Analysis.** Figure 1 corroborates our theoretical predictions, revealing three distinct learning dynamics determined by the schedule $\alpha$. These results empirically confirm that the step size schedule is a critical factor governing the stability and efficiency of NPG with TD learning. The narrow confidence intervals across 10 seeds highlight the statistical reliability of our findings, providing strong evidence for the optimal convergence for the $\Delta_k = \frac{1}{\sqrt{k}}$ schedule in Corollary 14. Moreover, as shown in Figure 1 (b), the learning curve for $\Delta_k = \frac{1}{\sqrt{k}}$ exhibits a decline after some episodes. This occurs because, as the neural network becomes more complex, the available samples are insufficient to fully support training, thereby weakening the effectiveness of optimization. A similar trend emerges when the CartPole experiments run for longer episodes (e.g., Figure 2 (a)). We report the runtime and memory cost for Figure 1 in Appendix L.2, which indicates that for high-dimensional continuous control tasks like HalfCheetah-v5 (17-dim state, 6-dim action), our method maintains reasonable run time and memory.

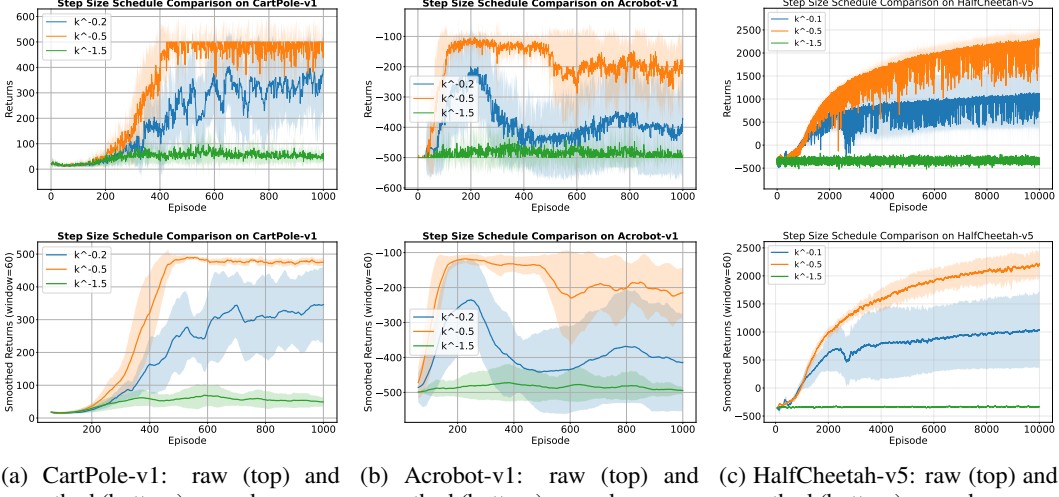

(a) CartPole-v1: raw (top) and smoothed (bottom) rewards.

(b) Acrobot-v1: raw (top) and smoothed (bottom) rewards.

(c) HalfCheetah-v5: raw (top) and smoothed (bottom) rewards.

Figure 1: Convergence analysis on CartPole-v1, Acrobot-v1, and HalfCheetah-v5. Each column stacks raw (top) and smoothed (bottom, window size 60) rewards. All environments show the same trend: $\Delta_k = k^{-0.5}$ converges, $\Delta_k = k^{-0.2}/k^{-0.1}$ diverges, and $\Delta_k = k^{-1.5}$ stagnates.

## 6.3 RUNNING EFFICIENCY ANALYSIS

To analyze the sample efficiency, we compare our Algorithm with the clip-PPO(Schulman et al., 2017). The key architectural difference between these methods lies in their data requirements for value estimation: clip-PPO can be viewed as a second-order approximation of implicit PPO with $t$-step TD and relies on Generalized Advantage Estimation (GAE). GAE requires complete trajectory segments $(s_0, a_0, r_0, s_1, a_1, \ldots, s_t, a_t)$ to recursively compute advantage targets, rendering its update rule non-local in time. Instead, Our NPG employs an action-value critic ($Q(s,a)$) and learns from single-step Temporal Difference (TD) errors. This approach is highly data-efficient, as it only requires $(s_0, a_0, r_0, s_1, a_1)$ to learn Q.

**Results Analysis.** Our experiments demonstrate that our NPG with TD learning significantly outperforms clip-PPO in both sample efficiency and computational efficiency. As shown in Figure 2

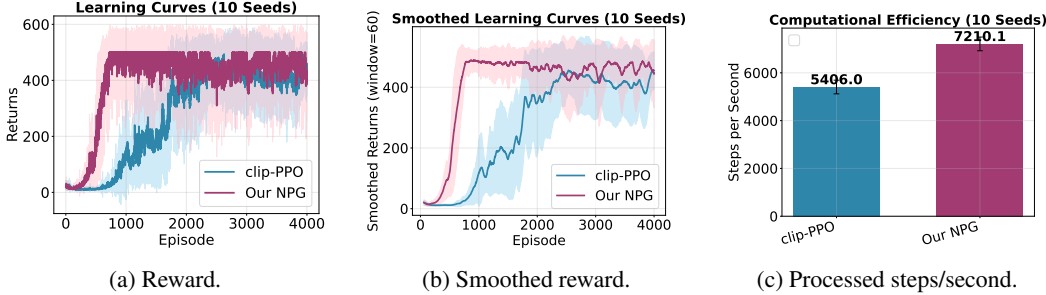

(a) Reward.   (b) Smoothed reward.   (c) Processed steps/second.

Figure 2: Performance and efficiency comparison on CartPole-v1 over 10 seeds.

(a–b), our NPG consistently solves the task, achieving the maximum reward of 500. In contrast, clip-PPO converges more slowly and exhibits high variance, with only some seeds reaching the optimal solution. Since our algorithm depends on sampling quality (Assumption 1 and 6), choosing a suitable distribution enables fast convergence, as confirmed in Figure 2.

Figure 2(c) shows our NPG is more efficient, processing more state-action pairs (i.e. $\omega_t^{(i)}, t = 0, 1$) per second compared to PPO's pairs (i.e. $\omega_t^{(i)}$) per second. This computational advantage stems directly from our algorithm's design. The update for our NPG is based on a computationally lightweight, one-step TD-error calculation. In contrast, clip-PPO must perform a more complex, recursive GAE calculation over entire trajectory segments for each update.

We further compare our NPG with discrete SAC (Haarnoja et al., 2018), DQN (Mnih et al., 2015), and implicit PPO (explained in Appendix K). The results of `Cartpole-v1` in Appendix L.3 Figure 3 shows that implicit PPO shows similar performance as our NPG, and akin to implicit SGD, it results in more stable iterations. Discrete SAC fails to find the optimal solution. DQN is computationally efficient but unstable, converging to the optima only in some seeds.

## 7    CONCLUSIONS AND FUTURE STUDIES

We generalize TD and NPG to infinite-dimensional RKHSs. The framework is illustrated using four commonly studied RKHSs, and numerical experiments further confirm that the theoretical results align well with empirical performance. In specific, we

1. Propose a KRR formulation for solving the functional $N$-step TD problem using finitely many $N$-step trajectory samples, and provide the corresponding sampling complexity;
2. Generalize the PPO update rules to functional minimization problems and derive functional-form solutions for both explicit PPO (NPG) and implicit PPO (the original formulation). Building on these solutions, we establish the sample complexity of RKHS-based PPO required to achieve the $\mathcal{O}_p(k^{-1/2})$ convergence rate.
3. Empirical results validate our theory on both low-dimensional discrete and high-dimensional continuous control benchmarks (`CartPole-v1`, `Acrobot-v1`, `HalfCheetah-v5`), showing that the $k^{-1/2}$ step-size schedule yields fast and stable convergence. Besides, the proposed $N$-step TD extensions show the bias–variance trade-offs.

We also believe that our results can be further improved and generalized. First, the kernelized TD and PPO methods developed in this study may be extended to the more general Mirror Descent methods for policy optimization. Second, our convergence analysis for TD is conducted under the $L_2$ metric to establish minimax optimality. Although the $L_\infty$ metric may be more appropriate for RL setting, the fixed-point KRR formulation for TD differs substantially from the standard KRR formulation. We therefore leave the convergence analysis under the $L_\infty$ metric for future work.

## ACKNOWLEDGEMENTS

We thank the anonymous reviewers and the area chair for their constructive feedback, which helped improve the paper. This work was supported in part by the Key Program of the National Natural Science Foundation of China (NSFC) under Grant No. 72495131; the Shenzhen Stability Science Program 2023; the Shenzhen Science and Technology Program No. JCYJ20250604141038013; and the Longgang District Key Laboratory of Intelligent Digital Economy Security. This work was also supported in part by the National Natural Science Foundation of China under Grant No. 72301076 and the Science and Technology Commission of Shanghai Municipality under Grant No. 23PJ1400800.

## REPRODUCIBILITY STATEMENT

We have included all the experimental design in our paper. Details of the computational setup, including hardware configuration and software environment, network structure, as well as the choice of hyperparameters, are documented in Appendix L.4. We also release our code.

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

# A   PROOF OF LEMMA 3

*Proof.* Form the definition of $Q^\pi$ and triangle inequality, we have

$$\|Q^\pi\|_{\mathcal{H}} \le \sum_{t=0}^{\infty} \gamma^t \|\mathbb{E}[r(s_t, a_t)|s_0, a_0]\|_{\mathcal{H}}. \tag{24}$$

So we only need to prove $\|\mathbb{E}[r(s_t, a_t)|s_0 = \cdot, a_0 = \cdot]\|_{\mathcal{H}} \le C_1 R$ for some $C_1 < \infty$ for all $t$. For the case $t = 0$, we use Assumption 2 to have

$$|\mathbb{E}[r(s_0, a_0)|s_0 = \cdot, a_0 = \cdot]\|_{\mathcal{H}} = \|r\|_{\mathcal{H}} \le R.$$

For general $t$, we rewrite $\mathbb{E}[r(s_t, a_t)|s_0 = \cdot, a_0 = \cdot]$ in an integral form

$$\mathbb{E}[r(s_t, a_t)|s_0, a_0]$$
$$= \int r(s_t, a_t) \left( \prod_{\tau=1}^{t} \pi(a_\tau|s_\tau) P(s_\tau|s_{\tau-1}, a_\tau - 1) \right) P(s_1|s_0, a_0) \pi(a_0|s_0) ds_1 da_1 \cdots ds_t da_t$$
$$= \int r(s_t, a_t) P\left((s_1) \to (s_t, a_t)\right) P(s_1|s_0, a_0) \pi(a_0|s_0) \tag{25}$$

where $P\left((s_1) \to (s_t, a_t)\right)$ is the distribution of $(s_t, a_t)$ conditioned on $s_1$. Let $\{\phi_i\}$ be the eigenfunction of $K$ then the RKHS norms of $P(s|\cdot, \cdot)$ and $\pi$ are then

$$\|P(s|\cdot, \cdot)\|_{\mathcal{H}}^2 = \sum_i p_i(s)^2 \le R, \quad \|\pi\|_{\mathcal{H}}^2 = \sum \pi_i^2 \le R \tag{26}$$

where $p_i(s) = \langle P(s|\cdot, \cdot), \phi_i \rangle_{\mathcal{H}}$ and $\pi_i = \langle \pi, \phi_i \rangle_{\mathcal{H}}$.

The coefficient regarding $\mathbb{E}[r(s_t, a_t)|s_0, a_0]$ projected onto $\phi_i$ is then

$$E_i = \langle \mathbb{E}[r(s_t, a_t)|s_0, a_0], \phi_i \rangle_{\mathcal{H}} = \int r(s_t, a_t) P\left((s_1) \to (s_t, a_t)\right) \langle P(s_1|s_0, a_0) \pi(a_0|s_0), \phi_i \rangle_{\mathcal{H}}$$
$$\le \max_{(s,a) \in \mathcal{S} \times \mathcal{A}} r(s, a) \max_{s \in \mathcal{S}} \langle P(s|s_0, a_0) \pi(a_0|s_0), \phi_i \rangle_{\mathcal{H}}$$
$$\le \max_{\omega \in \mathcal{S} \times \mathcal{A}} K(\omega, \omega) R \max_{s \in \mathcal{S}} \langle P(s|s_0, a_0) \pi(a_0|s_0), \phi_i \rangle_{\mathcal{H}} \tag{27}$$

where the last line is because $\max_\omega r(\omega) = \langle r, K(\cdot, \omega) \rangle_{\mathcal{H}} \le \|K(\cdot, \omega)\|_{\mathcal{H}} \|r\|_{\mathcal{H}} \le \max_\omega K(\omega, \omega) R$.

From equation 27, we only need to calculate the RKHS norm of function $P(s|s_0, a_0) \pi(a_0|s_0)$. From equation 26, we can derive that

$$\|P(s|s_0, a_0) \pi(a_0|s_0)\|_{\mathcal{H}}^2 = \left\| \left( \sum_i p_i(s) \phi_i \right) \left( \sum_i \pi_i \phi_i \right) \right\|_{\mathcal{H}}^2$$
$$= \sum_i p_i(s)^2 \pi_i^2 \le \left( \sup_s \sup_i p_i(s)^2 \right) \|\pi\|_{\mathcal{H}}^2 \le R^2 \|\pi\|_{\mathcal{H}}^2. \tag{28}$$

By combining equation 26 equation 28, we can conclude that

$$\|\mathbb{E}[r(s_t, a_t)|s_0 = \cdot, a_0 = \cdot]\|_{\mathcal{H}} \le C \|\pi\|_{\mathcal{H}}^2. \tag{29}$$

Substituting equation 29 into equation 24, we can have the final result.   $\square$

# B   PROOF OF PROPOSITION 8

*Proof of equation 7.* We first briefly introduce the concept of covariance operators, which is crucial in the proof. We adopt the shorthand notation $\Phi(\omega) = K(\omega, \cdot)$ and $\mu^\pi(\omega, \omega') = \mu^\pi((s, a)), (s', a') = \mu(s) \pi(s|a) P(s'|s, a) \pi(a'|s')$. Then the covariance operator $C_{\omega_0, \omega_0}$ and cross-covariance operator $C_{\omega_0, \omega_1}$ on the RKHS $\mathcal{H}$ are defined as:

$$C_{\omega_0, \omega_0} = \mathbb{E}_{(S,A) \sim \mu(s) \pi(a|s)} [\Phi(S, A) \otimes \Phi(S, A)],$$
$$C_{\omega_0, \omega_1} = \mathbb{E}_{((S,A),(S',A')) \sim \mu^\pi} [\Phi(S, A) \otimes \Phi(S', A')]$$

where $f \otimes g$ denote the tensor product between functions $f$ and $g$. Accordingly, for any $f_0, f_1 \in \mathcal{H}$, $C_{\omega_0,\omega_0}$ and $C_{\omega_0,\omega_1}$ satisfy

$$\langle f_0, C_{\omega_0,\omega_0} f_1 \rangle_{\mathcal{H}} = \mathbb{E}_{(S,A)\sim\mu(s)\pi(a|s)}[f_0(S,A)f_1(S,A)],$$
$$\langle f_0, C_{\omega_0,\omega_1} f_1 \rangle_{\mathcal{H}} = \mathbb{E}_{((S,A),(S',A'))\sim\mu^\pi}[f_0(S,A)f_1(S',A')]].$$

We can also define the empirical version of operators $C_{\omega_0,\omega_0}$ and $C_{\omega_0,\omega_1}$ as follows:

$$\hat{C}_{\omega_0,\omega_0} = \frac{1}{n}\sum_{i=1}^{n}\Phi(\omega_0^{(i)})\otimes\Phi(\omega_0^{(i)}), \quad \hat{C}_{\omega_0,\omega_1} = \frac{1}{n}\sum_{i=1}^{n}\Phi(\omega_0^{(i)})\otimes\Phi(\omega_1^{(i)}).$$

With the concept of covariance operator, we can have the following lemma which can yield a specific formulation for the solution of equation 6:

**Lemma 15.** *Solution to equation 6 is equivalent to the following equation:*

$$\hat{C}_{\omega_0,\omega_0}\hat{Q}^\pi - \left[\hat{C}_{\omega_0,\omega_0}r + \gamma\hat{C}_{\omega_0,\omega_1}\hat{Q}^\pi\right] + \lambda\hat{Q}^\pi = 0. \tag{30}$$

From Lemma 15, we can prove that the solution $\hat{Q}^\pi$ resides in a finite-dimensional space:

**Lemma 16.** *Let $\hat{\mathcal{H}} = \{K(\omega_i, \cdot), i = 1, \cdots, n\}$ be the finite dimensional space spanned by kernel functions. We have $\hat{Q}^\pi \in \hat{\mathcal{H}}$.*

From Lemma 16, we know that $\hat{Q}^\pi$ must be of the form:

$$\hat{Q}^\pi = \sum_{i=1}^{n} K(\cdot, \omega_i)b_i = K(\cdot, \boldsymbol{\omega}_0)\mathbf{b}. \tag{31}$$

Substitute equation 31 into equation 30, togethe with the definition of empirical operators $\hat{C}_{\omega_0,\omega_0}$ and $\hat{C}_{\omega_0,\omega_1}$, we then have

$$\frac{1}{n}\sum_{i=1}^{n}\Phi(\omega_0^{(i)})\sum_{j=1}^{n}b_j K(\omega_0^{(i)},\omega_0^{(j)}) - \frac{1}{n}\sum_{i=1}^{n}\Phi(\omega_0^{(i)})\left[r(\omega_0^{(i)}) + \gamma\sum_{j=1}^{n}b_j K(\omega_1^{(i)},\omega_0^{(j)})\right] \tag{32}$$

$$+ \lambda\sum_{i=1}^{n}b_i\Phi(\omega_0^{(i)}) = 0.$$

Rearrange equation 32 and write it in vector form, we have

$$\Phi(\boldsymbol{\omega}_0)\left[\mathbf{K}\mathbf{b} + \lambda n\mathbf{b} - \gamma\mathbf{C}\mathbf{b}\right] = \Phi(\boldsymbol{\omega}_0)\mathbf{r}. \tag{33}$$

So, we have

$$\mathbf{b} = [\mathbf{K} + \lambda n\mathbf{I} - \gamma\mathbf{C}]^{-1}\mathbf{r}.$$

$\square$

### B.1 PROOF OF LEMMA 15

*Proof.* Solution to the KRR equation 6 is the minimizer of the following functional

$$\mathcal{J}[f] = \frac{1}{n}\sum_{i=1}^{n}\left(f(\omega_0^{(i)}) - r(\omega_0^{(i)}) - \gamma\hat{Q}^\pi(\omega_1^{(i)})\right)^2 + \lambda\|f\|_{\mathcal{H}}^2. \tag{34}$$

The functional $\mathcal{J}$ is Fréchet differentiable obviously with RKHS derivative $\nabla_f\mathcal{J}[f]$ as follows:

$$\frac{1}{2}\partial_\delta\mathcal{J}[f + \delta u]\big|_{\delta=0} \tag{35}$$

$$=\langle\frac{1}{n}\sum_{i=1}^{n}\left(f(\omega_0^{(i)}) - r(\omega_0^{(i)}) - \gamma\hat{Q}^\pi(\omega_1^{(i)})\right)K(\omega_0^{(i)},\cdot) + \lambda f, u\rangle_{\mathcal{H}},$$

$$=\langle\nabla_f\mathcal{J}[f], u\rangle_{\mathcal{H}}, \quad \forall u \in \mathcal{H}. \tag{36}$$

From the definition of empirical covariance operators, we can notice that

$$\frac{1}{n} \sum_{i=1}^{n} \left( f(\omega_0^{(i)}) - r(\omega_0^{(i)}) \right) K(\omega_0^{(i)}, \cdot) = \hat{C}_{\omega_0, \omega_0}[f - r], \tag{37}$$

$$\frac{1}{n} \sum_{i=1}^{n} \hat{Q}^\pi(\omega_1^{(i)}) K(\omega_0^{(i)}, \cdot) = \hat{C}_{\omega_0, \omega_1} \hat{Q}^\pi. \tag{38}$$

Substitute equation 37 and equation 38 into equation 35, we then have:

$$\nabla_f \mathcal{J}[f] = \hat{C}_{\omega_0, \omega_0} f - \left[ \hat{C}_{\omega_0, \omega_0} r + \gamma \hat{C}_{\omega_0, \omega_1} \hat{Q}^\pi \right] + \lambda f. \tag{39}$$

According to definition, $\hat{Q}^\pi$ is also the minimizer of $\mathcal{J}$. Substitute $f = \hat{Q}^\pi$ into equation 39, we can conclude that the minimization of equation 34 is equivalent to equation 6. $\qquad \square$

### B.2 PROOF OF LEMMA 16

*Proof.* Rearrange equation 30 as follows

$$\lambda \hat{Q}^\pi = \hat{C}_{\omega_0, \omega_0} r + \gamma \hat{C}_{\omega_0, \omega_1} \hat{Q}^\pi - \hat{C}_{\omega_0, \omega_0} \hat{Q}^\pi. \tag{40}$$

According to equation 37 and equation 38, the right hand side of equation 40 can be written as

$$\hat{C}_{\omega_0, \omega_0} \left[ r - \hat{Q}^\pi \right] + \gamma \hat{C}_{\omega_0, \omega_1} \hat{Q}^\pi$$
$$= \frac{1}{n} K(\cdot, \omega_0^{(i)}) \left[ r(\omega_0^{(i)}) - \hat{Q}^\pi(\omega_0^{(i)}) + \hat{Q}^\pi(\omega_1^{(i)}) \right] \in \hat{\mathcal{H}}.$$

The right-hand side of equation 40 resides in $\hat{\mathcal{H}}$. So, the left hand side $\hat{Q}^\pi \in \hat{\mathcal{H}}$. $\qquad \square$

## C PROOF OF PROPOSITION 9

*Proof.* According to the definition of the covariance operator, the Bellman equation equation 3 is equivalent to the following form:

$$C_{\omega_0, \omega_0} Q^\pi = C_{\omega_0, \omega_0} r + \gamma C_{\omega_0, \omega_1} Q^\pi. \tag{41}$$

Substitute equation 41 into Lemma 15, we can have

$$\left[ \hat{C}_{\omega_0, \omega_0} + \lambda \mathrm{I} \right] \hat{Q}^\pi - C_{\omega_0, \omega_0} Q^\pi = \left[ \hat{C}_{\omega_0, \omega_0} - C_{\omega_0, \omega_0} \right] r + \gamma \left[ \hat{C}_{\omega_0, \omega_1} \hat{Q}^\pi - C_{\omega_0, \omega_1} Q^\pi \right]. \tag{42}$$

Recall that $\mathcal{D}^\pi = \hat{Q}^\pi - Q^\pi$, we can rewrite equation 42 as

$$\left[ \hat{C}_{\omega_0, \omega_0} + \lambda \mathrm{I} - \gamma \hat{C}_{\omega_0, \omega_1} \right] \mathcal{D}^\pi \tag{43}$$
$$= \left[ \hat{C}_{\omega_0, \omega_0} + \lambda \mathrm{I} - C_{\omega_0, \omega_0} \right] (r - Q^\pi) - \lambda r + \gamma \left[ \hat{C}_{\omega_0, \omega_1} - C_{\omega_0, \omega_1} \right] Q^\pi$$
$$= \left[ \hat{C}_{\omega_0, \omega_0} + \lambda \mathrm{I} \right] (r - Q^\pi) + \gamma \hat{C}_{s_0, s_1} Q^\pi - \lambda r$$
$$\quad + \left( C_{\omega_0, \omega_0} (Q^\pi - r) - \gamma C_{\omega_0, \omega_1} Q^\pi \right)$$
$$= \left[ \hat{C}_{\omega_0, \omega_0} + \lambda \mathrm{I} \right] (r - Q^\pi) + \gamma \hat{C}_{\omega_0, \omega_1} Q^\pi - \lambda r \tag{44}$$

where the last line is from equation 41. Taking the Hilbert space inner product with $\mathcal{D}^\pi$ in equation 43 and equation 44:

$$\langle \mathcal{D}^\pi, \left[ \hat{C}_{\omega_0, \omega_0} + \lambda \mathrm{I} - \gamma \hat{C}_{\omega_0, \omega_1} \right] \mathcal{D}^\pi \rangle_{\mathcal{H}} \tag{45}$$
$$= \langle \mathcal{D}^\pi, \left[ \hat{C}_{\omega_0, \omega_0} + \lambda \mathrm{I} \right] (r - Q^\pi) + \gamma \hat{C}_{\omega_0, \omega_1} Q^\pi - \lambda r \rangle_{\mathcal{H}}. \tag{46}$$

According to the definition of empirical covariance operators, equation 45 can be further rewritten as

$$\langle \mathcal{D}^\pi, \left[\hat{C}_{\omega_0,\omega_0} + \lambda \mathrm{I} - \gamma \hat{C}_{\omega_0,\omega_1}\right] \mathcal{D}^\pi \rangle_{\mathcal{H}} \tag{47}$$

$$= \frac{1}{n}\sum_{i=1}^{n} \mathcal{D}^\pi(\omega_0^{(i)})^2 - \frac{\gamma}{n}\sum_{i=1}^{n}\mathcal{D}^\pi(\omega_0^{(i)})\mathcal{D}^\pi(\omega_1^{(i)}) + \lambda\|\mathcal{D}^\pi\|_{\mathcal{H}}^2$$

and equation 46 can be further rewritten as

$$\langle \mathcal{D}^\pi, \left[\hat{C}_{\omega_0,\omega_0} + \lambda_K \mathrm{I}\right](r - Q^\pi) + \gamma \hat{C}_{\omega_0,\omega_1}Q^\pi - \lambda r \rangle_{\mathcal{H}} \tag{48}$$

$$= \langle \mathcal{D}^\pi, \hat{C}_{\omega_0,\omega_0}(r - Q^\pi) + \gamma \hat{C}_{\omega_0,\omega_1}Q^\pi \rangle_{\mathcal{H}} - \lambda\langle\mathcal{D}^\pi, Q^\pi\rangle_{\mathcal{H}}.$$

Because equation 47 and equation 48 are equals, by rearranging their terms, we have

$$\frac{1}{n}\sum_{i=1}^{n}\mathcal{D}^\pi(\omega_0^{(i)})^2 - \frac{\gamma}{n}\sum_{i=1}^{n}\mathcal{D}^\pi(\omega_0^{(i)})\mathcal{D}^\pi(\omega_1^{(i)}) \tag{49}$$

$$= \langle \mathcal{D}^\pi, \hat{C}_{\omega_0,\omega_0}(r - Q^\pi) + \gamma \hat{C}_{\omega_0,\omega_1}Q^\pi \rangle_{\mathcal{H}} - \lambda\langle\mathcal{D}^\pi, \hat{Q}^\pi\rangle_{\mathcal{H}}. \tag{50}$$

We use the definitions of empirical and population covariance operators again, the first term of equation 50 is

$$\langle \mathcal{D}^\pi, \hat{C}_{\omega_0,\omega_0}(r - Q^\pi) + \gamma \hat{C}_{\omega_0,\omega_1}Q^\pi \rangle_{\mathcal{H}} \tag{51}$$

$$= \frac{1}{n}\sum_{i=1}^{n}\mathcal{D}^\pi(\omega_0^{(i)})\left(r(\omega_0^{(i)}) - Q^\pi(\omega_0^{(i)}) + \gamma Q^\pi(\omega_1^{(i)})\right) = \frac{1}{n}\sum_{i=1}^{n}\varepsilon_i\mathcal{D}^\pi(\omega_0^{(i)}).$$

Substituting equation 51 into equation 50, we can have the final result. $\qquad\square$

## D  PROOF OF THEOREM 10

In order to prove Theorem 10, we first need the following theorem, which we will provide its proof in Appendix E.

**Theorem 17.** *Let* $\lambda = \mathcal{O}((1 - c\gamma)^{\frac{\beta}{2+2\beta}}n^{-\frac{1}{2+2\beta}}|\log n|^{\frac{\kappa}{1+\beta}})$. *Under Assumptions 1, 2, 5, and 6 we can have*

$$\sqrt{\frac{1}{n}\sum_{i=1}^{n}\left|\mathcal{D}^\pi(\omega_0^{(i)})\right|^2} \leq \mathcal{O}_p\left((1 - c\gamma)^{-\frac{2+\beta}{2+2\beta}}n^{-\frac{1}{2+2\beta}}|\log n|^{\frac{\kappa}{1+\beta}}\right)\|Q^\pi\|_{\mathcal{H}}, \tag{52}$$

$$\|\hat{Q}^\pi\|_{\mathcal{H}} \leq \mathcal{O}_p(1)\|Q^\pi\|_{\mathcal{H}}.$$

*Proof of Theorem 10:* By triangle inequality and Theorem 17, we have

$$\begin{aligned}\|f_t - Q^\pi\|_n &\leq \|f_t - \hat{Q}^\pi\|_n + \|\hat{Q}^\pi - Q^\pi\|_n \\ &\leq \|f_t - \hat{Q}^\pi\|_n + \mathcal{O}_p\left((1 - c\gamma)^{-\frac{2+\beta}{2+2\beta}}n^{-\frac{1}{2+2\beta}}|\log n|^{\frac{\kappa}{1+\beta}}\right),\end{aligned} \tag{53}$$

and

$$\|f_t\|_{\mathcal{H}} \leq \|\hat{Q}^\pi\|_{\mathcal{H}} + \|f_t - \hat{Q}^\pi\|_{\mathcal{H}} + \|\hat{Q}^\pi - Q^\pi\|_{\mathcal{H}} \leq \mathcal{O}_p(1)\|Q^\pi\|_{\mathcal{H}} + \|f_t - \hat{Q}^\pi\|_{\mathcal{H}}. \tag{54}$$

Therefore, we only need to prove that the convergence rates of $f_t$ to $\hat{Q}^\pi$ under $\|\cdot\|_n$ and $\|\cdot\|_{\mathcal{H}}$ have the same order as their counterpart in Theorem 17.

Note that by the selections of decay weight $\alpha$ and step size $\eta$, $\alpha/\eta = \lambda$. Then we can use equation 11 and Hölder inequality to have

$$\begin{aligned}\|f_t - \hat{Q}^\pi\|_n^2 &= \frac{1}{n}(\mathbf{b}_t - \mathbf{b}^\pi)^\top \mathbf{K}^2(\mathbf{b}_t - \mathbf{b}^\pi) \\ &\leq \left(\max_\omega K(\omega,\omega)\right)^2 n(\mathbf{b}_t - \mathbf{b}^\pi)^2 \\ &= Cn\left([((1-\alpha)\mathrm{I} - \eta\mathbf{K} + \eta\gamma\mathbf{C}]^t[\mathbf{b}_0 - \mathbf{b}^\pi]\right)^2 \\ &\leq Cn\Lambda_2^t(\mathbf{b}_0 - \mathbf{b}^\pi)^2\end{aligned} \tag{55}$$

where $\Lambda_2$ is the maximum eigenvalue $[((1-\alpha)\mathbf{I} - \eta\mathbf{K} + \eta\gamma\mathbf{C}]^\top [((1-\alpha)\mathbf{I} - \eta\mathbf{K} + \eta\gamma\mathbf{C}]$ and $C = (\max_\omega K(\omega,\omega))^2$.

Note that the maximum and minimum eigenvalue of the matrix $\mathbf{K} - \gamma\mathbf{C}$ is upper bounded by $(1+\gamma)\max_\omega K(\omega,\omega)n$ and lower bounded by $-\gamma\max_\omega K(\omega,\omega)n$, respectively. Therefore, if we can select $\eta$ such that

$$
\begin{aligned}
(1 - \alpha - (1+\gamma)\max_\omega K(\omega,\omega)n\eta) &\le C_1 < 1. \\
(1 - \alpha + \gamma\max_\omega K(\omega,\omega)n\eta) &\le C_2 < 1,
\end{aligned}
\tag{56}
$$

then $\Lambda_2 \le C_3^2 < 1$ where $C_3 = \max\{C_1, C_2\}$. We also need an extra condition so that $f_t$ converges to the estimator $\hat{Q}^\pi$ with correctly selected $\lambda$ as in Theorem 17, we also have

$$
\alpha = \eta\lambda n.
\tag{57}
$$

Solving equation 56 and equation 57 to get

$$
0 < \eta \le \frac{1 - C_1}{n(1+\gamma)C + \lambda}
$$

and then by letting $t = \log_{C_3}\left([n(\mathbf{b}_0 - \mathbf{b}^\pi)^2]^{-1}\lambda\right)$, we have

$$
\|f_t - \hat{Q}^\pi\|_n^2 \le Cn(\mathbf{b}_0 - \mathbf{b}_1)^2 C_3^t = Cn(\mathbf{b}_0 - \mathbf{b}^\pi)^2 C_3^{\log_{C_3}[n(\mathbf{b}_0 - \mathbf{b}^\pi)]^{-1}\lambda} \le \mathcal{O}(\lambda).
\tag{58}
$$

Similarly, for the convergence in RKHS norm, we have

$$
\begin{aligned}
\|f_t - \hat{Q}^\pi\|_{\mathcal{H}}^2 &= (\mathbf{b}_t - \mathbf{b}^\pi)^\top \mathbf{K} (\mathbf{b}_t - \mathbf{b}^\pi) \\
&\le \left(\max_\omega K(\omega,\omega)\right) n (\mathbf{b}_t - \mathbf{b}^\pi)^2 \\
&= \sqrt{C}n\left([((1-\alpha)\mathbf{I} - \eta\mathbf{K} + \eta\gamma\mathbf{C}]^t [\mathbf{b}_0 - \mathbf{b}^\pi]\right)^2 \\
&\le \sqrt{C}n\Lambda_2^t(\mathbf{b}_0 - \mathbf{b}^\pi)^2.
\end{aligned}
\tag{59}
$$

Note that the order of equation 59 is the same as that of equation 55 except that the constant term is changed from $C$ to $\sqrt{C}$. Therefore, the same iteration number $t$ is enough for the following convergence

$$
\|f_t - \hat{Q}^\pi\|_{\mathcal{H}}^2 \le \mathcal{O}(\lambda).
\tag{60}
$$

Substitute equation 58 into equation 53 and equation 60 into equation 54, we can have the final results. □

# E    PROOF OF THEOREM 17

We first slightly modify the error decomposition equation 13 to get the following inequality

$$
\begin{aligned}
&\frac{1}{n}\sum_{i=1}^n \left(\mathcal{D}^\pi(\omega_0^{(i)})^2 - \gamma\mathcal{D}^\pi(\omega_0^{(i)})\mathcal{D}^\pi(\omega_1^{(i)})\right) + \lambda\|\hat{Q}^\pi\|_{\mathcal{H}}^2 \\
&= \frac{1}{n}\sum_{i=1}^n \varepsilon_i\mathcal{D}^\pi(\omega_0^{(i)}) + \lambda\langle Q^\pi, \hat{Q}^\pi\rangle_{\mathcal{H}}.
\end{aligned}
\tag{61}
$$

By further rewriting equation 61, we obtain the oracle that is essential to the proof:

$$
\frac{1}{n}\sum_{i=1}^n \left(\mathcal{D}^\pi(\omega_0^{(i)})^2 - \gamma\mathcal{D}^\pi(\omega_0^{(i)})\mathcal{D}^\pi(\omega_1^{(i)})\right) + \lambda\|\hat{Q}^\pi\|_{\mathcal{H}}^2 = \underbrace{\frac{1}{n}\sum_{i=1}^n \varepsilon_i\mathcal{D}^\pi(\omega_0^{(i)})}_{\text{variance}} + \underbrace{\lambda\langle\hat{Q}^\pi, Q^\pi\rangle_{\mathcal{H}}^2}_{\text{bias}}.
\tag{62}
$$

For the left-hand side of equation 62, we can use the standard central limit theorem for Monte Carlo integration to have

$$
\begin{aligned}
\gamma \frac{1}{n} \sum_{i=1}^{n} \mathcal{D}^{\pi}(\omega_0^{(i)}) \mathcal{D}^{\pi}(\omega_1^{(i)}) &\leq \gamma \big| \mathbb{E}[\mathcal{D}^{\pi}(\omega_0) \mathcal{D}^{\pi}(\omega_1)] \big| + \mathcal{O}_p(n^{-\frac{1}{2}}) \, |\mathbb{E}[\mathcal{D}^{\pi}(\omega_0) \mathcal{D}^{\pi}(\omega_1)]| \\
&\leq \left( \gamma + \mathcal{O}_p(n^{-\frac{1}{2}}) \right) \sqrt{\mathbb{E}[|\mathcal{D}^{\pi}(\omega_0)|^2] \mathbb{E}[|\mathcal{D}^{\pi}(\omega_1)|^2]} \\
&\leq c \left( \gamma + \mathcal{O}_p(n^{-\frac{1}{2}}) \right) \mathbb{E}[|\mathcal{D}^{\pi}(\omega_0)|^2] \\
&\leq c\gamma \frac{1}{n} \sum_{i=1}^{n} \mathcal{D}^{\pi}(\omega_0^{(i)})^2 + \mathcal{O}_p(n^{-\frac{1}{2}}) \mathbb{E}[|\mathcal{D}^{\pi}(\omega_0)|^2],
\end{aligned}
\tag{63}
$$

where the third line of equation 63 follows from Assumption 6 that

$$
\begin{aligned}
\mathbb{E}[|\mathcal{D}^{\pi}(\omega_1)|^2] &= \int |\mathcal{D}^{\pi}(s_1, a_1)|^2 \, \pi(a_1|s_1) P(s_1|s_0, a_0) \pi(a_0|s_0) \mu(s_0) ds_1 da_1 ds_0 da_0 \\
&= \int |\mathcal{D}^{\pi}(s_1, a_1)|^2 \, \pi(a_1|s_1) \left[ \int P(s_1|s_0, a_0) \pi(a_0|s_0) \mu(s_0) ds_0 da_0 \right] ds_1 da_1 \\
&\leq c \int |\mathcal{D}^{\pi}(s_1, a_1)|^2 \, \pi(a_1|s_1) \mu(s_1) ds_1 da_1 \\
&= c^2 \mathbb{E}[|\mathcal{D}^{\pi}(\omega_0)|^2].
\end{aligned}
$$

For the right-hand side of equation 62, we use empirical processes (Geer, 2000) to estimate the variance term in equation 62. In order to do so, we first show that the Bellman residual is zero-mean sub-Gaussian. From the definition of Bellman operator equation 3, the expectation of the Bellman residual is 0, i.e., $\mathbb{E}\varepsilon_i = 0$ for all $i = 1, \cdots, n$. From Assumption equation 2 and Lemma 3, we can know that the Bellman residual is bounded:

$$
|r(\omega_0^i) + \gamma Q^{\pi}(\omega_1^{(i)}) - Q^{\pi}(\omega_0^{(i)})| \leq \|r\|_{\mathcal{H}} + (1-\gamma)\|Q^{\pi}\|_{\mathcal{H}}
$$

where the right-hand side because $\max_\omega f(\omega) = \langle f, K(\cdot, \omega)\rangle_{\mathcal{H}} \leq \|K(\cdot, \omega)\|_{\mathcal{H}} \|f\|_{\mathcal{H}}$ for any $f \in \mathcal{H}$. So $\varepsilon_i$ is sub-Gaussian.

For natation simplicity, first define the empirical inner product and empirical norm for any pair of functions $f_1$ and $f_2$ defined on $\{\omega_0^{(i)}\}$:

$$
\langle f_1, f_2 \rangle_n = \frac{1}{n} \sum_{i=1}^{n} f_1(\omega_0^{(i)}) f_2(\omega_0^{(i)}), \quad , \|f_1\|_n^2 = \langle f_1, f_1 \rangle_n.
$$

The sub-Gaussianity of Bellman residual $\{\varepsilon_i\}$ allows us to apply the following lemma to estimate the variance terms.

**Lemma 18.** *Suppose the RKHS $\mathcal{H}$ satisfies Assumption 5. Suppose that $\{\varepsilon_i\}_{i=1}^{n}$ are Bellman residual as defined in equation 12. Then, for all $t$ large enough,*

$$
\sup_{f \in \mathcal{H}} \frac{n^{\frac{1}{2}} \langle \varepsilon, f \rangle_n}{\|f\|_n^{1-\beta} \|f\|_{\mathcal{H}}^{\beta} \left| \log \frac{\|f\|_n}{\|f\|_{\mathcal{H}}} \right|^{\kappa}} > t,
\tag{64}
$$

*with probability at most $C_2 \exp(-C_1 t^2)$ with some positive constants $C_1$ and $C_2$.*

We can substitute equation 62 and equation 64 in Lemma 18 into the oracle equation 62:

$$
\begin{aligned}
&(1 - c\gamma)\|\mathcal{D}^{\pi}\|_n^2 + \lambda \|\hat{Q}^{\pi}\|_{\mathcal{H}}^2 \\
&\leq \mathcal{O}_p\left(n^{-1/2}\right) \|\mathcal{D}^{\pi}\|_n^{1-\beta} \|\mathcal{D}^{\pi}\|_{\mathcal{H}}^{\beta} \left| \log \frac{\|\mathcal{D}^{\pi}\|_n}{\|\mathcal{D}^{\pi}\|_{\mathcal{H}}} \right|^{\kappa} + \lambda \langle Q^{\pi}, \hat{Q}^{\pi} \rangle_{\mathcal{H}}.
\end{aligned}
\tag{65}
$$

**Case I** $\|\hat{Q}^{\pi}\|_{\mathcal{H}} \geq \|Q^{\pi}\|_{\mathcal{H}}$, equation 65 yields two subcases, either

$$
\begin{aligned}
&(1 - c\gamma))\|\mathcal{D}^{\pi}\|_n^2 + \lambda \|\hat{Q}^{\pi}\|_{\mathcal{H}}^2 \\
&\leq \mathcal{O}_p\left(n^{-1/2}\right) \|\mathcal{D}^{\pi}\|_n^{1-\beta} \|\hat{Q}^{\pi}\|_{\mathcal{H}}^{\beta} \left| |\log \|\mathcal{D}^{\pi}\|_n| + |\log \|\hat{Q}^{\pi}\|_{\mathcal{H}}| \right|^{\kappa}
\end{aligned}
\tag{66}
$$

or

$$(1 - c\gamma)\|\mathcal{D}^\pi\|_n^2 + \lambda\|\hat{Q}^\pi\|_{\mathcal{H}}^2 \leq \lambda\langle Q^\pi, \hat{Q}^\pi\rangle_{\mathcal{H}} + \mathcal{O}_p(n^{-\frac{1}{2}})\|\hat{Q}^\pi\|_{\mathcal{H}}^2. \tag{67}$$

Solving equation 66 yields

$$\begin{aligned}
\|\mathcal{D}^\pi\|_n &= (1 - c\gamma)^{\frac{\beta}{2}-1}\mathcal{O}_p(n^{-\frac{1}{2}})\lambda^{-\beta}(|\log n| + |\log \lambda|)^\kappa, \\
\|\hat{Q}^\pi\|_{\mathcal{H}} &= (1 - c\gamma)^{\frac{\beta-1}{2}}\mathcal{O}_p(n^{-\frac{1}{2}})\lambda^{-\frac{1+\beta}{2}}(|\log n| + |\log \lambda|)^\kappa.
\end{aligned} \tag{68}$$

Solving equation 67 yields

$$\begin{aligned}
\|\mathcal{D}^\pi\|_n &\leq \frac{2}{1 - c\gamma}\lambda\|Q^\pi\|_{\mathcal{H}}, \\
\|\hat{Q}^\pi\|_{\mathcal{H}} &\leq 2\|Q^\pi\|_{\mathcal{H}}.
\end{aligned} \tag{69}$$

**Case II** $\|\hat{Q}^\pi\|_{\mathcal{H}} \leq \|Q^\pi\|_{\mathcal{H}}$, equation 65 yields another two subcases, either

$$(1 - c\gamma)\|\mathcal{D}^\pi\|_n^2 \leq \mathcal{O}_p(n^{-1/2})\|\mathcal{D}^\pi\|_n^{1-\beta}\|Q^\pi\|_{\mathcal{H}}^\beta, \tag{70}$$

or

$$(1 - c\gamma)\|\mathcal{D}^\pi\|_n^2 \leq 2\lambda\|Q^\pi\|_{\mathcal{H}}^2. \tag{71}$$

It follows that either equation 70 holds or

$$\|\mathcal{D}^\pi\|_n \leq (1 - c\gamma)^{\frac{-1}{1+\beta}}\mathcal{O}_p(n^{-\frac{1}{2+2\beta}}). \tag{72}$$

Now we can summarize equation 69, equation 70, equation 71, and equation 72. We can notice that by selecting

$$\lambda = \mathcal{O}((1 - c\gamma)^{\frac{\beta}{2+2\beta}}n^{-\frac{1}{2+2\beta}}|\log n|^{\frac{\kappa}{1+\beta}}), \tag{73}$$

we can have the final result,

$$\|\mathcal{D}^\pi\|_n \leq \mathcal{O}_p\left((1 - c\gamma)^{-\frac{2+\beta}{2+2\beta}}n^{-\frac{1}{2+2\beta}}|\log n|^{\frac{\kappa}{1+\beta}}\right)\|Q^\pi\|_{\mathcal{H}}, \tag{74}$$

$$\|\hat{Q}^\pi\|_{\mathcal{H}} \leq \mathcal{O}_p(1)\|Q^\pi\|_{\mathcal{H}}. \tag{75}$$

### E.1 PROOF OF LEMMA 18

To proof Lemma 18, we first need the following lemma from Geer (2000)

**Lemma 19** (Corollary 8.3 in Geer (2000)). *Let $a > 0$, $R > 0$, and $\mathcal{H}$ be a function space. Suppose that $\sup_{g \in \mathcal{H}}\|f\|_{L_\infty} \leq R$ for and $\varepsilon = \{\varepsilon_i\}_{i=1}^n$ are independent zero-mean sub-Gaussian random variables. If there exists some universal positive constant $C$ satisfying*

$$a \geq Cn^{-\frac{1}{2}}\left(\int_0^R H^{\frac{1}{2}}(u, \|\cdot\|_{L_\infty}, \mathcal{H})du \vee R\right), \tag{76}$$

*then we have*

$$\mathbb{P}\left(\sup_{f \in \mathcal{H}}|\langle\varepsilon, f\rangle_n| \geq a\right) \leq 2\exp\left(-\tilde{C}\frac{na^2}{R^2}\right), \tag{77}$$

*for some positive $\tilde{C}$.*

*Proof.* Let $\mathcal{B} = \{f : \|f\|_{\mathcal{H}} \leq 1\}$ be the unit ball in $\mathcal{H}$. Recall from Assumption 5 that

$$H(\delta, \|\cdot\|_{L_\infty}, \mathcal{B}) \leq C\delta^{-2\beta}|\log\delta|^{2\kappa},$$

for some positive constant $C$, $\kappa$, and $\beta \in [0, 1)$. Hence,

$$\int_0^R (H(u, \|\cdot\|_{L_\infty}, \mathcal{B}))^{\frac{1}{2}}du \leq C\int_0^R u^{-\beta}|\log u|^\kappa du \leq CR^{1-\beta}|\log R|^\kappa.$$

One may readily check that for all $R < 1$ and $a \geq n^{-1/2}CR^{1-\beta}\big|\log R\big|^{\kappa}$, the condition equation 76 is met, and thus by equation 77, we have

$$\mathbb{P}\left(\sup_{f \in \mathcal{H}, \|f\|_n \leq R} n^{\frac{1}{2}}|\langle \boldsymbol{\omega}, f\rangle_n| \geq R^{1-\beta}|\log R|^{\kappa}\right) = \mathbb{P}\left(\sup_{f \in \mathcal{H}, \|f\|_n \leq R}|\langle \boldsymbol{\omega}, f\rangle_n| \geq a\right)$$
$$\leq 2\exp(-C_1\frac{na^2}{R^2}), \tag{78}$$

for some positive constant $C_1$.

Let $h = f/\|f\|_{\mathcal{H}}$. Then, the left-hand-side of the inequality equation 64 becomes

$$\sup_{h \in \mathcal{H}} \frac{|\langle \boldsymbol{\varepsilon}, h\rangle_n|}{\|h\|_n^{1-\beta}\big|\log \|h\|_n\big|^{\kappa}}.$$

We can then give the following upper bound for its tail distribution:

$$\mathbb{P}\left(\sup_{h \in \mathcal{H}} \frac{|\langle \boldsymbol{\varepsilon}, h\rangle_n|}{\|h\|_n^{1-\beta}\big|\log \|h\|_n\big|^{\kappa}} > t\right)$$
$$= \mathbb{P}\left(\bigcup_{i=0}^{\infty}\left\{\sup_{h \in \mathcal{H}} \frac{|\langle \boldsymbol{\varepsilon}, h\rangle_n|}{\|h\|_n^{1-\beta}\big|\log \|h\|_n\big|^{\kappa}} > t\right\}\right)$$
$$\leq \mathbb{P}\left(\bigcup_{i=0}^{\infty}\left\{\sup_{h \in \mathcal{H}, \|h\|_n \in (2^{-i-1}, 2^{-i}]} n^{\frac{1}{2}}|\langle \boldsymbol{\varepsilon}, h\rangle_n| > t2^{-(i+1)(1-\beta)}(i+1)^{\kappa}\right\}\right)$$
$$\leq \sum_{i=0}^{\infty} \mathbb{P}\left(\sup_{h \in \mathcal{H}, \|h\|_n \leq 2^{-i}} n^{\frac{1}{2}}|\langle \boldsymbol{\varepsilon}, f\rangle_n| > t2^{-(i+1)(1-\beta)}i^{\kappa}\right)$$
$$\leq \sum_{i=0}^{\infty} 2\exp(-C_2 t^2 2^{\beta i} i^{\kappa})$$
$$\leq \sum_{i=1}^{\infty} 2\big(\exp(-C_2 t^2)\big)^i$$
$$\leq C_3 \exp(-C_2 t^2),$$

where the fifth line follows from applying equation 78 with $R = 2^{-i}$ and $a = tn^{-1/2}2^{-(i+1)(1-\beta)}i^{\kappa}$. $\square$

## F    PROOF OF COROLLARY 11

*Proof.* The proof relies on the following so-called sampling inequality of RKHSs

$$\|f\|_{L_2(\sigma_0^\pi)}^2 \leq \|f\|_n^2 + \delta_{\mathcal{H},n}\|f\|_{\mathcal{H}}^2 \tag{79}$$

where $\delta_{\mathcal{H},n}$ depends on the distribution of data $\omega_0^{(i)}$ and the structure of RKHS $\mathcal{H}$.

**Case I:** $\mathcal{H}_{\mathcal{S}} = \{s\}_{s=1}^S$ **and** $\mathcal{A} = \{a\}_{a=1}^A$

In this case, we can note the RKHS is a discrete set $\{\omega_j\}_{j=1}^M$ where $M = SA$, $\sigma_0^\pi(\omega_j) = \frac{W_j}{M}$ is the weight on element $\omega_j$ (according to our assumption, $c < W_j < C$), and data $\{\omega_0^{(i)}\}$ are i.i.d. samples from $\{\omega_j\}_{j=1}^M$ following $\sigma_0^\pi$.

Because the kernel $K(\omega, \omega') = \delta_{s=s'}\delta_{a=a'} = \delta_{\omega=\omega'}$, we can derive that the RKHS is equipped with the inner product

$$\langle f, g\rangle_{\mathcal{H}} = \sum_{j=1}^M f_j g_j, \quad f, g \in \mathcal{H}$$

where $f_j$ and $g_j$ are the values of $f$ and $g$ on $\omega_j$, respectively. Also, the $L_2$ norm and empirical norm are

$$\|f\|^2_{L_2(\sigma_0^\pi)} = \sum_{j=1}^M f_j^2 W_j, \quad \|f\|_n^2 = \frac{1}{n}\sum_{i=1}^n f^2(\omega_i^0).$$

Let $I_{i,j}$ denote the indicator function that $\omega_0^{(i)} = \omega_j$, then we can rewrite the empirical norm as follows:

$$
\begin{aligned}
\|f\|_n^2 =& \frac{1}{n}\sum_{i=1}^n f^2(\omega_i^0) = \sum_{j=1}^M f_j^2 \frac{\sum_{i=1}^n I_{i,j}}{n} \\
=& \sum_{j=1}^M f_j^2 \frac{W_j}{M} + \sum_{j=1}^M f_j^2 \left( \frac{\sum_{i=1}^n I_{i,j}}{n} - \frac{W_j}{M} \right) \\
=& \|f\|^2_{L_2(\sigma_0^\pi)} + \sum_{j=1}^M f_j^2 \left( \frac{Z_j}{n} - \frac{W_j}{M} \right)
\end{aligned}
\tag{80}
$$

where $[Z_j]_{j=1}^M$ is Multinomial distributed with $n$ trials, $M$ mutually exclusive events, and probability vector $[W_j/m]_{j=1}^M$.

By properties of the Multinomial distribution, the last term in equation 80 has expectation

$$\mathbb{E}\sum_{j=1}^M f_j^2 \left( \frac{Z_j}{n} - \frac{W_j}{M} \right) = \sum_{j=1}^M f_j^2 \left( \frac{nW_j/m}{n} - \frac{W_j}{M} \right) = 0, \tag{81}$$

and variance

$$
\begin{aligned}
\mathrm{Var}\left( \sum_{j=1}^M f_j^2 \left( \frac{Z_j}{n} - \frac{W_j}{M} \right) \right) =& \frac{1}{n^2}\sum_{j,l=1}^m f_j^2 f_l^2 \mathrm{Cov}\left( Z_j, Z_l \right) \\
=& \frac{1}{n^2}\left( \sum_{j=1}^m f_j^2 \frac{nW_j}{M}(1 - \frac{W_j}{M}) - \sum_{j\neq l} f_j^2 f_l^2 \frac{nW_j W_l}{M^2} \right) \\
\leq& \mathcal{O}(\frac{1}{n^{1+\nu}}\|f\|_{\mathcal{H}}^2)
\end{aligned}
\tag{82}
$$

where the last line is from our assumption that $M \geq n^\nu$ for some $\nu \in (0,1)$.

Substitute equation 81 and equation 82 into equation 80, it is straightforward to derive the sampling inequality

$$\|f\|_n^2 \leq \|f\|^2_{L_2(\sigma_0^\pi)} + \mathcal{O}_p\left( \frac{1}{n^{(1+\nu)/2}} \right) \|f\|_{\mathcal{H}}^2. \tag{83}$$

It is straightforward to see that the covering number of $\mathcal{H}$ under $L_\infty$ is directly the covering number of $\mathcal{H}$ is $M/\delta$ so its entropy is

$$H(\delta, \|\cdot\|_{L_\infty}, \mathcal{H}) \leq \nu\log n + |\log\delta|.$$

Substituting this result into the sampling inequality, we can have the final result:

$$
\begin{aligned}
\|f_t - Q^\pi\|_{L_2(\sigma_0^\pi)} \leq& \|f_t - Q^\pi\|_n + \mathcal{O}_p\left( \frac{1}{n^{(1+\nu)/4}} \right) \|f_t - Q^\pi\|_{\mathcal{H}} \\
\leq& \mathcal{O}_p\left( (1-c\gamma)^{-1} n^{-\frac{1}{2}} |\log n|^{1/2} \right) + \mathcal{O}_p\left( \frac{1}{n^{(1+\nu)/4}} \right) \|f_t - Q^\pi\|_{\mathcal{H}} \\
\leq& \mathcal{O}_p\left( (1-c\gamma)^{-1} n^{-\frac{1}{2}} |\log n|^{1/2} \right) + \mathcal{O}_p\left( \frac{1}{n^{(1+\nu)/4}} \right) \|Q^\pi\|_{\mathcal{H}}
\end{aligned}
$$

where the second line is from the empirical error in Theorem 10 with $\beta = 0$ and $\kappa = 1/2$, and the last line is from the RKHS norm of $f_t$ in Theorem 10.

**Case II: $\mathcal{H}_{\mathcal{S}}$ is a Sobolev space and $\mathcal{A} = \{a\}_{a=1}^{A}$**

For spaces of continuous functions, we first need to define a concept called the filled distance of data set so that we can apply the sampling inequality.

**Definition 20.** *Given $n$ points $\mathbf{X} = \{\mathbf{x}_i\}_{i=1}^{n}$ on the set $\mathcal{X}$ equipped with norm $\|\cdot\|$, the fill distance of the set $\mathbf{X}$ is*
$$q_{\mathbf{X}} = \max_{\mathbf{x} \in \mathcal{X}} \min_{\mathbf{x}_i \in \mathbf{X}} \|\mathbf{x}_i - \mathbf{x}\|_2.$$

Using the concept of fill distance, we have the following Lemma from Tuo et al. (2020); Rieger (2008); Utreras (1988)

**Proposition 21** (Proposition 2.6 in Tuo et al. (2020)). *Let $\mathcal{H}$ be a Sobolev space for function defined on a compact and convex set $\mathcal{X}$ with smoothness parameter $m$. Let $\mathbf{X} = \{\mathbf{x}_i\}_{i=1}^{n}$ be a set of scattered point on $\mathcal{X}$ with fill distance $q_{\mathbf{X}}$ and $\|\cdot\|_n$ is the empirical norm induced by $\mathbf{X}$. Then there exists a constant $C$ (depending only on $m$ and $\mathcal{X}$) such that for any $f \in \mathcal{H}$:*
$$\|f\|_{L_2(\mathcal{X})}^2 \leq C \left( \|f\|_n^2 + q_{\mathbf{X}}^{2m} \|f\|_{\mathcal{H}}^2 \right).$$

We also need the following proposition for the distribution of fill distance:

**Proposition 22** (Proposition 3 in Krieg & Sonnleitner (2024)). *Let $\{\mathbf{x}_i \sim \sigma\}_{i=1}^{n}$ be i.i.d. distributed on a compact and convex set $\mathcal{X}$. Let $\Phi(\varepsilon) = \varepsilon^d$. If there exists a positive number $\varepsilon_0$ such that $\sigma(B_\varepsilon(\mathbf{x})) \geq \Phi(\varepsilon)$ for all $\varepsilon \leq \varepsilon_0$ and $\mathbf{x} \in B$ where $B_\varepsilon(\mathbf{x})$ is a sphere centered at $\mathbf{x}$ with radius $\varepsilon$. Then there exist positive constants $c_1$, $c_2$, $c_3$, and $b_0$ such that for any $b > b_0$ we have*
$$\Pr\left( \max_{\mathbf{x} \in \mathcal{S}} \inf_{i} \|\mathbf{x} - \mathbf{x}_i\| \geq c_1 \Phi^{-1}(\frac{b \log n}{n}) \right) \leq c_2 n^{1 - c_3 b}.$$

From Proposition 22, we can derive that the filled distance on $\mathcal{X}$ with intrinsic dimension $d$ is
$$q_{\mathbf{X}} = \mathcal{O}_p(n^{-1/d} \log n). \tag{84}$$

Lastly, from Geer (2000); Birman & Solomjak (1967), we have the entropy of Sobolev spaces $\mathcal{H}$ with smoothness $m$ defined on compact $d$-dimensional domain is
$$H(\delta, \|\cdot\|_{L_\infty}, \mathcal{H}) \leq C_1 \delta^{-\frac{d}{m}} \tag{85}$$
where $C_1$ is some constant independent of $\delta$.

Now we are ready to prove the sampling inequality in the form equation 79 for our RKHS. According to our assumption, $\mathcal{H}_{\mathcal{S}}$ is a Sobolev space with smoothness parameter $m$ embedded on a domain with intrinsic dimension $d$ and $\mathcal{H}_{\mathcal{A}}$ is a discrete set. So it is straightforward to derive that the overall RKHS $\mathcal{H}$ is a vector-valued function $(f_1, \cdots, f_A)$ where each $f_a \in \mathcal{H}_{\mathcal{S}}$, and $\mathcal{H}$ is define as
$$\mathcal{H} = \left\{ f = (f_1, \cdots, f_A) : \|f\|_{\mathcal{H}}^2 = \sum_{a=1}^{A} \|f_a\|_{\mathcal{H}_{\mathcal{S}}}^2 < \infty \right\}.$$

Because $f_t, Q^\pi \in \mathcal{H}$, we then have

$$
\begin{aligned}
\|f_t - Q^\pi\|_{L_2(\sigma_0^\pi)}^2 &= \sum_{a=1}^{A} \|f_{a,t} - Q_a^\pi\|_{L_2(\sigma_0^\pi)}^2 \\
&\leq \sum_{a=1}^{A} C \left( \|f_{a,t} - Q_a^\pi\|_n^2 + q_{\boldsymbol{\omega}_0}^{2m} \|f_{a,t} - Q_a^\pi\|_{\mathcal{H}_{\mathcal{S}}}^2 \right) \\
&\leq \sum_{a=1}^{A} C \left( \|f_{a,t} - Q_a^\pi\|_n^2 + \mathcal{O}_p(n^{-\frac{2m}{d}} |\log n|^{2m}) \|f_{a,t} - Q_a^\pi\|_{\mathcal{H}_{\mathcal{S}}}^2 \right) \\
&\leq \mathcal{O}_p \left( (1 - c\gamma)^{-\frac{4m+d}{2m+2d}} n^{-\frac{2m}{2m+d}} \right) + \mathcal{O}_p(n^{-\frac{2m}{d}} |\log n|^{2m}) \|Q^\pi\|_{\mathcal{H}}^2 \\
&\leq \mathcal{O}_p \left( (1 - c\gamma)^{-\frac{4m+d}{2m+2d}} n^{-\frac{2m}{2m+d}} \right)
\end{aligned}
\tag{86}
$$

where the second line is from Proposition 21, the third line is from the distribution of fill distance equation 84, the fourth line is from Theorem 8 with $\beta = \frac{d}{2m}$ and $\kappa = 0$.

**Case III: $\mathcal{H}_S$ is a NTK and $\mathcal{A} = \{a\}_{a=1}^{A}$**

According to Chen & Xu (2020), the NTK $N$ associated to a two-layer neural network on $\mathbb{S}^{d-1}$ is equivalent to the Laplace kernel $e^{-\|s-s'\|}$ on $\mathbb{S}^{d-1}$. The Laplace kernel is a Matérn kernel with smoothness parameter $\nu = 1/2$. Also, according to Corollary 1 in Tuo & Jeff Wu (2016), RKHS induced by Matérn kernel with smoothness parameter $\nu$ is equivalent to (fractional) Sobolev space with smoothness $m = \nu + d/2$.

Therefore, we can conclude that $\mathcal{H}$ is equivalent to a Sobolev space with smoothness $(d+1)/2$ embedded on $\mathbb{S}^{d-1}$, which has intrinsic dimension $d-1$. So Case III is a special case of Case II, i.e. Sobolev space with smoothness $m = (d+1)/2$ embedded in the $d-1$ dimension. By substituting the value of $m$ and $d-1$ into equation 16, we can have the final result.

**Case IV: $\mathcal{H}_S$ and $\mathcal{H}_\mathcal{A}$ are Gaussian RKHSs**

Note that $K(\omega, \omega') = e^{-\|\omega - \omega'\|^2}$ is a Gaussian kernel on the hypercube $[0,1]^d$ with $d = d_s + d_a$. So $\mathcal{H}$ is a Gaussian RKHS on $[0,1]^d$.

From Chapter 4 in Rieger (2008), we have the sampling inequality:

$$\|f\|_{L_2(\sigma_0^\pi)} \leq e^{c \log(cq\omega_0)/\sqrt{q\omega_0}} \|f\|_\mathcal{H} + c\|f\|_n, \quad \forall f \in \mathcal{H} \tag{87}$$

for some generic constant $c > 0$. Substitute the distribution of fill distance in equation 84 into equation 87, we can have

$$\|f\|_{L_2(\sigma_0^\pi)} \leq c\|f\|_n + \mathcal{O}_p(e^{-n^{1/\sqrt{d}}})\|f\|_\mathcal{H}, \quad \forall f \in \mathcal{H}. \tag{88}$$

From Theorem 3 in Kühn (2011), the entropy of $\mathcal{H}$ is

$$H(\delta, \|\cdot\|_\infty, \mathcal{B}) \leq C_1 |\log \delta|^{d+1} \tag{89}$$

where $C_1$ is some generic constant.

So we can have

$$\begin{aligned}
\|f_t - Q^\pi\|_{L_2(\sigma^\pi 0)} &\leq c\|f_t - Q^\pi\|_n + \mathcal{O}_p(e^{-n^{1/\sqrt{d}}})\|f_t - Q^\pi\|_\mathcal{H} \\
&\leq \mathcal{O}_p((1 - c\gamma)n^{-\frac{1}{2}}|\log n|^{(d+1)/2}) + \mathcal{O}_p(e^{-n^{1/\sqrt{d}}})\|Q^\pi\|_\mathcal{H} \\
&\leq \mathcal{O}_p((1 - c\gamma)n^{-\frac{1}{2}}|\log n|^{(d+1)/2})
\end{aligned}$$

where the first line is from the sampling inequality equation 88 and the second line is from entropy equation 89 and Theorem 10 with $\beta = 0$ and $\kappa = (d+1)/2$.

$\square$

# G  PROOF OF LEMMA 12

*Proof.* We generalize the proof for Proposition 3.1 in Liu et al. (2019), following its main lines of reasoning while incorporating additional techniques of RKHS. For notation simplicity, denote the $L_2$ inner product of on $\mathcal{A}$ as $\langle f, g \rangle_\mathcal{A} = \int_\mathcal{A} f(a)g(a)da$.

The Lagrangian of equation 19 takes the form

$$\mathbb{E}_n \left[ \Delta_k \langle f^{(k)}(s, \cdot) \pi(\cdot|s) \rangle_\mathcal{A} - \mathrm{KL}\left(\pi(\cdot|s) \| \pi^k(\cdot|s)\right) \right] + \mathbb{E}_n \left[ \left( \int_\mathcal{A} \pi(a|s) - 1 \right) \Lambda(s) \right] \tag{90}$$

where $\Lambda(s)$ is the $L_2$ Lagrange multiplier defined on $\{s_0^{(i)}\}$. By taking the functional derivative of equation 90 with respective to $\pi$, we can have

$$\Delta_k f^{(k)}(s, a) + F(s, a) - (\log \pi(a|s) + 1 + \log Z(s)) + \Lambda(s) = 0 \tag{91}$$

where $Z(s) = \int_{\mathcal{A}} e^{F(s,a)} da$. Then equation 91 should holds on any $s \in \{s_0^{(i)}\}$ and $a \in \mathcal{A}$ such that $|F(s,a)|$ and $\log Z(s)$ is bounded. According to Assumption 1, any $(s,a) \in \{s_0^{(i)}\} \times \mathcal{A}$ satisfies this condition. Then we can solve equation 91, which yields

$$\pi(a|s) = \pi^k(a|s) \exp\left\{\Delta_k f^{(k)}(s,a) + \Lambda(s) - 1 - \log Z(s)\right\}. \tag{92}$$

In equation 92, we can note that $\pi^k(a|s)$ must be proportional to $\pi^k(a|s) \exp\{\Delta_k f^{(k)}(s,a)\}$ because it is a density on $a$. $\qquad\square$

## H    PROOF OF THEOREM 13

*Proof.* From direct calculations, we have for any $s \in \mathcal{S}$

$$
\begin{aligned}
&\mathrm{KL}\left(\pi^*(\cdot|s)||\pi^{k+1}(\cdot|s)\right) - \mathrm{KL}\left(\pi^*(\cdot|s)||\pi^k(\cdot|s)\right)\\
=& \langle \log \frac{\pi^k}{\pi^{k+1}}, \pi^* - \pi^{k+1} \rangle_{\mathcal{A}} - \langle \log \frac{\pi^{k+1}}{\pi^k}, \pi^{k+1} \rangle_{\mathcal{A}}\\
=& - \Delta_k \langle f^{(k)}, \pi^* - \pi^k \rangle_{\mathcal{A}} + \Delta_k \langle f^{(k)}, \pi^{k+1} - \pi^k \rangle_{\mathcal{A}} - \langle \log \frac{\pi^{k+1}}{\pi^k}, \pi^{k+1} \rangle_{\mathcal{A}} \qquad (93)\\
=& - \Delta_k \langle Q^{(k)}, \pi^* - \pi^k \rangle_{\mathcal{A}} - \Delta_k \langle f^{(k)} - Q^{(k)}, \pi^* - \pi^{k+1} \rangle_{\mathcal{A}}\\
& - \Delta_k \langle Q^{(k)}, \pi^k - \pi^{k+1} \rangle_{\mathcal{A}} - \langle \log \frac{\pi^{k+1}}{\pi^k}, \pi^{k+1} \rangle_{\mathcal{A}}\\
\leq& - \Delta_k \langle Q^{(k)}, \pi^* - \pi^k \rangle_{\mathcal{A}} - \Delta_k \langle f^{(k)} - Q^{(k)}, \pi^* - \pi^{k+1} \rangle_{\mathcal{A}} \qquad (94)\\
& - \int_{\mathcal{A}} \Delta_k Q^{(k)}(s,a) \left(\pi^k - \pi^{k+1}\right) da - \frac{1}{2}\left(\int_{\mathcal{A}} \left|\pi^{k+1}(a|s) - \pi^k(a|s)\right| da\right)^2\\
\leq& - \Delta_k \langle Q^{(k)}, \pi^* - \pi^k \rangle_{\mathcal{A}} - \Delta_k \langle f^{(k)} - Q^{(k)}, \pi^* - \pi^{k+1} \rangle_{\mathcal{A}}\\
& + \|\Delta_k Q^{(k)}(s,)\|_{L_\infty(\mathcal{A})} \int_{\mathcal{A}} \left|\pi^{k+1} - \pi^k\right| da - \frac{1}{2}\left(\int_{\mathcal{A}} \left|\pi^{k+1} - \pi^k\right| da\right)^2 \qquad (95)\\
\leq& - \Delta_k \langle Q^{(k)}, \pi^* - \pi^k \rangle_{\mathcal{A}} - \Delta_k \langle f^{(k)} - Q^{(k)}, \pi^* - \pi^{k+1} \rangle_{\mathcal{A}} + \Delta_k^2 \frac{\|r\|_{L_\infty}}{1 - \gamma}
\end{aligned}
$$

where equation 93 is from the NPG update rule equation 20, equation 94 is from Pinsker inequality, equation 95 is from the fact that $xy - y^2 \leq x^2$ and $Q^{(k)} \leq \max_{s,a} |r(s,a)|/(1-\gamma)$.

Taking expectation $\mathbb{E}_{S \sim \nu^*}$ on equation 95, we can have:

$$
\begin{aligned}
&\mathbb{E}_{S \sim \nu^*}\left[\mathrm{KL}\left(\pi^*(\cdot|s)||\pi^{k+1}(\cdot|s)\right) - \mathrm{KL}\left(\pi^*(\cdot|S)||\pi^{k+1}(\cdot|S)\right)\right]\\
\leq& \mathbb{E}_{S \sim \nu^*} - \Delta_k \langle Q^{(k)}(\cdot, S), \pi^*(\cdot|S) - \pi^k(\cdot|S) \rangle_{\mathcal{A}} \qquad (96)\\
& - \mathbb{E}_{S \sim \nu^*} \Delta_k \langle f^{(k)}(\cdot, S) - Q^{(k)}(\cdot, S), \pi^*(\cdot|S) - \pi^{k+1}(\cdot|S) \rangle_{\mathcal{A}} \qquad (97)\\
& + \Delta_k^2 \frac{\|r\|_{L_\infty}}{1 - \gamma}
\end{aligned}
$$

For equation 96, we can use the performance difference lemma in Kakade & Langford (2002) (equivalent to Liu et al. (2019) Lemma 5.1):

**Lemma 23** (Performance Difference Lemma). *For $\mathcal{R}[\pi]$ defined in equation 21, we have*

$$\mathcal{R}[\pi] - \mathcal{R}[\pi^*] = \mathbb{E}_{S \sim \nu^*}[\langle Q^\pi(S, \cdot), \pi(\cdot|S) - \pi^*(\cdot|S) \rangle_{\mathcal{A}}]. \tag{98}$$

From equation 98, we can directly have for equation 96

$$\mathbb{E}_{S \sim \nu^*} \Delta_k \langle Q^{(k)}(\cdot, S), \pi^*(\cdot|S) - \pi^k(\cdot|S) \rangle_{\mathcal{A}} = \Delta_k \left(\mathcal{R}[\pi^*] - \mathcal{R}[\pi^k]\right). \tag{99}$$

For equation 97,

$$
\begin{aligned}
&\left|\mathbb{E}_{S \sim \nu^*} \Delta_k \langle f^{(k)}(\cdot, S) - Q^{(k)}(\cdot, S), \pi^*(\cdot|S) - \pi^{k+1}(\cdot|S) \rangle_{\mathcal{A}}\right|\\
\leq& \Delta_k \|f^{(k)} - Q^k\|_{L_\infty} \left(\int \pi^*(a|s)\nu^*(s) da ds + \int \pi^{k+1}(a|s)\nu^*(s) da ds\right)\\
=& 2\Delta_k \|f^{(k)} - Q^k\|_{L_\infty}
\end{aligned}
$$

Substitute the above upper bounds for equation 96 and equation 97 into equation 95 and rearrange the terms, we have

$$\Delta_k \left( \mathcal{R}[\pi^*] - \mathcal{R}[\pi^k] \right) + \mathbb{E}_{S \sim \nu^*} \left[ \text{KL} \left( \pi^*(\cdot|S) || \pi^{k+1}(\cdot|S) \right) - \text{KL} \left( \pi^*(\cdot|S) || \pi^k(\cdot|S) \right) \right]$$
$$\leq 2\Delta_k \| f^{(k)} - Q^{(k)} \|_{L_\infty} + \Delta_k^2 (1 - \gamma)^{-1} \| r \|_{L_\infty}. \tag{100}$$

By summing over all the $k$ in equation 100, we can have the final result. $\qquad\square$

## I  PROOF OF COROLLARY 14

*Proof.* The proof mainly relies on the following $L_2$ and $L_\infty$ norm embedding inequality

$$\| f \|_{L_\infty} \leq C \| f \|_{L_2(\mu)}^{\varphi} \| f \|_{\mathcal{H}}^{1-\varphi} \tag{101}$$

for some constants $C$ and $0 \leq \varphi \leq 1$ independent of $f$ and $\mu$ is any measure equivalent to the uniform measure on $\mathcal{S} \times \mathcal{A}$. Based on Assumption 1 that $\nu^*$ is lower bounded on $\mathcal{S} \times \mathcal{A}$, we can use equation 101 and Theorem 10 to determine the parameters in Algorithm 1 such that the $L_\infty$ error satisfies

$$\| f^{(k)} - Q^{(k)} \|_{L_\infty} \leq \frac{1}{\sqrt{k}} \tag{102}$$

Then by substitute equation 102 and $\Delta_k = 1/\sqrt{k}$ into equation 22, we can have the target result. We prove the cases one by one

**Tabular:** On the discrete set, it is obvious that

$$\| f \|_{L_\infty} = \max_{\omega \in \mathcal{S} \times \mathcal{A}} |f(\omega)| \leq \sqrt{\sum_{\omega \in \mathcal{S} \times \mathcal{A}} f(\omega)^2} \leq \frac{\| f \|_{L_2(\nu^*)}}{\min_\omega \nu^*}. \tag{103}$$

By letting $n^{(k)} = \mathcal{O}(\frac{k \| \pi^k \|_{\mathcal{H}}^2}{(1-c\gamma)^2} \log \frac{k \| \pi^k \|_{\mathcal{H}}}{(1-c\gamma)})$ and $\beta = 0, \kappa = 1/2$ for the tabular case, we can derive from Theorem 17 that the required $\lambda^{(k)}$ for the optimal error rate is

$$\lambda^{(k)} = \mathcal{O}\left( n^{-\frac{1}{2}} |\log n|^1 \right) = \mathcal{O}\left( \frac{(1-c\gamma)}{\sqrt{k} \| \pi^k \|_{\mathcal{H}}} |\log \frac{k \| \pi^k \|_{\mathcal{H}}}{(1-c\gamma)}|^{\frac{1}{2}} \right).$$

We then substitute equation 103 into equation 15:

$$\| f^{(k)} - Q^{(k)} \|_{L_\infty} \leq \mathcal{O}_p \left( (1-c\gamma)^{-1} n^{-\frac{1}{2}} |\log n|^{1/2} \right) \| Q^{(k)} \|_{\mathcal{H}} + \mathcal{O}_p \left( \frac{1}{n^{(1+\nu)/4}} \right) \| Q^{(k)} \|_{\mathcal{H}}$$
$$\leq \mathcal{O}_p \left( \frac{1}{\sqrt{k}} \right) + \mathcal{O}_p \left( \frac{1}{n^{(1+\nu)/4}} \right) \| Q^{(k)} \|_{\mathcal{H}} \tag{104}$$

where the second line is from Lemma 3.

From equation 104, we can note that we also need to make sure that the second term on the right-hand side is in the order $k^{-1/2}$. So we may need more samples so that $\mathcal{O}_p \left( \frac{1}{n^{(1+\nu)/4}} \right) \| Q^{(k)} \|_{\mathcal{H}} \leq \mathcal{O}(k^{-1/2})$. By solving this equation, we can have the target result.

**Sobolev with intrinsic dimension $d$:** According to the Gagliardo-Nirenberg inequality, when $\mathcal{H}$ is a Sobolev space with smoothness parameter $m$ and dimension $d$, we have

$$\| f \|_{L_\infty} \leq C \| f \|_{L_2(\nu^*)}^{\frac{2m-d}{2m}} \| f \|_{\mathcal{H}}^{\frac{d}{2m}}. \tag{105}$$

By letting $n^{(k)} = \mathcal{O}\left( \frac{\| \pi^k \|_{\mathcal{H}}^{\frac{2(2m+d)}{(2m-d)}} k^{\frac{2m+d}{2m-d}}}{(1-c\gamma)^{\frac{2m+d/2}{m}}} \right)$ and $\beta = \frac{d}{2m}, \kappa = 0$ for the Sobolev case, we can derive from Theorem 17 that the required $\lambda^{(k)}$ for the optimal error rate is

$$\lambda = \mathcal{O}((1-c\gamma)^{\frac{d/2}{2m+d}} n^{-\frac{m}{2m+d}}|) = \mathcal{O}\left( \frac{(1-c\gamma)}{\| \pi \|_{\mathcal{H}}^{\frac{2m}{2m-d}} k^{\frac{m}{2m-d}}} \right)$$

Similar to the tabular case, we then substitute equation 105 into equation 16:

$$\|f^{(k)} - Q^{(k)}\|_{L_\infty} \leq \mathcal{O}_p\left(\left((1-c\gamma)^{-\frac{2m+d/2}{2m+d}}n^{-\frac{m}{2m+d}}\right)^{\frac{2m-d}{2m}}\|\pi^{(k)}\|_{\mathcal{H}}\right) \leq \mathcal{O}_p\left(\frac{1}{\sqrt{k}}\right).$$

**NTK:** As we have shown in Section F Case III that the NTK $N$ associated to a two-layer neural network on $\mathbb{S}^{d-1}$ is equivalent to the (fractional) Sobolev space with smoothness $m = 1/2 + d/2$ with intrinsic dimension $d-1$. Substitute the smoothness $(1+d)/2$ and intrinsic dimension $d-1$ into the Sobolev case, we can have the result.

**Gaussian:** we use Lemma E.4 in Ding et al. (2024) for the interpolation inequality in Gaussian RKHS. It states that for any $f \in \mathcal{H}$, there exists a universal constant $C > 0$ such that

$$\|f\|_{L_\infty} \leq C\|f\|_{L_2(\nu^*)}^{1-\epsilon}\|f\|_{\mathcal{H}}^{\epsilon} \tag{106}$$

for any $\epsilon > 0$.

By letting $n^{(k)} = \mathcal{O}\left(\frac{\|\pi^k\|_{\mathcal{H}}^{\frac{2}{1-\epsilon}}k^{\frac{1}{1-\epsilon}}}{(1-c\gamma)^2}\log\frac{\|\pi^k\|_{\mathcal{H}}k}{1-c\gamma}\right)$ and $\beta = 0$, $\kappa = (d+1)/2$ for the Gaussian case, we can derive from Theorem 17 that the required $\lambda^{(k)}$ for the optimal error rate is

$$\lambda = \mathcal{O}(n^{-\frac{1}{2}}|\log|^{\frac{d+1}{2}}) = \mathcal{O}\left(\frac{(1-c\gamma)}{\|\pi\|_{\mathcal{H}}^{\frac{1}{1-\epsilon}}\sqrt{k}^{\frac{1}{1-\epsilon}}}\right), \quad \forall\epsilon \in (0,1).$$

Similar to the tabular case, we then substitute equation 106 into equation 18:

$$\|f^{(k)} - Q^{(k)}\|_{L_\infty} \leq \mathcal{O}_p\left(\left((1-c\gamma)^{-1}n^{-\frac{1}{2}}|\log n|^{\frac{d+1}{2}}\right)^{\frac{1}{1-\epsilon}}\|\pi^{(k)}\|_{\mathcal{H}}\right) \leq \mathcal{O}_p\left(\frac{1}{\sqrt{k}}\right).$$

$\square$

## J  N-STEP TD LEARNING

In the main paper, we introduced the learning of the Q-function using samples generated from one-step transitions evolved from an initial distribution $\mu$. The one-step TD method can be naturally extended to $N$-step TD by viewing the $N$-step dynamics of the original MDP as the one-step transition dynamics of an enlarged MDP. The additional sampling steps reduce sampling variance and lead to faster convergence.

In every update of policy $\pi$, we first sample i.i.d. $\{s_0^{(i)} \sim \mu_0\}_{i=1}^n$ from some chosen distribution $\mu_0$ and, for each $i$, generate subsequent states and actions $a_0^{(i)}$ and $\{(s_t^{(i)}, a_t^{(i)})\}_{t=1}^N$ following the Markov chain as follows:

$$s_0^{(i)} \sim \mu_0, \quad a_0^{(i)} \sim \pi(\cdot|s_0^{(i)}), \quad s_{t+1}^{(i)} \sim P(\cdot|s_t^{(i)}, a_t^{(i)}), \quad a_t^{(i)} \sim \pi(\cdot|s_t^{(i)}). \tag{107}$$

In the $N$-step TD learning, Assumption 6 can be relaxed to the following assumption:

**Assumption 6B.** *The marginal density of the $N$-step state $s_N$ is given as*

$$\mathbb{P}(s_N = s)$$
$$= \int P(s|s_{N-1}, a_{N-1})\left(\prod_{\tau=1}^{N-1}\pi(a_\tau|s_\tau)P(s_\tau|s_{\tau-1}, a_{\tau-1})\right)\pi(a_0|s_0)\mu_0(s_0)(\prod_{\tau=0}^{N-1}ds_\tau da_\tau).$$

*There exists a constant $c$ such that $\mathbb{P}(s_N = s) \leq c\mu_0(s)$ and $c\gamma^N < 1$.*

The constraint $c\gamma < 1$ in Assumption 6 is replaced by $c\gamma^N < 1$ in Assumption 6B. In $N$-step TD learning, the constant $c\gamma$ appearing in the convergence rate in Theorem 10, Corollary 11, and Corollary 14 can accordingly be replaced by the smaller constant $c\gamma^N$, leading to a faster convergence. The intuition behind this improvement is that Assumption 6B is exactly equivalent to Assumption 6 on an MDP with the smaller discount factor $\gamma^N$, a modified transition kernel $P(s_{t+N} \mid s_t, a_t)$, and a modified reward $\bar{r}(s_0, a_0) = \sum_{t=0}^{N-1}\gamma^t r(s_t, a_t)$.

Given dataset $(\omega_0^{(i)}, \omega_1^{(i)}, \cdots, , \omega_N^{(i)})_{i=1}^n$ with $\omega_t^{(i)} = (s_t^{(i)}, a_t^{(i)})$, $t = 0, 1, \cdots, N$ for any given policy $\pi$, we aim to learn the Q-function $Q^\pi$ associated with the $N$-step Bellman equation in the following form:

$$Q^\pi(s, a) = \mathbb{E}[\sum_{t=0}^{N-1} \gamma^t r(s_t, a_t)|s_0 = s, a_0 = a] + \gamma^N \mathbb{E}[Q^\pi(s_t, a_t)|s_0 = s, a_0 = a]. \tag{108}$$

Note that equation 108 can be also treated as the Bellman equation associated to an MDP with discount factor $\gamma^N$, one-step expected reward function $\mathbb{E}[\sum_{t=0}^{N-1} \gamma^t r(s_t, a_t)|s_0 = s, a_0 = a]$, and transition kernel $P(s_N|s_0, a_0)$. So it is natural to derive that equation 108 can be approximated by solving the following fixed-point KRR over the whole RKHS $\mathcal{H}$:

$$\hat{Q}^\pi = \min_{f \in \mathcal{H}} \frac{1}{n} \sum_{i=1}^n \left( f(\omega_0^{(i)}) - \left( \sum_{t=0}^{N-1} \gamma^t r(\omega_t^{(i)}) \right) - \gamma^N \hat{Q}^\pi(\omega_N^{(i)}) \right)^2 + \lambda \|f\|_{\mathcal{H}}^2. \tag{109}$$

The $N$-step fixed point KRR equation 109 is almost the same as the one-step case equation 6 except that we replace the terms $r(\omega_0^{(i)})$ and $\gamma \hat{Q}^\pi(\omega_1^{(i)})$ in equation 6 by $\left( \sum_{t=0}^{N-1} \gamma^t r(\omega_t^{(i)}) \right)$ and $\gamma^N \hat{Q}^\pi(\omega_N^{(i)})$ in equation 109, respectively. In the following subsections, we first introduce the connection between the equation 108 and equation (109) and derive the solution of 109. Next, we introduce the generalization from the one-step to the $N$-step TD update. Finally, we analyze the convergence pattern of $N$-step TD.

### J.1 EMPIRICAL N-STEP BELLMAN EQUATION

We first generalize the definition of cross-covariance operator on the RKHS $\mathcal{H}$ as defined in Appendix B. According to equation 107, the distribution of $(\omega_0, \omega_{t+1}) = ((s_0, a_0)), (s_{t+1}, a_{t+1}))$ is given as

$$\mu_{t+1}^\pi((s_0, a_0)), (s_{t+1}, a_{t+1}))$$
$$= \int \left( \prod_{\tau=0}^t \pi(a_{\tau+1}|s_{\tau+1}) P(s_{\tau+1}|s_\tau, a_\tau) \right) ds_1 da_1 \cdots ds_t da_t \pi(a_0|s_0) \mu_0(s_0)$$

Then the generalized $N$-step cross-covariance operator and its empirical counterpart for $(\omega_0, \omega_t)$ can be defined as

$$C_{\omega_0, \omega_t} = \mathbb{E}_{((S,A),(S',A')) \sim \mu_t^\pi}[\Phi(S, A) \otimes \Phi(S', A')], \quad \hat{C}_{\omega_0, \omega_t} = \frac{1}{n} \sum_{i=1}^n \Phi(\omega_0^{(i)}) \otimes \Phi(\omega_t^{(i)}).$$

Using the cross-covariance operators $\{C_{\omega_0, \omega_t}\}_{t=0}^N$, we then can rewrite the Bellman equation in equation 108 as:

$$C_{\omega_0, \omega_0} Q^\pi = \left( \sum_{t=0}^{N-1} \gamma^t C_{\omega_0, \omega_t} r \right) + \gamma^N C_{\omega_0, \omega_N} Q^\pi. \tag{110}$$

Equation 110 is exactly equivalent to the Bellman equation defined in equation 41. Since we only have access to the samples $\{(\omega_0^{(i)}, \cdots, \omega_N^{(i)})\}_{i=1}^n$, we can only work with the following penalized empirical counterpart of equation 110:

$$\hat{C}_{\omega_0, \omega_0} \hat{Q}^\pi - \left[ \left( \sum_{t=0}^{N-1} \gamma^t \hat{C}_{\omega_0, \omega_t} r \right) + \gamma^N \hat{C}_{\omega_0, \omega_N} \hat{Q}^\pi \right] + \lambda \hat{Q}^\pi = 0. \tag{111}$$

We call equation 111 the $N$-step empirical Bellman equation. This allows us to establish a generalized version of Lemma 15, which connects the $N$-step empirical Bellman equation to KRR.

**Lemma 24.** *Solution to KRR 109 is equivalent to the $N$-step empirical Bellman equation 111.*

*Proof.* Solution $\hat{Q}^\pi$ to the KRR equation 109 is the minimizer of the following functional

$$\mathcal{J}[f] = \frac{1}{n} \sum_{i=1}^n \left( f(\omega_0^{(i)}) - \left( \sum_{t=0}^{N-1} \gamma^t r(\omega_t^{(i)}) \right) - \gamma^N \hat{Q}^\pi(\omega_N^{(i)}) \right)^2 + \lambda \|f\|_{\mathcal{H}}^2. \tag{112}$$

The functional $\mathcal{J}$ is Fréchet differentiable obviously with RKHS derivative $\nabla_f \mathcal{J}[f]$ as follows:

$$\frac{1}{2}\partial_\delta \mathcal{J}[f + \delta u]\big|_{\delta=0} \tag{113}$$

$$=\langle \frac{1}{n}\sum_{i=1}^{n}\left(f(\omega_0^{(i)}) - \left(\sum_{t=0}^{N-1}\gamma^t r(\omega_t^{(i)})\right) - \gamma^N \hat{Q}^\pi(\omega_N^{(i)})\right)K(\omega_0^{(i)},\cdot) + \lambda f, u\rangle_{\mathcal{H}},$$

$$=\langle \nabla_f \mathcal{J}[f], u\rangle_{\mathcal{H}}, \quad \forall u \in \mathcal{H}. \tag{114}$$

Proceeding as in the proof of Lemma 15 yields

$$\nabla_f \mathcal{J}[f] = \hat{C}_{\omega_0,\omega_0}\hat{Q}^\pi - \left[\left(\sum_{t=0}^{N-1}\gamma^t \hat{C}_{\omega_0,\omega_t} r\right) + \gamma^N \hat{C}_{\omega_0,\omega_N}\hat{Q}^\pi\right] + \lambda \hat{Q}^\pi. \tag{115}$$

According to definition, $\hat{Q}^\pi$ is also the minimizer of $\mathcal{J}$. Substitute $f = \hat{Q}^\pi$ into equation 115, we can conclude that the minimization of equation 112 is equivalent to equation 109. □

By reasoning exactly the same as Lemma 16, we can also show that solution $\hat{Q}^\pi$ to the KRR equation 109 is finite-dimensional:

**Lemma 25.** Let $\hat{\mathcal{H}} = \{K(\omega_i, \cdot), i = 1, \cdots, n\}$ be the finite dimensional space spanned by kernel functions and $\hat{Q}^\pi$ be the solution to the KRR equation 109. We have $\hat{Q}^\pi \in \hat{\mathcal{H}}$.

Similar to the proof for Proposition 8, we can use Lemma 24 and Lemma 25 to derive the explicit form for the solution $\hat{Q}^\pi$ to the KRR equation 109.

From Lemma 25, we know that $\hat{Q}^\pi$ must be of the form:

$$\hat{Q}^\pi = \sum_{i=1}^{n} K(\cdot, \omega_i)b_i = K(\cdot, \boldsymbol{\omega}_0)\mathbf{b}. \tag{116}$$

Substitute equation 116 into equation 111, togethe with the definition of empirical operators $\{\hat{C}_{\omega_0,\omega_t}\}_{t=0}^{N}$, we then have

$$\frac{1}{n}\sum_{i=1}^{n}\Phi(\omega_0^{(i)})\sum_{j=1}^{n}b_j K(\omega_0^{(i)}, \omega_0^{(j)}) \tag{117}$$

$$-\frac{1}{n}\sum_{i=1}^{n}\Phi(\omega_0^{(i)})\left[\left(\sum_{t=0}^{N-1}\gamma^t r(\omega_t^{(i)})\right) + \gamma^N \sum_{j=1}^{n}b_j K(\omega_N^{(i)}, \omega_0^{(j)})\right]$$

$$+\lambda\sum_{i=1}^{n}b_i \Phi(\omega_0^{(i)}) = 0.$$

Rearrange equation 117 and write it in vector form, we have

$$\Phi(\boldsymbol{\omega}_0)\left[\mathbf{K}\mathbf{b} + \lambda n\mathbf{b} - \gamma^N \mathbf{C}_N \mathbf{b}\right] = \Phi(\boldsymbol{\omega}_0)\bar{\mathbf{r}}. \tag{118}$$

where $[\mathbf{C}_N]_{i,j} = K(\omega_N^{(i)}, \omega_0^{(j)})$ and $\bar{\mathbf{r}} = [\sum_{t=0}^{N-1}\gamma^t r(\omega_t^{(i)})]_i$. In summary, we have the following proposition:

**Proposition 26.** Let $\hat{Q}^\pi$ be the solution to the KRR equation 109. Then $\hat{Q}^\pi = K(\cdot, \boldsymbol{\omega}_0)\mathbf{b}$ where

$$\mathbf{b} = \left[\mathbf{K} + \lambda n\mathbf{I} - \gamma^N \mathbf{C}_N\right]^{-1}\bar{\mathbf{r}}. \tag{119}$$

## J.2 N-STEP KERNEL TEMPEROAL-DIFFERENCE LEARNING

From Proposition equation 26, we see that the KRR estimator in equation 109 has exactly the same form as the KRR formulation in equation 8. In specific,

$$\mathbf{b}^\pi = \min_{\mathbf{b}\in\mathbb{R}^n}\frac{1}{n}\sum_{i=1}^{n}\left(K(\omega_0^{(i)}, \boldsymbol{\omega}_0)\mathbf{b} - \bar{\mathbf{r}}_i - \gamma^N K(\omega_N^{(i)}, \boldsymbol{\omega}_0)\mathbf{b}^\pi\right)^2 + \lambda\mathbf{b}^\top\mathbf{K}\mathbf{b}. \tag{120}$$

We can observe that the differences between equation 120 and equation 8 are that the constants $r(\omega_0^{(i)})$, $\gamma$, and $K(\omega_1^{(i)}, \omega_0)$ in equation 8 are replaced by $\bar{\mathbf{r}}_i = \sum_{t=0}^{N-1} \gamma^t r(\omega_t^{(i)})$, $\gamma^N$, and $K(\omega_N^{(i)}, \omega_0)$ in equation 120, respectively. The changes in the constants in equation 120 do not affect the procedure for computing the solution $\mathbf{b}^\pi$. We can apply exactly the same iterative rule used for solving equation 8 to obtain $\mathbf{b}^\pi$ in equation 120:

$$f_{t+1} = (1 - \alpha_t) f_t - \eta_t \sum_{i=1}^{n} \left( f_t(\omega_0^{(i)}) - \bar{\mathbf{r}}_i - \gamma^N f_t(\omega_N^{(i)}) \right) K(\omega_0^{(i)}, \cdot), \tag{121}$$

It can also be noticed that equation 121 can be converted to a form similar to semi-gradient TD(T) (Sutton et al., 1998) if we represented it by $\mathbf{b}_t = \mathbf{K}^{-1} f_t(\boldsymbol{\omega}_0)$:

$$\mathbf{b}_{t+1} = (1 - \alpha_t) \mathbf{b}_t - \eta_t \left( f_t(\boldsymbol{\omega}_0) - \bar{\mathbf{r}} - \gamma^N f_t(\boldsymbol{\omega}_T) \right). \tag{122}$$

Similar to the one-step case, the difference between equation 122 and semi-gradient TD(T) in Sutton (1988) lies in the fact that equation 122 uses a functional gradient in the infinite-dimensional RKHS, which can be treated as a preconditioning to improve the performance of the algorithm.

## J.3 CONVERGENCE ANALYSIS OF N-STEP KERNEL TD

We first generalize the error decomposition in Proposition 9 for one-step TD. Define for the $N$-step KRR equation 109 the difference function $\mathcal{D}^\pi = \hat{Q}^\pi - Q^\pi$ as before and the $N$-step Bellman residual:

$$\varepsilon_i = \left( \sum_{t=0}^{N-1} \gamma^t r(\omega_t^{(i)}) \right) + \gamma^N Q^\pi(\omega_N^{(i)}) - Q^\pi(\omega_0^{(i)}). \tag{123}$$

We have the following error decomposition for the $N$-step kernel TD
**Proposition 27.**

$$\frac{1}{n} \sum_{i=1}^{n} \left( \mathcal{D}^\pi(\omega_0^{(i)})^2 - \gamma^N \mathcal{D}^\pi(\omega_0^{(i)}) \mathcal{D}^\pi(\omega_N^{(i)}) \right) = \frac{1}{n} \sum_{i=1}^{n} \varepsilon_i \mathcal{D}^\pi(\omega_0^{(i)}) - \lambda \langle \mathcal{D}^\pi, \hat{Q}^\pi \rangle_{\mathcal{H}}. \tag{124}$$

*Proof.* Substitute equation 110 into equation 111, we can have

$$\left[ \hat{C}_{\omega_0, \omega_0} + \lambda \mathrm{I} \right] \hat{Q}^\pi - C_{\omega_0, \omega_0} Q^\pi = \left[ \sum_{t=0}^{N-1} \gamma^t \left( \hat{C}_{\omega_0, \omega_t} - C_{\omega_0, \omega_t} \right) r \right] + \gamma^N \left[ \hat{C}_{\omega_0, \omega_N} \hat{Q}^\pi - C_{\omega_0, \omega_N} Q^\pi \right]. \tag{125}$$

Recall that $\mathcal{D}^\pi = \hat{Q}^\pi - Q^\pi$, we can rewrite equation 125 as

$$\left[ \hat{C}_{\omega_0, \omega_0} + \lambda \mathrm{I} - \gamma^N \hat{C}_{\omega_0, \omega_N} \right] \mathcal{D}^\pi \tag{126}$$

$$= \left( \sum_{t=0}^{N-1} \gamma^t \hat{C}_{\omega_0, \omega_t} r \right) + \gamma^N \hat{C}_{\omega_0, \omega_N} Q^\pi - \hat{C}_{\omega_0, \omega_0} Q^\pi - \lambda Q^\pi \tag{127}$$

Taking the Hilbert space inner product with $\mathcal{D}^\pi$ in equation 126 and equation 127:

$$\langle \mathcal{D}^\pi, \left[ \hat{C}_{\omega_0, \omega_0} + \lambda \mathrm{I} - \gamma^N \hat{C}_{\omega_0, \omega_N} \right] \mathcal{D}^\pi \rangle_{\mathcal{H}} \tag{128}$$

$$= \langle \mathcal{D}^\pi, \left( \sum_{t=0}^{N-1} \gamma^t \hat{C}_{\omega_0, \omega_t} r \right) + \gamma^N \hat{C}_{\omega_0, \omega_N} Q^\pi - \hat{C}_{\omega_0, \omega_0} Q^\pi - \lambda Q^\pi \rangle_{\mathcal{H}}. \tag{129}$$

According to the definition of empirical covariance operators, equation 128 can be further rewritten as

$$\langle \mathcal{D}^\pi, \left[ \hat{C}_{\omega_0, \omega_0} + \lambda \mathrm{I} - \gamma^N \hat{C}_{\omega_0, \omega_N} \right] \mathcal{D}^\pi \rangle_{\mathcal{H}} \tag{130}$$

$$= \frac{1}{n} \sum_{i=1}^{n} \mathcal{D}^\pi(\omega_0^{(i)})^2 - \frac{\gamma^N}{n} \sum_{i=1}^{n} \mathcal{D}^\pi(\omega_0^{(i)}) \mathcal{D}^\pi(\omega_N^{(i)}) + \lambda \|\mathcal{D}^\pi\|_{\mathcal{H}}^2$$

and using the definition of empirical cross-covariance operators, equation 129 can be further rewritten as

$$\langle \mathcal{D}^\pi, \left( \sum_{t=0}^{N-1} \gamma^t \hat{C}_{\omega_0,\omega_t} r \right) + \gamma^N \hat{C}_{\omega_0,\omega_N} Q^\pi - \hat{C}_{\omega_0,\omega_0} Q^\pi - \lambda Q^\pi \rangle_{\mathcal{H}} \qquad (131)$$

$$= \frac{1}{n} \sum_{i=1}^{n} \varepsilon_i \mathcal{D}^\pi(\omega_0^{(i)}) - \lambda \langle \mathcal{D}^\pi, Q^\pi \rangle_{\mathcal{H}}.$$

Because equation 130 and equation 131 are equals, by rearranging their terms, we can have the final result. □

From Proposition 26 and Proposition 27, the $N$-step TD can be entirely reformulated as an equivalent one-step TD with Bellman residuals having smaller variance. This transformation requires changing only Assumption 6 to Assumption 6B and entails a direct mapping: the $N$-step discount $\gamma^N$ becomes the new one-step discount factor, the state-action sequence $\omega_N$ serves as the one-step state-action input, $\bar{\mathbf{r}}$ is treated as the associated one-step reward, and equation 123 is the corresponding one-step Bellman residual. Consequently, replacing the one-step TD in the main paper with $N$-step TD leaves the convergence results of Theorem 10, Corollary 11, and Corollary 14 unchanged, requiring only the substitution of $\gamma$ with $\gamma^N$. This substitution results in faster convergence rates.

## J.4 Extension to Kernelized TD($\rho$)

Following equation 110, we can further rewrite the Bellman equation using cross-covariance operators in the following form

$$C_{\omega_0,\omega_0} Q^\pi = (1-\rho) \sum_{N=1}^{N^*} \rho^{N-1} \left[ \left( \sum_{t=0}^{N-1} \gamma^t C_{\omega_0,\omega_t} r \right) + \gamma^N C_{\omega_0,\omega_N} Q^\pi \right]. \qquad (132)$$

Following the same reasoning as the previous subsections, we can directly derive that if we take an $N^*$-step samples $\{\omega_0^{(i)}, \cdots, \omega_{N^*}^{(i)}\}_{i=1}^n$, then we can have the empirical counterpart of equation 132 :

$$\hat{C}_{\omega_0,\omega_0} \hat{Q}^\pi - (1-\rho) \sum_{N=1}^{N^*} \rho^{N-1} \left[ \left( \sum_{t=0}^{N-1} \gamma^t \hat{C}_{\omega_0,\omega_t} r \right) + \gamma^N \hat{C}_{\omega_0,\omega_N} \hat{Q}^\pi \right] + \lambda \hat{Q}^\pi = 0. \qquad (133)$$

which can be reformulated as the following fixed-point KRR

$$\hat{Q}^\pi = \min_{f \in \mathcal{H}} \frac{1}{n} \sum_{i=1}^{n} \left( f(\omega_0^{(i)}) - (1-\rho) \sum_{N=1}^{N^*} \rho^{N-1} \left[ \left( \sum_{t=0}^{N-1} \gamma^t r(\omega_t^{(i)}) \right) + \gamma^N \hat{Q}^\pi(\omega_N^{(i)}) \right] \right)^2 + \lambda \|f\|_{\mathcal{H}}^2. \qquad (134)$$

Following the same reasoning, we have the following update rule for solving equation 134

$$f_{t+1} = (1-\alpha_t) f_t - \eta_t \sum_{i=1}^{n} \left( f_t(\omega_0^{(i)}) - \bar{\mathbf{r}}_i - (1-\rho) \sum_{N=1}^{N^*} \rho^{N-1} \gamma^N f_t(\omega_N^{(i)}) \right) K(\omega_0^{(i)}, \cdot), \qquad (135)$$

where $\bar{\mathbf{r}}_i = (1-\rho) \sum_{N=1}^{N^*} \rho^{N-1} \sum_{t=0}^{N-1} \gamma^t r(\omega_t^{(i)})$.

The main difference between TD($\lambda$) and $N$-step TD is that, by carefully selecting $\rho$ and $N^*$, the corresponding Bellman residual can be made arbitrarily small at the cost of increasing sampling steps $N^*$. This yields a faster convergence rate in terms of the number of trajectories $n$. Consequently, the convergence behavior of TD($\rho$) may differ from that of $N$-step TD, and we leave a detailed analysis of this for future study.

## K Implicit Proximal Policy Optimization

As we have mentioned, in contrast to the original formulation of PPO (Schulman et al., 2017), the variant in 19 replaces $KL\left(\pi^k(\cdot|s), |, \pi(\cdot|s)\right)$ with $KL\left(\pi(\cdot|s), |, \pi^k(\cdot|s)\right)$. This choice yields a closed-form update for $\pi^{k+1}$, and we therefore refer to it as explicit PPO. In this section, we show that the original PPO can be treated as an implicit counterpart of 19.

## K.1 IMPLICIT PPO IN RKHS

The update rule of the original version of PPO in RKHS $\mathcal{H}$ is given by

$$F^{(k+1)} = \arg\max_{F \in \mathcal{H}} \mathbb{E}_{s \sim \mu_0^k} \left[ \Delta_k \langle f^{(k)}(s,\cdot), \pi_F(\cdot|s)\rangle_{\mathcal{A}} - KL\left(\pi^k(\cdot|s)\|\pi_F(\cdot|s)\right) \right] \quad (136)$$

where policy is represented as $\pi_F \propto \exp\{F\}$, $\pi^{(k)} \propto \exp\{F^{(k)}\}$ is the policy from the last update, and $f^{(k)}$ is the kernel TD estimator of the Q-function $Q^{(k)}$ trained for T iterations. The solution to equation 136 does not admit a closed-form expression, but it has an implicit representation analogous to implicit SGD—also referred to as the proximal Robbins–Monro method in Toulis et al. (2021)—given as follows:

**Lemma 28.** *The solution to equation 136 satisfies:*

$$\pi^{k+1}(a|s) = \pi^k(a|s) + \Delta_k \pi^{k+1}(a|s) \left[ f^{(k)}(s,a) - \langle f^{(k)}(s,\cdot), \pi^{k+1}(\cdot|s)\rangle_{\mathcal{A}} \right] \quad (137)$$

*for almost all $(s,a) \in \mathcal{S} \times \mathcal{A}$.*

*Proof.* We first define the functional over functions in $\mathcal{H}$:

$$\mathcal{J}[F] = \mathbb{E}_{s \sim \mu_0^k} \left[ \Delta_k \langle f^{(k)}(s,\cdot), \pi_F(\cdot|s)\rangle_{\mathcal{A}} - KL\left(\pi_F(\cdot|s)\|\pi^k(\cdot|s)\right) \right]$$

Let $\mathcal{H}^*$ be the dual space of $\mathcal{H}$. We can take the Fréchet functional derivative of $\mathcal{J}$ at a function $F$ along a $g \in \mathcal{H}^*$ and set it to 0:

$$\partial_\delta \mathcal{J}[F + \delta g]\Big|_{\delta=0} = 0. \quad (138)$$

According to equation 136, $F^{(k+1)}$ is a minimizer of $\mathcal{J}$ so it satisfies equation 138 for any $g \in \mathcal{H}^*$. From direct calculations, we can have

$$\begin{aligned}
0 = \int_{\mathcal{S}} \Bigg[ & \Delta_k \int_{\mathcal{A}} f^{(k)}(s,a)\pi^{k+1}(a|s)g(s,a)da \\
& - \Delta_k \int_{\mathcal{A}} f^{(k)}(s,a)\pi^{k+1}(a|s)da \int_{\mathcal{A}} \pi^{k+1}(a|s)g(s,a)da \\
& + \int_{\mathcal{A}} \left[\pi^k(a|s) - \pi^{k+1}\right] g(s,a)da \Bigg] d\mu_0^k(s).
\end{aligned} \quad (139)$$

Equation 139 must hold for any $g \in \mathcal{H}^*$. So inside the integral, we must have for almost all $(s,a) \in \mathcal{S} \times \mathcal{A}$:

$$g(s,a) \left( \Delta_k \pi^{k+1}(a|s) \left[ f^{(k)}(s,a) - \int_{\mathcal{A}} f^{(k)}(s,\alpha)\pi^{k+1}(\alpha|s)d\alpha \right] + \left[\pi^k(a|s) - \pi^{k+1}\right] \right) = 0. \quad (140)$$

By factoring out $g(s,a)$ and rearranging the terms inside the brackets in equation 140, we obtain equation 137. $\qquad \square$

We can note that the term $\pi^{k+1}(a|s)\left[f^{(k)}(s,a) - \langle f^{(k)}(s,\cdot), \pi^{k+1}(\cdot|s)\rangle_{\mathcal{A}}\right]$ is exactly the functional derivative of $\mathcal{U}[F] := \langle f^{(k)}(s,\cdot), \pi_F(\cdot|s)\rangle_{\mathcal{A}}$ evaluated at $f^{(k+1)}$. The functional $\mathcal{U}$ serves as a surrogate of the expected total reward functional $\mathcal{R}$ in equation 21, with the stationary distribution $\nu^*$ replaced by the initial distribution $\mu_0^k$. Consequently, equation 137 is directly analogous to an implicit SGD update:

$$\pi^{k+1} = \pi^k + \Delta_k \partial \mathcal{U}[f^{(k+1)}].$$

Equation 137 is a non-linear Fredholm integral equation of the second kind. In the case that $\mathcal{A}$ is discrete and the step size $\Delta_k$ is small enough, it follows directly from the contraction mapping theorem that it can be solved using the following iterative method

$$\pi(s|a) \leftarrow \pi^k(s|a) + \Delta_k \pi(a|s) \left[ f^{(k)}(s,a) - \langle f^{(k)}(s,\cdot), \pi(\cdot|s)\rangle_{\mathcal{A}} \right], \quad \forall a \in \mathcal{A}. \quad (141)$$

However, developing a numerical method for solving equation 137 with general $\mathcal{S} \times \mathcal{A}$ lies beyond the scope of this work. We therefore assume that it can be solved with sufficiently high accuracy such that the numerical error is negligible. So implicit PPO can be summarized as Algorithm 2.

---

**Algorithm 2** Implicit PPO in RKHS

---

1: **Require**: MDP $(\mathcal{S}, \mathcal{A}, P, r, \gamma)$, RKHS $\mathcal{H}$
2: Initialize $\pi^0 \propto \exp\{f^{(0)}\}$ for some $f^{(0)} \in \hat{\mathcal{H}}$
3: **for** $k = 0, 1, , 2, \ldots$ **do**
4:      Select an initial sampling distribution $\mu_0^k$ and number of samples $n \leftarrow n^{(k)}$
5:      Generate $\{[s_0^{(i)}, a_0^{(i)}, s_1^{(i)}, a_1^{(i)}] \sim \mu_0^k(s_0) \pi^k(a_0|s_0) P(s_1|s_0, a_0) \pi^k(a_1|s_1)\}_{i=1}^n$
6:      Set $T \leftarrow T^{(k)}, \alpha \leftarrow \alpha^{(k)}, \eta \leftarrow \eta^{(k)}$
7:      Initialize $f^{(k)}$
8:      **for** $t = 1, \cdots, T$ **do**
9:          Update $f^{(k)} \leftarrow (1 - \alpha) f^{(k)} - \eta \sum_{i=1}^n \left( f^{(k)}(\omega_0^{(i)}) - r(\omega_0^{(i)}) - \gamma f^{(k)}(\omega_1^{(i)}) \right) K(\omega_0^{(i)}, \cdot)$
10:      **end for**
11:      Solve $\pi^{k+1}(a|s) = \pi^k(a|s) + \Delta_k \pi^{k+1}(a|s) \left[ f^{(k)}(s, a) - \langle f^{(k)}(s, \cdot), \pi^{k+1}(\cdot|s) \rangle_{\mathcal{A}} \right]$
12: **end for**

---

### K.2 GLOBAL CONVERGENCE OF IMPLICIT PPO

Similar to the explicit case, we can have a fundamental inequality for the one-step improvement equation 21.

**Theorem 29.** *In Algorithm 2,*

$$
\inf_k \left( \mathcal{R}[\pi^k] - \mathcal{R}[\pi^*] \right)
$$
$$
\leq \frac{\left( \sum_k (2\Delta_k + \Delta_k^2) \|f^{(k)} - Q^{(k)}\|_{L_\infty} \right) + \left( \sum_k \Delta_k^2 (1-\gamma)^{-1} \|r\|_{L_\infty} \right) + \mathbb{E}_{S \sim \nu^*} KL \left( \pi^*(\cdot|S) \| \pi^0(\cdot|S) \right)}{\sum_k \Delta_k}.
$$
(142)

*Proof.* For polices $\pi^{k+1}$ and $\pi^k$, we have, for $s \sim \nu^*$, the one-step improvement in KL

$$
\mathbb{E}_{s \sim \nu^*} \left[ KL \left( \pi^*(\cdot|s) \| \pi^{k+1}(\cdot|s) \right) - KL \left( \pi^*(\cdot|s) \| \pi^k(\cdot|s) \right) \right] \tag{143}
$$
$$
= \mathbb{E}_{s \sim \nu^*} \left[ \int_{\mathcal{A}} \pi^*(a|s) \log \frac{\pi^k(a|s)}{\pi^{k+1}(a|s)} da \right]
$$
$$
\leq \mathbb{E}_{s \sim \nu^*} \int_{\mathcal{A}} \pi^*(a|s) \frac{\pi^k(a|s) - \pi^{k+1}(a|s)}{\pi^{k+1}(a|s)} da
$$
$$
= \mathbb{E}_{s \sim \nu^*} \left[ -\int_{\mathcal{A}} \pi^*(a, s) \Delta_k \int_{\mathcal{A}} \left[ f^{(k)}(s, a) - f^{(k)}(s, a') \right] \pi^{k+1}(a'|s) da' da \right]
$$
$$
= \mathbb{E}_{s \sim \nu^*} \left[ -\Delta_k \langle f^{(k)}(s, \cdot), \pi^*(\cdot|s) - \pi^{k+1}(\cdot|s) \rangle_{\mathcal{A}} \right]
$$
$$
= \mathbb{E}_{s \sim \nu^*} \left[ -\Delta_k \langle f^{(k)}(s, \cdot), \pi^*(\cdot|s) - \pi^k(\cdot|s) \rangle_{\mathcal{A}} + \Delta_k^2 \text{Var}_{A \sim \pi^{k+1}(\cdot|s)} \left[ f^{(k)}(A, s) \right] \right]
$$
$$
= \mathbb{E}_{s \sim \nu^*} \left[ \underbrace{-\Delta_k \langle Q^{(k)}(s, \cdot), \pi^*(\cdot|s) - \pi^k(\cdot|s) \rangle_{\mathcal{A}}}_{A} \right.
$$
$$
\left. + \underbrace{\Delta_k \langle Q^{(k)}(s, \cdot) - f^{(k)}(s, \cdot), \pi^*(\cdot|s) - \pi^k(\cdot|s) \rangle_{\mathcal{A}}}_{B} + \underbrace{\Delta_k^2 \text{Var}_{A \sim \pi^{k+1}(\cdot|s)} \left[ f^{(k)}(A, s) \right]}_{C} \right]
$$

where the third line is from inequality $\log x \leq x - 1$ for any $x \geq 0$, the fourth line is from the updating policy equation 137, and the sixth line is because

$$\Delta_k \langle f^{(k)}(s, \cdot), \pi^{k+1}(\cdot|s) - \pi^k(\cdot|s) \rangle_{\mathcal{A}}$$

$$= \Delta_k^2 \left[ \left( \sum_{a \in \mathcal{A}} \left| f^{(k)}(s, a) \right|^2 \pi^{k+1}(a|s) \right) - \left( \sum_{a \in \mathcal{A}} f^{(k)}(s, a) \pi^{k+1}(a|s) \right)^2 \right]$$

$$= \Delta_k^2 \mathrm{Var}_{A \sim \pi^{k+1}(\cdot|s)} \left[ f^{(k)}(A, s) \right].$$

For term A, we use Lemma 23 to directly have:

$$\mathbb{E}_{S \sim \nu^*} \Delta_k \langle Q^{(k)}(\cdot, S), \pi^*(\cdot|S) - \pi^k(\cdot|S) \rangle_{\mathcal{A}} = \Delta_k \left( \mathcal{R}[\pi^*] - \mathcal{R}[\pi^k] \right). \tag{144}$$

For term B

$$\left| \mathbb{E}_{S \sim \nu^*} \Delta_k \langle f^{(k)}(\cdot, S) - Q^{(k)}(\cdot, S), \pi^*(\cdot|S) - \pi^k(\cdot|S) \rangle_{\mathcal{A}} \right|$$

$$\leq \Delta_k \| f^{(k)} - Q^k \|_{L_\infty} \left( \int \pi^*(a|s) \nu^*(s) dads + \int \pi^k(a|s) \nu^*(s) dads \right) \tag{145}$$

$$= 2\Delta_k \| f^{(k)} - Q^k \|_{L_\infty}.$$

For term C

$$\mathrm{Var}_{A \sim \pi^{k+1}(\cdot|s)} \left[ f^{(k)}(A, s) \right] = \mathrm{Var}_{A \sim \pi^{k+1}(\cdot|s)} \left[ Q^{(k)}(A, s) + \left( f^{(k)}(A, s) - Q^{(k)}(A, s) \right) \right]$$

$$\leq \| Q^{(k)} \|_{L_\infty} + \| f^{(k)} - Q^{(k)} \|_{L_\infty} \leq \frac{\| r \|_{L_\infty}}{1 - \gamma} + \| f^{(k)} - Q^{(k)} \|_{L_\infty}. \tag{146}$$

Substitute the above upper bounds equation 144, equation 145, and equation 146 into equation 143 and rearrange the terms, we have

$$\Delta_k \left( \mathcal{R}[\pi^*] - \mathcal{R}[\pi^k] \right) + \mathbb{E}_{S \sim \nu^*} \left[ \mathrm{KL} \left( \pi^*(\cdot|S) || \pi^{k+1}(\cdot|S) \right) - \mathrm{KL} \left( \pi^*(\cdot|S) || \pi^k(\cdot|S) \right) \right]$$

$$\leq (2\Delta_k + \Delta_k^2) \| f^{(k)} - Q^{(k)} \|_{L_\infty} + \Delta_k^2 (1 - \gamma)^{-1} \| r \|_{L_\infty}. \tag{147}$$

By summing over all the $k$ in equation 147, we can have the final result. $\qquad \square$

We observe that Theorem 29 for implicit PPO is almost identical to Theorem 13, except for the difference in the constants $2\Delta_k + \Delta_k^2$ versus $2\Delta_k$ in the $L_\infty$ error term $\| f^{(k)} - Q^{(k)} \|_{L_\infty}$. However, as $\Delta_k \to 0$, the difference vanishes. As a result, the sampling requirement in Corollary 14 for Algorithm 1 also applies to Algorithm 2, since both algorithms share the same policy evaluation error and their one-step improvements differ only by a multiplicative constant:

**Corollary 30.** *Let* $\{\pi^k\}_{k=1}^{k^*}$ *be the policies induced by Algorithm 2. Set* $\Delta_k = 1/\sqrt{k}$. *For settings listed in Corollary 11, set* $n^{(k)}$ *and* $\lambda^{(k)}$ *according to Table 1. Set* $\alpha^{(k)}, \eta^{(k)}$, *and* $T^{(k)}$ *according to Theorem 10 with* $\lambda = \lambda^{(k)}$. *Then under the same conditions as Corollary 11, we have*

$$\inf_{1 \leq k \leq k^*} \left( \mathcal{R}[\pi^*] - \mathcal{R}[\pi^k] \right) \leq \mathcal{O}_p(\frac{1}{\sqrt{k^*}}). \tag{148}$$

# L  EXPERIMENT SUPPLEMENTS

## L.1  RL ENVIRONMENTS

*CartPole-v1 (Barto et al., 2012)*: A classic control task with 4-dimensional continuous state space (cart position, cart velocity, pole angle, pole angular velocity) and 2 discrete actions (push left/right). The goal is to balance a pole on a cart, with an optimal reward of 500. The default duration of an episode is 500 time steps.

*Acrobot-v1 (Sutton et al., 1998)*: An underactuated robotics task with 6-dimensional continuous state space (cosine/sine of both joint angles, angular velocities) and 3 discrete actions (torque values

$\{-1, 0, +1\}$). The objective is to swing the free end above a target height, with an optimal reward of $-100$. The default duration of an episode is 500 time steps.

*HalfCheetah-v5 (Wawrzyński, 2009)*: A continuous control task from the MuJoCo physics simulator with 17-dimensional continuous state space and 6-dimensional continuous action space. The state space (Box$(-\infty, \infty, (17, ), \text{float64})$) includes the root position and orientation, joint angles, and angular velocities for 6 hinge joints (back/front thigh, shin, and foot). The action space (Box$(-1, 1, (6, ), \text{float32})$) represents torques applied to these joints. The goal is to maximize forward velocity, with rewards typically ranging from 0 to several thousand, depending on the policy performance. The Half Cheetah never terminates, and the default duration of an episode is 1,000 time steps.

## L.2 RUN TIME AND MEMORY COST ANALYSIS

To thoroughly evaluate the computational complexity and scalability of our proposed NPG algorithm, we conducted a comprehensive analysis of runtime and memory costs across tasks with varying state-action dimensions and under different step size schedules. The results for discrete control tasks (CartPole-v1 and Acrobot-v1) are summarized in Table 2, while Table 3 presents the performance on the higher-dimensional continuous control task (HalfCheetah-v5). The results correspond to Figure 1: the discrete environment runs 1,000 episodes, and the continuous environment runs 10,000 episodes.

Table 2: Performance comparison on Cartpole-v1 and Acrobot-v1.

| Environment | Cartpole-v1 | | | Acrobot-v1 | | |
|---|---|---|---|---|---|---|
| Step size | $k^{-0.2}$ | $k^{-0.5}$ | $k^{-1.5}$ | $k^{-0.2}$ | $k^{-0.5}$ | $k^{-1.5}$ |
| Running time (s) | $30.26 \pm 4.49$ | $49.04 \pm 7.59$ | $7.85 \pm 2.42$ | $80.82 \pm 20.12$ | $49.77 \pm 25.28$ | $96.07 \pm 0.54$ |
| Peak Memory cost (MB) | $282.06 \pm 6.79$ | $272.77 \pm 0.10$ | $273.00 \pm 0.04$ | $194.28 \pm 60.62$ | $230.04 \pm 0.17$ | $230.52 \pm 0.16$ |

Table 3: Performance comparison on HalfCheetah-v5.

| Step size | $k^{-0.1}$ | $k^{-0.5}$ | $k^{-1.5}$ |
|---|---|---|---|
| Running time (s) | $834.23 \pm 18.56$ | $911.84 \pm 48.86$ | $917.94 \pm 72.95$ |
| Peak Memory cost (MB) | $202.65 \pm 70.77$ | $116.47 \pm 8.26$ | $160.79 \pm 58.42$ |

**Analysis:** Our method demonstrates high computational efficiency. In lower-dimensional settings, for CartPole-v1 (state dimension 4, action dimension 2), the average runtime remains under 50 seconds across all step sizes; in Acrobot-v1 (state dimension 6, action dimension 3), runtimes are similarly efficient, ranging from approximately 50 to 96 seconds. $k^{-0.5}$ often yields a favorable balance between convergence speed and stability.

For the high-dimensional HalfCheetah-v5 task (state dimension 17, action dimension 6), the algorithm scales gracefully. Despite the significant increase in state-action space complexity and episode length (1000 steps), the runtime remains manageable, averaging between 834 and 918 seconds for 10,000 episodes.

Memory usage remains stable and relatively low across all environments, confirming the practical feasibility of our approach.

In summary, our proposed NPG algorithm exhibits robust scalability to high-dimensional environments. As our NPG does not require the ordered trajectory sampling, if we use the GPU device, the runtime would be smaller due to parallel computing.

## L.3 BASELINE ALGORITHMS

We compare our proposed NPG method with the following baseline algorithms:

- **Clip-PPO with GAE (Generalized Advantage Estimation)**: A standard policy gradient method that uses a value function $V(s)$ to estimate advantages via GAE (Schulman et al., 2017). The policy update employs a clipped surrogate objective $L^{\text{clip}}(\theta) = \mathbb{E}_t\big[\min\big(r_t(\theta)A_t, \text{clip}(r_t(\theta), 1 - \epsilon, 1 + \epsilon)A_t\big)\big]$, where $r_t(\theta)$ is the importance ratio be-

tween new and old policies, and advantages are computed as $A_t = \delta_t + (\gamma\lambda)\delta_{t+1} + \cdots$ with $\delta_t = r_t + \gamma V(s_{t+1}) - V(s_t)$.

- **Discrete Soft Actor-Critic (SAC)**: An off-policy actor-critic algorithm adapted for discrete action spaces (Haarnoja et al., 2018). SAC maintains two Q-networks to reduce overestimation bias and uses automatic entropy tuning. The policy is updated to maximize the expected Q-value minus an entropy term, encouraging exploration while maintaining sample efficiency.

- **Deep Q-Network (DQN)**: A value-based method that learns action-value functions $Q(s, a)$ using temporal difference learning (Mnih et al., 2015). DQN employs experience replay and target networks for stable learning. Actions are selected using $\epsilon$-greedy exploration, and the policy is implicitly defined as the greedy action selection $\pi(s) = \arg\max_a Q(s, a)$.

- **Implicit PPO**: We explain the details in Appendix K.

**Results Analysis.** We show the results of Cartpole-v1 as an example in Figure 3. Our results indicate that implicit PPO achieves performance similar to our NPG, and it converges to the optima faster. The reason is that the implicit way, like implicit SGD, performs more stably in policy iteration. The classic clip-PPO is less efficient and more unstable than our NPG, as GAE used in clip-PPO requires the complete trajectory; instead, our NPG learns from one-step TD error. Discrete SAC failed to find the optimal solution in this setting. DQN, while efficient, exhibited unstable performance across seeds.

As for runtime, although our NPG and implicit PPO have the largest total runtime when run for the full 4,000 episodes in Figure 3, both algorithms already converge within the first 1,000 episodes. If we instead measure runtime only over these first 1,000 episodes—consistent with the setting in Figure 1 and Table 2—the training of NPG and implicit PPO would complete in under 50 seconds.

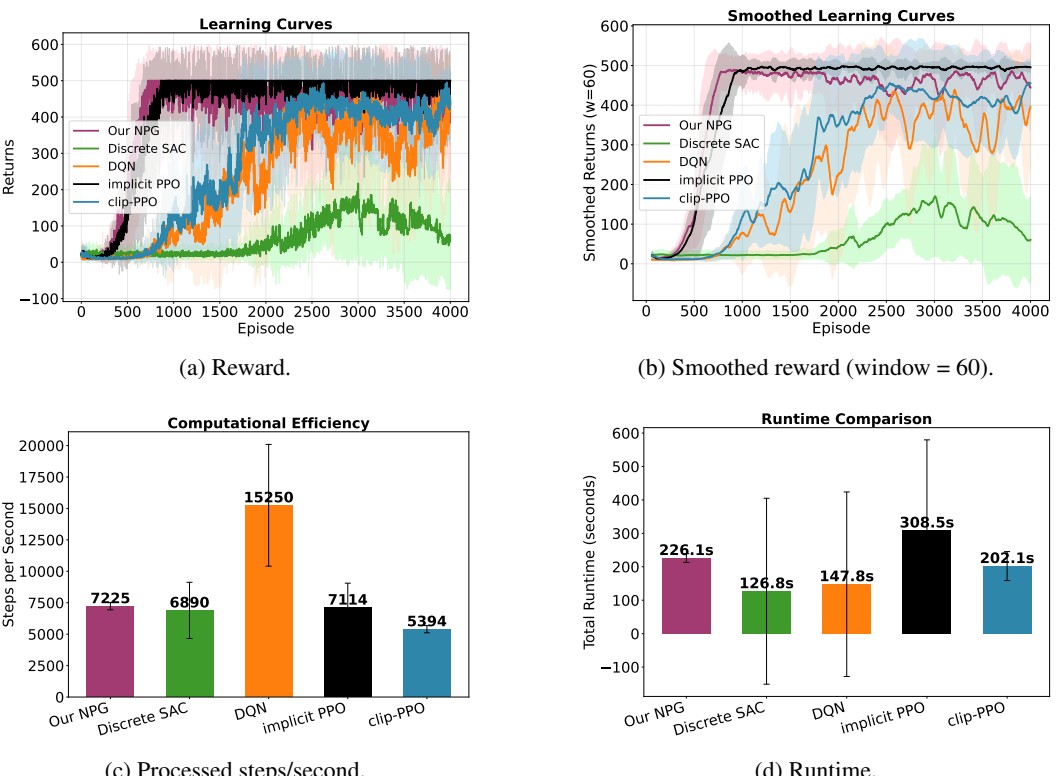

(a) Reward.

(b) Smoothed reward (window = 60).

(c) Processed steps/second.

(d) Runtime.

Figure 3: Performance and efficiency comparison on CartPole-v1 under baselines over 10 seeds.

### L.4 HYPERPARAMETER CONFIGURATION

Table 4 shows the hyperparameter setting of convergence analysis experiments in Section 6.2 as well as the runtime and memory cost analysis in Section L.2. We use the CPU with Apple M4, Arm64.

Table 4: Hyperparameters of convergence analysis in Figure 1.

| Hyperparameter | CartPole-v1 | Acrobot-v1 | HalfCheetah-v5 |
|---|---|---|---|
| Optimizer | | Adam | |
| Network | | MLP with two hidden layers | |
| Hidden width $h_{\text{dim}}$ for two hidden layer | | 64 | |
| TD Error Loss Coefficient | | 1.0 | |
| Learning rate $\eta$ | $1 \times 10^{-3}$ | $2.5 \times 10^{-4}$ | $3 \times 10^{-4}$ |
| Activation | ReLU | ReLU | Tanh |
| Batch size | 32 | 32 | 2048 |
| Algorithm 1 Optimization Epochs ($T$) | 4 | 4 | 10 |

Table 5 presents the hyperparameter configuration used in our comparative study (Section 6.3) between classic PPO with GAE and our novel NPG implementation with TD learning on CartPole-v1.

Table 5: Hyperparameter Configuration for CartPole-v1 Comparison Study in Figure 2.

| Parameter | Value | Unit | Description |
|---|---|---|---|
| *Environment Configuration* | | | |
| Training Episodes | 4,000 | episodes | Total training episodes |
| Update Frequency | 128 | steps | Steps per PPO update |
| *Learning Parameters* | | | |
| Learning Rate | $1 \times 10^{-3}$ | - | Adam optimizer rate |
| Discount Factor ($\gamma$) | 0.99 | - | Future reward discounting |
| GAE Lambda ($\lambda$) | 0.95 | - | GAE bias-variance trade-off |
| PPO $\epsilon$ | 0.2 | - | PPO clip parameter |
| *KL Penalty Configuration* | | | |
| KL Schedule | $k^{-0.5}$ | - | Dynamic $\beta = k^{\text{schedule\_pow}}$ |
| Initial KL Penalty | 1.0 | - | Starting $\beta_0$ coefficient |
| Entropy Coefficient | 0.01 | - | Policy entropy regularization |
| *Training Configuration* | | | |
| Value Loss Coefficient | 1.0 | - | Critic loss weighting |
| PPO Epochs | 4 | - | Policy update iterations |
| Batch Size | 64 | samples | Mini-batch size |
| *Architecture* | | | |
| Hidden Dimensions | 64 | neurons | Network layer width |
| Activation | ReLU | - | Non-linear activation |
| Optimizer | Adam | - | Gradient descent |
| Package | Pytorch | - | |
| Device | CPU | - | Apple M4, Arm64 |

### L.5 N-STEP TD LEARNING

We verify the generalization of our algorithm to N-step TD learning. The experimental results under three look-ahead horizon lengths $N$ on the `HalfCheetah-v5` are presented in Figure 4. We observe that the convergence pattern is similar as Figure 1, where the step size under $\Delta_k = k^{-\frac{1}{2}}$ achieves

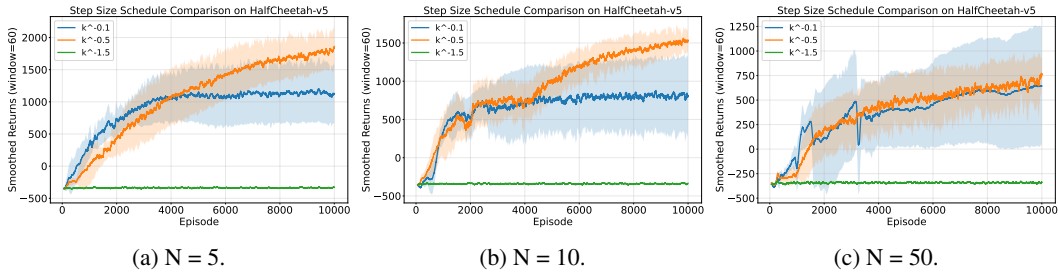

(a) N = 5.    (b) N = 10.    (c) N = 50.

Figure 4: Smoothed Reward of N-step TD learning on HalfCheetah-v5.

the optimal performance. Moreover, we find a clear trade-off between bias and variance. Shorter horizons ($N = 5$ and $N = 10$) yield stable learning curves with lower variance. However, increasing the horizon to $N = 50$ destabilizes the training. It is attributed to the high variance accumulated over long trajectories, which outweighs the benefits of reduced bias in the value estimation.

## M    USE OF LLMS

In this paper, LLMs were used solely for writing polishing in several paragraphs, like the Experiment section. All the key ideas, proofs, research, and writing are created completely by human authors.

