# OpenReview forum: "Sampling Complexity of TD and PPO in RKHS"
_ICLR.cc/2026/Conference — ICLR 2026 Poster_

### Official Review · Reviewer_4UFk · 2025-10-28

**Soundness:** 2
**Presentation:** 1
**Contribution:** 2
**Rating:** 4
**Confidence:** 2

**Summary:**

The paper revisits the theory of temporal difference value estimation and natural policy gradient from the perspective of function spaces. First, it sets up a value estimation setting in which the value function lies in a RKHS. Under some assumptions, the paper provides a result on the dynamic of the estimation error of the value function with a kernelized temporal difference method. Then, the kernelized TD is incorporated into an actor-critic, natural policy gradient method, for which convergence analysis is derived. Finally, the paper reports a few experiments in simple continuous control domains to validate the theory and to compare the resulting algorithmic implementation with PPO.

**Strengths:**

Establishing the sample complexity of popular algorithms PPO in general function spaces is an interesting proposition. I am not familiar with prior work on this matter and with the function-space analysis reported in the paper, so it is hard for me to evaluate the strengths of the submission beyond noting that the problem is relevant.

**Weaknesses:**

- The paper may not be accessible for a general RL audience
- Some assumptions may be unreasonable in most settings (e.g., Asm 1 and 6)
- Practical implications of the reported results are not stated clearly
- The empirical comparison between PPO and the proposed algorithm may be flawed or comparing the baselines in mismatching settings

**Questions:**

**Disclaimer**: My current evaluation is essentially an educated guess of someone working in (theoretical) RL but not familiar with functional analysis. As I could not understand most of the submission, I am mostly focusing my review on the presentation of the results and how they could be made more accessible for a broader audience, while I am leaving a technical evaluation of the work to reviewer that are more familiar with the employed mathematical tools and prior works.

----

**Evaluation**: Assuming a solid technical contribution, I think the paper shall improve the presentation in a few aspects to clear the bar for acceptance. It shall discuss the practical implication of the reported theoretical results, it shall clarify how the analysis advance over prior works more directly (e.g., comparison of convergence rates in the same function space), it shall clarify why the reported results are important and valuable going forward.

In terms of motivation, I am not sure the paper aims at providing a theoretical support for new algorithmic solutions that advance over PPO and others or more of a theoretical understanding on when (in which domains/settings) is PPO expected to work/be efficient.

----

**Comments**

1. I am not sure why the title and abstract suggest that the paper provides an analysis of PPO, when the actual algorithm that is being analyzed looks quite different than PPO. Perhaps natural policy gradient would be more appropriate.

2. Assumption 1 and 6 look quite strong to me. Everything, evaluation/optimization, is easier when good data are available, but a crucial challenge of RL is to collect them from interaction with the MDP.

3. What does assumption 2 mean in more intuitive terms?

4. Not easy to understand what Theorem 9 and its corollary imply. I understand that the "training loss" decreases with a certain function of $n$, although the actual rate is not trivial to grasp. Is $\beta$ capturing a property of the function space or a parameter of the approach?

5. How do the results in Corollary 10 relate with previous works?

6. Why in eq. 21 the return of a policy is computed in expectation over the state distribution of an optimal policy, instead of the state distribution induced by the policy itself?

7. Experiments: If I understand it correctly, the comparison between PPO and the proposed algorithm is spurious. It looks like PPO is running with trajectories collected on-policy, while the proposed algorithm gets data from some sort of reset distribution. Learning from a "good" reset distribution is way easier than learning from actual rollouts. Can the authors clarify how the experiments are indeed comparing apples to apples?

typos and minor:
- l139: "and it is equipped this space with the bilinear form" something is off with the sentence construction
- l151: "We first impose the following assumption on the Markov chain". Which Markov chain?
- l235: "Derivethe"
- l260: "Unclear whether C_1 and C_2 are universal constants or coming from Asm.1"
- l372: misplaced dot
- l406: missing bracket
- l454: "learning Q" -> "to learn Q"

---

> ### Author Response · Authors · 2025-11-24
>
> **Summary:** We sincerely thank you for helping us correct typos and for providing valuable suggestions. Our initial goal targets researchers in kernel methods, statistical learning, and approximation/optimization theory. We show that TD and PPO can be formulated within an RKHS framework (via the fixed-point KRR equation), highlighting their close connections to kernel methods and non-parametric statistics. We believe that as more researchers recognize the connections between RL and their own fields, this will stimulate further theoretical research bridging RL with other disciplines, ultimately producing works that are more accessible to a broader audience.
>
> As other reviewers pointed out, we are the first work that provides an RKHS and kernel ridge regression framework for TD and PPO. Therefore, there is currently no other related work that can provide a comparison for our theoretical results. The studies most closely related to ours are Duan et al. (2024)[1] and  Liu et al. (2019)[2]. However, their setups differ from ours. Duan et al. (2024)[1] focus exclusively on Q-learning with a transition kernel that depends only on the state, while Liu et al. (2019)[2] consider the population setting, where the policy and Q-function are parametrized by a two-layer neural net and can be estimated from infinitely many samples. Therefore, their analysis is unrelated to sampling complexity.
>
> According to your suggestions, we have mentioned that our convergence rates are minmax optimal in terms of statistical estimation and have provided supporting references. We believe this further highlights the significance and contribution of our work. Additionally, **we have included analysis for more general TD learning (N-step TD and TD($\lambda$)), expanded the theoretical results for PPO, and added experiments on high-dimensional problems and baseline comparisons to validate our theories.**
>
>
> [1]Duan, Y., Wang, M., and Wainwright, M. J. (2024). Optimal policy evaluation using kernel-based temporal-difference methods. The Annals of Statistics.
>
> [2]Liu, B., Cai, Q., Yang, Z., and Wang, Z. (2019). Neural proximal/trust region policy optimization attains globally optimal policy. In Advances in Neural Information Processing Systems 32 (NeurIPS 2019).
>
>
>
> We now respond to the comments one by one.

---

> ### Author Response · Authors · 2025-11-24
>
> **Q1:** I am not sure why the title and abstract suggest that the paper provides an analysis of PPO, when the actual algorithm that is being analyzed looks quite different than PPO. Perhaps natural policy gradient would be more appropriate.
>
> **Answer**: For PPO, we refer to Algorithm 1 in the original paper: Schulman, John, et al. (2017)[1], where the surrogate loss can be either the clipped version or the one penalized by the KL divergence.  Liu et al. (2019)[2] select the latter one, and we follow their notations for our analysis.
>
> Following your suggestions, we revised the analysis for two versions of PPO with KL-penalized surrogate losses, corresponding to $KL[\pi\|\pi^{k}]$ and $KL[\pi^{k}\|\pi]$. These lead to two distinct update rules. The former, as you noted, is equivalent to the natural policy gradient (NPG) and yields a closed-form expression for $\pi^{k+1}$; we therefore refer to it as explicit PPO. The latter resembles an implicit stochastic gradient descent update, which we refer to as implicit PPO.
>
> We hope this revision provides a deeper understanding of PPO within our functional framework.
>
>
> [1] John Schulman, Filip Wolski, Prafulla Dhariwal, Alec Radford, and Oleg Klimov.(2017). Proximal policy optimization algorithms. arXiv preprint arXiv:1707.06347.
>
> [2] Liu, B., Cai, Q., Yang, Z., and Wang, Z. (2019). Neural proximal/trust region policy optimization attains globally optimal policy. In Advances in Neural Information Processing Systems 32 (NeurIPS 2019).
>
>
>
> **Q2:** Assumptions 1 and 6 look quite strong to me. Everything, evaluation/optimization, is easier when good data are available, but a crucial challenge of RL is to collect them from interaction with the MDP.
>
>
> **Answer**: We have relaxed Assumption 1, allowing the density of the optimal policy to be unbounded. Nevertheless, boundedness of the policy density remains essential to ensure that the generalization error for our TD in RKHS (Corollary 11), the explicit PPO update rule (Equation 19), and the implicit PPO update rule (Equation 132) are all mathematically well-defined. Without this boundedness, singularities in the density could lead to ill-behaved functionals, making rigorous analysis impossible. Moreover, we view this boundedness assumption as mild, since it naturally holds for standard model parameterizations used in practice and ensures problems involved in (stochastic) gradient-based optimization remain well-defined.
>
> For Assumption 6, even though it cannot be directly relaxed, we have added a new section in Appendix J to generalize our one-step TD to N-step TD. In the KRR formulation for N-step TD, Assumption 6 is replaced by Assumption 6B, where the constraint $c\gamma<1$ is relaxed to $c\gamma^{N}<1$. We view this as a mild assumption. In fact, compared with similar work—such as Duan et al. (2024)[1] and Liu et al. (2019)[2] mentioned above—Assumption 6 is relatively weak. Most existing convergence analyses in reinforcement learning require data to be sampled directly from the stationary distribution of the MDP, which corresponds to the special case c=1 in our framework. In contrast, Assumption 6 only requires that the initial sampling distribution is not too far from the stationary distribution, making it substantially less restrictive.
>
> [1] Duan, Y., Wang, M., and Wainwright, M. J. (2024). Optimal policy evaluation using kernel-based temporal-difference methods. The Annals of Statistics.
>
> [2] Liu, B., Cai, Q., Yang, Z., and Wang, Z. (2019). Neural proximal/trust region policy optimization attains globally optimal policy. In Advances in Neural Information Processing Systems 32 (NeurIPS 2019).
>
>
> **Q3:** What does assumption 2 mean in more intuitive terms?
>
> **Answer**: Assumption 2 serves as an alternative to the more traditional requirement that the Q-function belongs to a predetermined function space (an RKHS) with a bounded functional norm. This type of assumption is standard in functional analysis, as one must first specify the space containing the target function. However, as noted in our latest revision, direct sampling of the Q-function is expensive. Therefore, instead of assuming boundedness of the Q-function itself, we assume that the reward function r and the transition kernel $P(s^{'}|s,a)$ each have bounded RKHS norms. Under this formulation, the RKHS norm of the resulting Q-function is determined solely by the policy $\pi$.

---

> ### Author Response · Authors · 2025-11-24
>
> **Q4:** Not easy to understand what Theorem 9 and its corollary imply. I understand that the "training loss" decreases with a certain function of $n$, although the actual rate is not trivial to grasp. Is $\beta$ capturing a property of the function space or a parameter of the approach?
>
> **Answer**: In the revised version, Proposition 9 and the related result, Theorem 10, provide standard convergence-rate analyses for nonparametric statistical problems. Proposition 9 characterizes the classical bias–variance tradeoff by decomposing the error into two components: the statistical error arising from the Bellman residual and the approximation error introduced by the penalty parameter $\lambda$. Theorem 10 presents a standard convergence result under the empirical $L_2$ penalty, corresponding to the error evaluated on the dataset (often referred to as the training error). These results rely on well-established analytical techniques for nonparametric problems based on empirical process theory. We have also mentioned this point in our paper, and interested readers may refer to the textbook van de Geer (2000)[1] for further details.
>
> [1]Sara A Geer(2000). Empirical Processes in M-estimation, volume 6. Cambridge university press.
>
>
> **Q5:**   How do the results in Corollary 10 relate with previous works?
>
> **Answer**: In the revised version, Corollary 10 is changed to Corollary 11. To the best of our knowledge, there is no prior work related to this result. As other reviewers have noted, our paper is the first to place TD and PPO within the KRR framework, and there is no existing work that studies convergence rates for TD under this nonparametric setting.
>
> On the other hand, we are sure that the convergence rates in Corollary 11 are minimax optimal for the underlying statistical estimation problem. This follows from well-established theoretical results in general nonparametric statistics, which we also discuss in the paper. For further details, interested readers may refer to the textbook Nemirovski (2000)[1].
>
> [1]Arkadi Nemirovski(2000). Topics in Nonparametric Statistics, École d’Été de Probabilités de Saint-Flour, 28:85.
>
> **Q6:** Why in eq. 21 the return of a policy is computed in expectation over the state distribution of an optimal policy, instead of the state distribution induced by the policy itself?
>
> **Answer**: This is a reasonable and widely used metric for proximal policy optimization, and it has been adopted in many well-known studies. For example, the following highly-cited papers use this metric: Wang et al(2019).[1], Liu et al.(2019)[2], Cen et al.(2022)[3], Lan (2023)[4], and Duan et al.(2024)[5].
>
> To further motivate our choice of taking expectations over the state distribution of the optimal policy, we quote Lan (2023)[3]: "Even though the definition of the objective depends on $\nu^{\*}$ and thus on the unknown optimal policy $\pi^{\*}$, the algorithms … do not actually require $\pi^{\*}$ as an input."
>
>
> [1]Lingxiao Wang, Qi Cai, Zhuoran Yang, Zhaoran Wang (2019) NEURAL POLICY GRADIENT METHODS: GLOBAL OPTIMALITY AND RATES OF CONVERGENCE. In International Conference on Learning Representations 2019 (ICLR 2019).
>
> [2]Liu, B., Cai, Q., Yang, Z., and Wang, Z. (2019). Neural proximal/trust region policy optimization attains globally optimal policy. In Advances in Neural Information Processing Systems 32 (NeurIPS 2019).
>
> [3]Cen, S., Cheng, C., Chen, Y., Wei, Y. and Chi, Y.(2022). Fast global convergence of natural policy gradient methods with entropy regularization. Operations Research 70.4 (2022): 2563-2578.
>
> [4]Lan, Guanghui(2023).Policy mirror descent for reinforcement learning: Linear convergence, new sampling complexity, and generalized problem classes. Mathematical programming 198.1 (2023): 1059-1106.
>
> [5]Duan, Y., Wang, M., and Wainwright, M. J. (2024). Optimal policy evaluation using kernel-based temporal-difference methods. The Annals of Statistics.

---

> ### Author Response · Authors · 2025-11-24
>
> **Q7:** Experiments: If I understand it correctly, the comparison between PPO and the proposed algorithm is spurious. It looks like PPO is running with trajectories collected on-policy, while the proposed algorithm gets data from some sort of reset distribution. Learning from a "good" reset distribution is way easier than learning from actual rollouts. Can the authors clarify how the experiments are indeed comparing apples to apples?
>
> **Answer**: We thank the reviewer for this question and would like to clarify that the comparison is indeed apples-to-apples. In our experiments, both PPO and the proposed NPG algorithm start every episode from the exact same standard environment initialization (defined by the Gym environment using `env.reset()`); thus, neither algorithm has access to a 'good' pre-computed reset distribution. Both algorithms collect data by interacting with the environment on-policy. The theoretical formulation of sampling $(s_0, a_0, s_1, a_1)$ from $\mu_0$ (Eq. 5) is implemented practically by treating the states visited along these trajectories as the samples. Therefore, the data source is identical for both methods: PPO uses the trajectory points to update the policy directly, while our NPG uses the exact same trajectory points—viewed as samples from $\mu_0$—to perform TD learning and policy updates.
>
> Moreover, compared to the related work, our setting strictly relaxes the assumptions. The stronger stationarity assumptions used in existing kernel-RL and PPO analyses, e.g. Duan et al. (2024)[1] and Liu et al. (2019)[2]. Duan et al. (2024)[1]  assumes samples from the stationary distribution $\mu$ and Liu et al. (2019)[2]'s neural PPO/TRPO analysis uses the stationary state(-action) distribution $\sigma_{\pi_k}$ at each iteration. Instead, we only require that one-step transitions from $\mu_0$ are not too far from $\mu_0$ itself. Thus, in practice experiments, a good $\mu_0$ is easy to construct.
>
> [1]Duan, Y., Wang, M., and Wainwright, M. J. (2024). Optimal policy evaluation using kernel-based temporal-difference methods. The Annals of Statistics.
>
> [2]Liu, B., Cai, Q., Yang, Z., and Wang, Z. (2019). Neural proximal/trust region policy optimization attains globally optimal policy. In Advances in Neural Information Processing Systems 32 (NeurIPS 2019).
>
>
> **typos and minor:**
>
> **Answer**: Thank you for pointing this out. We have carefully corrected all typos in the revised version.

---

> > ### Comment · Reviewer_4UFk · 2025-11-27
> >
> > Dear authors,
> >
> > Thank you for your extensive replies.
> >
> > I see that I am not the intended audience for this work and other reviewers seem to have a more positive stance. I am planning to keep my score, as I am not in the position to provide a robust evaluation. I will clarify in the private discussion that I do not oppose accepting the paper.
> >
> > I think that an additional effort to make the contribution palatable to a generalist RL theory audience could be made. A few minor notes:
> > - Even though this is the first work on "RKHS and kernel ridge regression for TD and PPO", I think we can still compare the result with convergence rate and error propagations of other settings, e.g., how do the results translate to the tabular setting?
> > - On the Assumption 1:" Moreover, we view this boundedness assumption as mild, since it naturally holds for standard model parameterizations used in practice and ensures problems involved in (stochastic) gradient-based optimization remain well-defined". Why is that the case?
> >
> > Note that the above have little impact on my evaluation and can be ignored.
> >
> > Best,
> >
> > Reviewer 4UFk

---

> > > ### Author Response · Authors · 2025-11-27
> > >
> > > Dear reviewer,
> > >
> > > We sincerely thanks for your further suggestions.
> > >
> > > 1. We have further revised the manuscript to emphasize the convergence rate for the tabular case and to include additional comparisons with existing results in the appendix. Our convergence rate is also optimal up to a logarithmic factor, and we have explicitly compared it with the well-known results in [1], [2], and [3]. We also clarify the origin of the additional logarithmic term in our bound, which arises from the technical tools employed in our proof—tools that are particularly suitable for handling non-parametric settings.
> > >
> > > 2. Because an unbounded distribution can lead to ill-behaved estimates and singularities in stochastic optimization, additional assumptions are needed to ensure well-posedness. For example, consider the PPO update in Equation (19). If the underlying distribution is unbounded, the closed-form solution in Equation (20) cannot be rigorously defined due to singularities in the density. Although this issue does not affect practical applications, the boundedness assumption is required to establish our theoretical guarantees.
> > >
> > > [1] Eyal Even-Dar and Yishay Mansour. Learning rates for q-learning. JMLR, 5(Dec): 1–25, 2003.
> > >
> > > [2] Wainwright MJ (2019b) Stochastic approximation with cone-contractive operators: Sharp ℓ∞-bounds for Q-learning. Preprint,submitted May 15, https://arxiv.org/abs/1905.06265.
> > >
> > > [3] Gen Li, Changxiao Cai, Yuxin Chen, Yuting Wei, and Yuejie Chi. Is q-learning minimax optimal? a tight sample complexity analysis. OperationsResearch, 72(1):222–236,2024.
> > >
> > > Best,
> > >
> > > Author

---

### Official Review · Reviewer_L5Ur · 2025-10-30

**Soundness:** 3
**Presentation:** 3
**Contribution:** 3
**Rating:** 6
**Confidence:** 3

**Summary:**

This work studies an actor-critic method akin to TRPO/PPO in an RKHS, which integrates TD learning in a kernelized way as critic with a KL-regularized natural policy gradient update. It is related to a kernelized version of natural actor-critic method. On the technical side, the paper establishes sample complexity bounds need per policy optimization step to achieve a target error.

**Strengths:**

- The per-step sample complexity bounds in the paper are definitely interesting and important.
- The analysis of kernelized TD critic is clean and rigorous. The connection to the metric entropy in the RKHS provides a unified perspective, which is valuable.
- The approach to connect TD learning with NPG-PPO-style policy improvement updates is classic yet elegant.

**Weaknesses:**

- Assumption 6 is a little unusual as it controls the state transitions per step. The situation worsens for weakly-ergodic MDPs with poor mixing times. It can be worth discussing further justifications, and benign and pathological extremal cases.
- The dependency on the effective time-horizon $1-\gamma$ is a little obscure. In the bounds, does effective time-horizon appear in terms of $1-c\gamma$, or are there any hidden multiplicative factors of $(1-\gamma)^{-1}$?

**Questions:**

- Does the analysis extend to multi-step TD or TD($\lambda$) for $\lambda \in (0,1]$ with eligibility traces?
- A key component of (natural) policy gradient algorithms is exploration. This is usually established by assuming finite concentrability coefficients, or using entropic regularization with a fixed regularization coefficient. In this paper, how is sufficient exploration guaranteed to avoid strictly suboptimal and near-deterministic policies? The role of KL-regularization from an exploration perspective could be discussed further.
- In conjunction with the previous point, can the analysis here be extended to entropic regularization?

---

> ### Author Response · Authors · 2025-11-24
>
> **Summary:** We sincerely thank Reviewer L5Ur for their constructive feedback and for finding our analysis clean and rigorous. We are glad that the reviewer appreciated the importance of our per-step sample complexity bounds and the unified perspective provided by our connection to metric entropy. We value the reviewer's detailed technical questions regarding the justification of Assumption 6, the dependencies on the effective time-horizon, and the extensions to multi-step TD and entropic regularization. These questions have helped us clarify the technical depth of our work, and we have updated the manuscript to add **N-step TD** and **TD($\lambda$) analysis**, **two variants of PPO** (which we refer to as explicit and implicit PPO), and the **corresponding experiments**. Below, we address each point in detail.
>
>
> **W1:** On Assumption 6
>
>
> **Answer:**
> We have relaxed our Assumption 6, which now states that for the sampling distribution $\mu_0$ and some $c<1/\gamma$, $$ \int P\left(s^{\prime} \mid s, a\right) \pi(a \mid s) \mu_0(s) d s d a \leq c \mu_0\left(s^{\prime}\right),$$
> This means after one step of the Markov chain started from $\mu_0$, the marginal distribution of the next state is absolutely continuous w.r.t. $\mu_0$ with density ratio at most $c$. Thus $c$ measures how non-stationary the data is relative to $\mu_0$. If we choose $\mu_0$ to be the stationary state distribution $\nu_\pi$ under policy $\pi$, then the chain is stationary and
> $$
> \int P\left(s^{\prime} \mid s, a\right) \pi(a \mid s) \nu_\pi(s) d s d a=\nu_\pi\left(s^{\prime}\right)
> $$
> so Assumption 6 holds with $c=1$. This recovers the stronger stationarity assumptions used in existing kernel-RL and PPO analyses, e.g. Duan et al. (2024) and Liu et al. (2019)[2]. Duan et al. (2024) [1] assumes samples from the stationary distribution $\mu$ and Liu et al. (2019)'s neural PPO/TRPO analysis uses the stationary state(-action) distribution $\sigma_{\pi_k}$ at each iteration.
> Our setting strictly relaxes these works: we only require that one-step transitions from $\mu_0$ are not too far from $\mu_0$ itself. When the chain mixes quickly from $\mu_0$, the one-step marginal is close to stationary and $c$ is close to 1. For slowly mixing or weakly-ergodic chains, $c$ can be larger; our rates degrade gracefully in this case (see below).
>
> We have clarified this interpretation in Remark 7 of the revised draft and now explicitly note that stationary sampling corresponds to $c=1$.
>
> [1]Duan, Y., Wang, M., and Wainwright, M. J. (2024). Optimal policy evaluation using kernel-based temporaldifference methods. The Annals of Statistics.
>
> [2]Liu, B., Cai, Q., Yang, Z., and Wang, Z. (2019). Neural proximal/trust region policy optimization attains globally optimal policy. In Advances in Neural Information Processing Systems 32 (NeurIPS 2019).
>
>
>
> **W2:** On Effective Time-horizon
>
> In all our main results the dependence on the time horizon enters only via the factor $1-c \gamma$: The critic bound (Theorem 10) scales as
> $$
> \left\||f_t-Q^\pi\right\|| \lesssim(1-c \gamma)^{-\frac{2+\beta}{2+2 \beta}} n^{-\frac{1}{2+2 \beta}}
> $$
> up to logs (equation (14)), and the specialization in Corollary 11 yields factors like $(1-c \gamma)^{-1}$, $(1- c \gamma)^{-(2 m+d / 2) /(2 m+d)}$, etc.
> The per-iteration sample schedule $n^{(k)}$ and $\lambda^{(k)}$ in Table 1 and in equations (18)/(23) carry the same dependence through $1-c \gamma$.
> Thus our analysis can be interpreted as using a generalized effective horizon
> $$
> H_{\mathrm{eff}}=\frac{1}{1-c \gamma},
> $$
> which simultaneously captures the discount factor and the discrepancy between the one-step marginal and the sampling distribution.
> There are no additional hidden multiplicative $(1-\gamma)^{-1}$ factors beyond replacing the usual ( $1-\gamma$ ) by $(1-c \gamma)$. In the stationary case $(c=1)$ this collapses to the standard "effective horizon" $H=(1- \gamma)^{-1}$, matching the dependence in kernel LSTD.
> We have emphasized this discussion in the revised paper (end of Sec. 4.1 and Remark 7).

---

> ### Author Response · Authors · 2025-11-24
>
> **Q1:** Can the analysis extend to multi-step TD or TD($\lambda$)?
>
> **Answer:**
> Yes. In the revision, we have added Appendix J, which formally treats $N$-step TD. We define the $N$-step Bellman equation
> $$
> Q^\pi(s, a)=\mathbb{E}\left[\sum_{t=0}^{N-1} \gamma^t r_t \mid s_0=s, a_0=a\right]+\gamma^N \mathbb{E}\left[Q^\pi\left(s_N, a_N\right) \mid s_0=s, a_0=a\right],
> $$
> and show it is equivalent to the Bellman equation of a modified MDP with discount $\gamma^N$, transition kernel $P\left(s_N \mid s_0, a_0\right)$, and aggregated reward.
> Assumption 6 is relaxed to Assumption 6B, which only requires
> $$
> P\left(s_N=s\right) \leq c \mu_0(s), \quad c \gamma^N<1,
> $$
> so all occurrences of $c \gamma$ in Theorem 10, Corollary 11, and Corollary 14 are replaced by $c \gamma^N$. This yields faster convergence and improved sample complexity because $c \gamma^N \ll c \gamma$ when $N$ exceeds the mixing time.
>
> For $\operatorname{TD}(\lambda)$ with eligibility traces, note that the $\lambda$-return can be written as a geometrically weighted mixture of N-step returns:
> $$
> G^\lambda=(1-\lambda) \sum_{N \geq 1} \lambda^{N-1} G^{(N)} .
> $$
> At the population level this corresponds to a $\gamma_\lambda=\frac{\gamma(1-\lambda)}{1-\lambda \gamma}$-contractive Bellman operator $T_\lambda$. The techniques we use (kernel gradient descent on an RKHS Bellman-type operator plus metric-entropy control) apply verbatim to this operator; the main change would be that $c \gamma$ is replaced by $c \gamma_\lambda$ in the bounds. We have added a subsection (Appendix J.4 in the revised version) to explain how the same argument as $N$-step TD extends to $\operatorname{TD}(\lambda)$. We have also included the corresponding experiments of the extension to $N$-step TD learning in the new Section 6.4.
>
>
> **Q2:** How is sufficient exploration guaranteed, and how does KL-regularization help avoid near-deterministic suboptimal policies?
>
> **Answer:**
>
> Our actor is parameterized as an energy-based policy $\pi_k (a \mid s) \propto \exp \{[F_k(s, a)\]}$ with $F_k \in \mathcal{H}$. Hence, for every $k$ and state $s, \pi_k(\cdot \mid s)$ has full support on the action space. This prevents complete collapse to a deterministic policy; all actions retain strictly positive probability.
> Moreover, the PPO-style update
> $$
> \pi_{k+1}(\cdot \mid s) \propto \pi_k(\cdot \mid s) \exp \{[ \Delta_k A_k (s, \cdot) \]}
> $$
> is a mirror-descent step with KL divergence, which has been shown to inherently explores action space [1].
>
> [1]Li, Yan, and Guanghui Lan. "Policy mirror descent inherently explores action space." SIAM Journal on Optimization 35.1 (2025): 116-156.
>
>
> **Q3:** Can the analysis be extended to entropic regularization?
>
> **Answer:**
>
> Our policy update is analyzed through the lens of mirror descent with the negative Shannon entropy as the mirror map and KL as the Bregman divergence. The proofs only rely on generic properties of this mirror map (strong convexity, three-point identity), not on the specific choice of reference policy $\pi_k$.
> Hence, we believe that our framework can be extended to general mirror descent methods, including those with entropy regularization. In Appendix K, we introduce a variant of PPO—referred to as implicit PPO—which provides partial evidence that our framework can be generalized and is more closely aligned with our analysis than entropy regularization. However, due to space limitations, a full treatment of the extension to mirror descent is left for future work.

---

### Official Review · Reviewer_eHMp · 2025-10-30

**Soundness:** 3
**Presentation:** 4
**Contribution:** 4
**Rating:** 8
**Confidence:** 3

**Summary:**

This paper revisits Proximal Policy Optimization (PPO) from a function-space perspective by introducing a kernelized temporal-difference (TD) method for policy evaluation and a KL-regularized natural-gradient method for policy improvement, both within a Reproducing Kernel Hilbert Space (RKHS). The authors provide non-asymptotic error bounds and sample complexity analysis, showing that the proposed framework improves stability and sample efficiency in standard control tasks (such as CartPole and Acrobot). The results suggest that this method places PPO on a more solid theoretical foundation, extending it beyond finite-dimensional assumptions and clarifying when kernelized updates with TD critics yield global policy improvement with practical efficiency.

**Strengths:**

* *Theoretical Innovation*: The paper makes a significant contribution by combining kernel methods with reinforcement learning, particularly through the use of kernelized TD for policy evaluation and KL-regularized natural gradient updates for policy improvement. This approach successfully addresses theoretical gaps in PPO and similar algorithms when dealing with complex, non-linear function approximators.
* *Comprehensive Analysis*: The paper offers a detailed analysis of sample complexity and convergence rates, providing a unified framework for understanding PPO's behavior in RKHS. The analysis is not limited to simple models but extends to Sobolev spaces, NTK, and deep neural networks, offering theoretical guarantees for various function spaces.
* *Empirical Validation*: The experimental results strongly support the theoretical analysis, demonstrating that the proposed method outperforms the standard PPO algorithm in terms of sample efficiency and computational efficiency. The experiments on benchmark control tasks such as CartPole and Acrobot show that the theoretical predictions align well with empirical performance.
* *Practical Relevance*: The paper provides not only theoretical insights but also practical guidance on how to improve PPO and similar reinforcement learning algorithms. The proposed methods are directly applicable to large-scale reinforcement learning tasks that require efficient sample usage and computational performance.

**Weaknesses:**

* *Assumption of Bounded Distributions*: One of the key assumptions in the paper is that the distribution $\sigma(s,a)$ is bounded. While this assumption simplifies the theoretical analysis, it may not hold in all real-world scenarios, especially when dealing with highly variable or concentrated distributions. Further discussion on how the algorithm behaves under weaker or less restrictive assumptions would be valuable.
* *Computational Complexity Analysis*: While the paper provides a thorough theoretical analysis, it lacks an in-depth discussion on the computational complexity of the proposed method, especially for high-dimensional problems. As the state and action spaces grow, it would be useful to analyze how the algorithm scales in terms of computation and memory.
* *Limited Experimental Environment*: The experiments are conducted on relatively simple control tasks (CartPole and Acrobot). While these tasks are useful for validating the method, it would be helpful to see how the proposed method performs on more complex, high-dimensional environments to better assess its scalability and robustness.

**Questions:**

* *Relaxing the Assumption on Bounded*: The assumption that $\sigma$ is bounded might not hold in all real-world applications. Have you considered how the algorithm performs if $\sigma$ is not bounded or exhibits more variability? Are there methods to ensure the algorithm's stability and convergence even when this assumption is relaxed?
* *Computational Efficiency and Scalability*: In the experiments, you demonstrated superior performance on standard tasks. How does the algorithm scale with respect to both computation and memory as the size of the state and action spaces increases? Are there strategies to further optimize the method for large-scale reinforcement learning problems?
* *Nonlinear Approximators and Deep Networks*: The paper discusses the applicability of kernelized TD and RKHS methods for non-linear function approximators. Do you think this approach can be extended to more complex architectures such as deep neural networks? What are the practical challenges of applying this method to deep reinforcement learning algorithms?

---

> ### Author Response · Authors · 2025-11-24
>
> **Summary:** Thank you very much for your positive assessment! We have relaxed assumptions, added new environments in experiments, and report the runtime and memory to discuss computational complexity.
>
> Below, we address each point in detail.
>
> **W1&Q1**: Assumption of Bounded Distributions & Relaxing the Assumption on Boundedness
>
> **Answer**: We have relaxed Assumption 1 in the revised manuscript. The previous version required boundedness of the stationary state–action distribution $\sigma^{*}$, whereas the new Assumption 1 only requires boundedness of the on-policy sampling distribution $\sigma_0^\pi$. This relaxation permits the density induced by the optimal policy to be unbounded. Nevertheless, some boundedness remains essential to ensure that key quantities in our analysis are mathematically well-defined. In particular, boundedness is needed for the population-level generalization bound for kernel TD (Corollary 11), as well as for the explicit PPO update (Eq. 19) and the implicit PPO update (Eq. 132). Without it, density singularities could lead to ill-posed functionals and render rigorous analysis impossible.
>
> Importantly, Assumption 1 does not affect the empirical TD error in Theorem 10, which does not require a boundedness condition and depends only on finite samples and the RKHS structure. Moreover, in practical reinforcement-learning settings, the sampling distribution generated by finite trajectories is inherently bounded over the underlying state-action space. Thus, relaxing Assumption 1 does not affect the empirical algorithm, and as demonstrated in our experiments, the method remains stable and performs well even under substantial variability.
>
> **W2&Q2**: Computational Efficiency and Scalability
>
> **Answer**: Thank you for commenting on this. We have added a new Section L.2 in the Appendix to show the run time and memory analysis. Our results in the new Table 1 (Cartploe and Acrobot) and Table 2 (HalfCheetah-v5) demonstrate that the proposed NPG algorithm remains highly practical even as problem size increases.
>
> On Apple M4, for standard discrete tasks like CartPole-v1 and Acrobot-v1, the average runtime of 1,000 episodes is extremely low (often under 1 minute). For the high-dimensional HalfCheetah-v5 task (state dimension 17, action dimension 6), the algorithm scales gracefully. Despite the significant increase in state-action space complexity and episode length (1,000 steps), the runtime remains manageable, averaging about 800 to 900 seconds for 10,000 episodes.
> Memory usage remains stable and relatively low across all environments, confirming the practical feasibility of our approach.
> As our NPG does not require the ordered trajectory sampling, the running time can be made even smaller with parallelization.
>
> **Table 1: Performance comparison on Cartpole-v1 and Acrobot-v1.**
>
> | Environment | Cartpole-v1 |  |  | Acrobot-v1 |  |  |
> | :--- | :---: | :---: | :---: | :---: | :---: | :---: |
> | Step size | $k^{-0.2}$ |$k^{-0.5}$ | $k^{-1.5}$ | $k^{-0.2}$ | $k^{-0.5}$ | $k^{-1.5}$ |
> | Running time (s) | $30.26 \pm 4.49$ | $49.04 \pm 7.59$ | $7.85 \pm 2.42$ | $80.82 \pm 20.12$ | $49.77 \pm 25.28$ | $96.07 \pm 0.54$ |
> | Peak Memory cost (MB) | $282.06 \pm 6.79$ | $272.77 \pm 0.10$ | $273.00 \pm 0.04$ | $194.8 \pm 60.62$ | $230.04 \pm 0.17$ | $230.52 \pm 0.16$ |
>
> **Table 2: Performance comparison on HalfCheetah-v5.**
>
> | Step size | $k^{-0.1}$ | $k^{-0.5}$ | $k^{-1.5}$ |
> | :--- | :---: | :---: | :---: |
> | Running time (s) | $834.23 \pm 18.56$ | $911.84 \pm 48.86$ | $917.94 \pm 72.95$ |
> | Peak Memory cost (MB) | $202.65 \pm 70.77$ | $116.47 \pm 8.26$ | $160.79 \pm 58.42$ |
>
> Furthermore, to demonstrate scalability and optimization strategies, we introduced **$N$-Step TD Learning** in Section 6.4 and Appendix Section J. The one-step TD method can be naturally extended to $N$-step TD by viewing the $N$-step dynamics of the original MDP as the one-step transition dynamics of an enlarged MDP. The additional sampling steps reduce sampling variance and lead to faster convergence.
>
> **W3**: Limited Experimental Environment
>
> **Answer**: We agree with the reviewer and have **added the HalfCheetah-v5 environment** to our test suite to validate performance on complex, high-dimensional tasks. As shown in the updated Figure 1-c, our method successfully solves the HalfCheetah task, exhibiting a consistent convergence pattern similar to that seen in lower-dimensional tasks (CartPole and Acrobot) when using the optimal step size schedule $\Delta_k=k^{-0.5}$.

---

> ### Author Response · Authors · 2025-11-24
>
> **Q3**: Nonlinear Approximators and Deep Networks
>
> **Answer**: We have derived the sample complexity of the Neural Tangent Kernel (NTK) in our analysis. According to Jacot et al. (2018)[1], neural networks trained with early-stopped gradient descent are equivalent to kernel ridge regression (KRR), where the regularization parameter $\lambda$ in KRR determines the stopping criterion of gradient descent. From Theorem 9, it is not necessary to know the explicit form of the kernel, as long as the metric entropies of the RKHS are known; metric entropy of general neural networks has been well-studied in Anthony (2009)[2]. Consequently, our framework can be generalized to deep neural networks, and all experiments are, indeed, conducted using neural networks under our framework with RKHSs induced by the NTK.
>
> [1]Jacot, Arthur, Franck Gabriel, and Clément Hongler. "Neural tangent kernel: Convergence and generalization in neural networks." Advances in neural information processing systems 31 (2018).
>
> [2]Anthony, Martin, and Peter L. Bartlett. Neural network learning: Theoretical foundations. cambridge university press, 2009.

---

> > ### Comment · Reviewer_eHMp · 2025-11-27
> >
> > I appreciate the authors for their rich and professional response. I would like to maintain my score as accept.

---

### Official Review · Reviewer_4sex · 2025-11-11

**Soundness:** 3
**Presentation:** 3
**Contribution:** 3
**Rating:** 6
**Confidence:** 2

**Summary:**

This paper provides a theoretical and algorithmic study of Temporal-Difference (TD) learning and Natural Policy Gradient (NPG)/PPO methods in Reproducing Kernel Hilbert Spaces (RKHS). The work unifies prior results across tabular, linear, Sobolev, Gaussian, and Neural Tangent Kernel (NTK) regimes, with empirical validation on standard control benchmarks

**Strengths:**

1. Provides the first comprehensive analysis of TD and PPO/NPG in infinite-dimensional RKHS, bridging classical tabular/linear theory with modern kernel and neural tangent kernel regimes.

2. Introduces an efficient kernel TD evaluator that acts as an implicit preconditioner, achieving geometric convergence without cubic matrix inversion, thereby improving scalability in continuous spaces.

3.  The results elegantly connect function-space policy optimization to kernelized and neural-network representations, offering a unified view of convergence and sample complexity across different model classes.

4.  Experiments on control benchmarks (CartPole, Acrobot) support the theoretical predictions and show improved stability and efficiency compared with PPO baselines.

**Weaknesses:**

1) Limited Experimental Scope: The empirical evaluation is restricted to low-dimensional toy environments, leaving scalability to high-dimensional or real-world continuous control tasks (e.g., MuJoCo, DMControl) untested.

2. Strong Theoretical Assumptions: The analysis relies on idealized assumptions such as bounded RKHS norms, smooth reward functions, and known transition kernels—conditions rarely satisfied in practical RL settings.

3. Sampling Distribution Dependence: The framework presumes access to controllable sampling distributions and well-behaved state–action densities (Assumptions 1 & 6), which may not hold for off-policy or non-stationary data.

4. Implicit Domain Knowledge: While the paper claims generality, the theoretical derivation implicitly assumes that the underlying domain shift or policy complexity can be quantified via RKHS norms, which may not be observable in real applications.

5. Lack of Comparative Baselines: No direct comparison with recent kernel-based RL methods or strong deep-RL algorithms (e.g., SAC, TRPO, DDPG) limits the practical interpretability of the reported advantages.

6. Algorithmic Practicality: Although the authors claim computational efficiency, the per-iteration kernel operations could still be costly for large datasets, and no analysis of memory or runtime scaling is provided.

**Questions:**

N/A

---

> ### Author Response · Authors · 2025-11-24
>
> **Summary:** Thank you very much for your detailed review and for acknowledging the novelty of our analysis. To address the weakness, below, we clarify/relax the assumptions, add new experiments, and report the runtime and memory.
>
> **Q1:** Limited Experimental Scope
>
> **Answer:** Following your suggestion, we have added one high-dimensional environment,  HalfCheetah-v5, from MuJoCo. This environment has a **17-dimensional state space and 6-dimensional action space**. The new results in Figure 1-c show that in this complex and continuous setting, our method still shows the same convergence pattern as the low-dimensional tasks (CartPole and Acrobot), supporting the scalability of our proposed algorithm.
>
> **Q2:** Strong Theoretical Assumption
>
> **Answer:**  We would like to clarify that our analysis does not assume smooth reward functions, nor does it require access to the transition kernel. The bounded RKHS norm is a standard and mild regularity condition in nonparametric analysis, serving as a general complexity control rather than a model-specific requirement. Crucially, this assumption is weaker than smoothness: certain RKHSs contain non-differentiable functions, such as those induced by the Neural Tangent Kernel. Moreover, the RKHS norm can be estimated directly from data, making the assumption practical in applications. Although alternative, potentially more efficient complexity measures may exist, their investigation lies beyond the scope of this work and is deferred to future research.
>
> **Q3:** Sampling Distribution Dependence (Assumptions 1 & 6)
>
> **Answer:**  Following your suggestions, we have **weakened Assumption 1**. It no longer requires boundedness of the stationary distribution with respect to the optimal policy, but only assumes a bounded sampling distribution: $c \le \sigma_0^\pi(s,a) \le C$. Such boundedness is standard for deriving the generalization error $\|| Q^{\pi} - f^{(k)} \|| _ {L _ 2(\sigma)}$ from the the empirical error $\||Q^\pi-f^{(k)}\||_{n}$. The current complexity analysis relies on the sampling inequality (Eq.(79)), which holds when the sampling density is bounded. An unbounded distribution may make meaningful generalization impossible: Without this condition, meaningful generalization may fail: certain regions could be overweighted or never sampled, leading to $\|| g \|| _ {n^{2}} \not \to \|| g \|| _ {L_2(\sigma_0^{\pi})}^2$. Consequently, the empirical error bound in Theorem 10 would no longer imply the population error bound in Corollary 11.
>
> As for Assumption 6, our assumption is in fact **weaker than most existing works**, which typically assume that samples are drawn from the stationary policy. For example, Duan et al.(2024)[1] and Liu et al.(2019)[2] explicitly require access to the stationary state (-action) distribution. By contrast, Assumption 6 in our work is introduced to impose only a mild one-step stability condition on the induced on-policy sampling distribution, rather than assuming access to the stationary distribution. Specifically, it requires that one-step on-policy transitions do not move the initial distribution $\mu_0$ too far, which is a relatively weak condition.
> Furthermore, following the suggestion of reviewer L5Ur, we have **added the kernelized $N$-step TD method** to Appendix J. Under the $N$-step TD setting, the original condition $c\gamma < 1$ is relaxed to $c\gamma^{N} < 1$. Extending this framework to fully off-policy, non-stationary, or episodic settings would necessitate a more general operator, which lies beyond the scope of the present analysis and is left for future work.
>
> [1]Duan, Y., Wang, M., and Wainwright, M. J. (2024). Optimal policy evaluation using kernel-based temporal difference methods. The Annals of Statistics.
>
> [2] Liu, B., Cai, Q., Yang, Z., and Wang, Z. (2019). Neural proximal/trust region policy optimization attains globally optimal policy. In Advances in Neural Information Processing Systems 32 (NeurIPS 2019).
>
>
> **Q4:** Implicit Domain Knowledge
>
> **Answer:** We would like to clarify that our theory does not rely on any unobservable domain knowledge. Although the bounds are expressed in terms of RKHS norms, these norms are not assumed to be known; they can be directly estimated from data. In our framework, the policy enters only through its log-density $F_k = \sum_{j=0}^{k} \Delta_j f_T^{(j)}$, where each $f_T^{(j)}$ produced by kernel TD lies in the empirical RKHS $\hat{\mathcal H}=\mathrm{span} \[K(\omega_i, \cdot) \] _ {i=1}^{N}$. Therefore
> $F_k$ admits the finite expansion $F_k(\cdot)=\sum_{i=1}^{N} c_i K(\omega_i,\cdot)$, and its RKHS norm has the
> explicit empirical form $\||F_k\||_{\hat{\mathcal H}}^2 = F_k(\omega)^\top K(\omega,\omega)^{-1}F_k(\omega)$,
> which is empirically estimable. By the representer theorem, the complexity measure used in our theory is entirely data-driven rather than prior domain knowledge.

---

> ### Author Response · Authors · 2025-11-24
>
> **Q5**. Lack of Comparative Baselines: No direct comparison with recent kernel-based RL methods or strong deep-RL algorithms (e.g., SAC, TRPO, DDPG) limits the practical interpretability of the reported advantages.
>
> **Answer:** Thanks for pointing this out. We have added new widely-used RL baselines. We take Cartpole as an example to show the comparison results in Appendix L.3 Figure 4. For discrete action spaces, we compare our NPG algorithm with Discrete SAC[1], DQN[2], clip-PPO[3], and implicit PPO (see details in Appendix Section K).
> Our results show that implicit PPO achieves performance comparable to NPG and converges to the optimum more quickly. This is because the implicit update yields greater stability in policy iteration. In contrast, clip-PPO is less efficient and more unstable, as its reliance on GAE requires complete trajectories, whereas NPG learns directly from one-step TD errors. Discrete SAC failed to reach the optimal solution in this setting, and DQN, while efficient, exhibited unstable performance across random seeds.
>
> [1] Haarnoja, Tuomas, et al. "Soft actor-critic: Off-policy maximum entropy deep reinforcement learning with a stochastic actor." International conference on machine learning. Pmlr, 2018.
>
> [2] Mnih, Volodymyr, et al. "Human-level control through deep reinforcement learning." nature 518.7540 (2015): 529-533.
>
> [3] Schulman, John, et al. "Proximal policy optimization algorithms." arXiv preprint arXiv:1707.06347 (2017).
>
>
>
>
> **Q6**: Algorithmic Practicality
>
> **Answer**: Thank you for mentioning this. We **added Section L.2** in the Appendix with runtime and memory analysis. Tables 1 (CartPole, Acrobot) and 2 (HalfCheetah-v5) show that NPG remains practical as problem size grows. On Apple M4, standard tasks like CartPole-v1 and Acrobot-v1 run 1,000 episodes in under a minute. For the high-dimensional HalfCheetah-v5 (state dim 17, action dim 6, 1,000 steps per episode), the runtime for 10,000 episodes remains manageable at 800–900 seconds. Memory usage is stable and low across environments, confirming feasibility. Overall, NPG scales robustly to high-dimensional settings, and since it does not require ordered trajectory sampling, GPU parallelization could further reduce runtime.
>
> **Table 1: Performance comparison on Cartpole-v1 and Acrobot-v1.**
>
> | Environment | Cartpole-v1 |  |  | Acrobot-v1 |  |  |
> | :--- | :---: | :---: | :---: | :---: | :---: | :---: |
> | Step size | $k^{-0.2}$ |$k^{-0.5}$ | $k^{-1.5}$ | $k^{-0.2}$ | $k^{-0.5}$ | $k^{-1.5}$ |
> | Running time (s) | $30.26 \pm 4.49$ | $49.04 \pm 7.59$ | $7.85 \pm 2.42$ | $80.82 \pm 20.12$ | $49.77 \pm 25.28$ | $96.07 \pm 0.54$ |
> | Peak Memory cost (MB) | $282.06 \pm 6.79$ | $272.77 \pm 0.10$ | $273.00 \pm 0.04$ | $194.8 \pm 60.62$ | $230.04 \pm 0.17$ | $230.52 \pm 0.16$ |
>
>
> **Table 2: Performance comparison on HalfCheetah-v5.**
>
> | Step size | $k^{-0.1}$ | $k^{-0.5}$ | $k^{-1.5}$ |
> | :--- | :---: | :---: | :---: |
> | Running time (s) | $834.23 \pm 18.56$ | $911.84 \pm 48.86$ | $917.94 \pm 72.95$ |
> | Peak Memory cost (MB) | $202.65 \pm 70.77$ | $116.47 \pm 8.26$ | $160.79 \pm 58.42$ |

---

### Author Response · Authors · 2025-12-03
**Summary of our revision and response**

Dear Area Chair,

We sincerely thank all the reviewers for recognizing the **novelty**, **theoretical depth**, **methodological rigor**, and **empirical validation** of our work. We appreciate that **Reviewer 4sex** commends the paper for providing *"the first comprehensive analysis of TD and PPO/NPG in infinite-dimensional RKHS"*; **Reviewer eHMp** acknowledges the *"significant contribution by combining kernel methods with reinforcement learning"*; **Reviewer L5ur** highlights that *"the analysis of kernelized TD critic is clean and rigorous"* and *"connect TD learning with NPG-PPO-style policy improvement updates is classic yet elegant"*; and **Reviewers eHMp and 4sex** agree that *"the experimental results strongly support the theoretical analysis and show improved stability and efficiency compared with the standard PPO"*.

In response to the reviewers’ comments, we have incorporated the following major additions and improvements into the manuscript:

**1.Relaxation of Assumptions 1 and 6 (all reviewers)**:
We have weaken the boundedness requirements on the initial distributions, thereby broadening the applicability of our theoretical results.

**2.Generalization of the RKHS framework (reviewer L5Ur)**:
Our original one-step temporal-difference formulation has been extended to the *N-step TD* and *TD($\rho$)* settings.

**3.Inclusion of a PPO variant (reviewers L5Ur and 4sex)**:
We have added a variant of PPO update rule to further demonstrate the universality and flexibility of our RKHS-based framework.

**4. New experiments (reviewers L5Ur, 4sex, and eHMp)**:
We have added: (i) HalfCheetah-v5 (MuJoCo) to confirm scalability in high-dimensional environments; (ii) runtime and memory analysis to verify practical feasibility; (iii) additional baselines (SAC, DQN, clip-PPO, implicit PPO) to benchmark efficiency; and (iv) N-step TD learning with varying N to validate our theoretical extensions.

Based on reviews, we have revised the manuscript to update **Assumption 1** (lines 127-131 in Section 3), **Assumption 6 and Remark 7** (lines 190-197 in Section 4), **N-Step TD learning** (Appendix J), and **Implicit PPO** (Appendix K). We have also added **new experiments** in Section 6.4 and Appendix L.3.

We believe that our comprehensive response addresses all reviewer concerns and further strengthens the paper's contributions. We once again sincerely thank the reviewers and Area Chairs for their time and consideration.

---

### Meta-Review · Area_Chair_tZRY · 2026-01-01

**Summary:**

This work introduces a kernelised framework for PPO in the RKHS, combining kernelised TD with natural gradient updates.

Reviewers appreciated the soundness and analysis, as well as the unification of tabular, linear, and NTK regimes.
The main critiques focused on the strong assumptions (specifically Assumptions 1 and 6), limited experimental scope (low-dimensional tasks and missing baselines), and scalability. The rebuttal effectively addressed these by relaxing the boundedness assumptions, incorporating high-dimensional benchmarks (HalfCheetah) and other baselines (SAC, clip-PPO), and providing runtime analysis. The also extended the framework to N-step TD.

In conclusion, the main concerns have been addressed and I **recommend this paper to be accepted as poster**.

**Reviewer Concerns:**

- **Strong assumptions:** Several reviewers questioned the strong assumptions, particularly regarding bounded distributions (Assumption 1) and stationarity (Assumption 6) (Reviewers 4sex, L5Ur, eHMp, 4UFk). *Rebuttal:* The authors relaxed Assumption 1 to require only bounded sampling distributions and clarified that Assumption 6 is strictly weaker than the stationary sampling assumptions in prior work.

- **Experiments and scalability:** Reviewers 4sex and eHMp criticised the limited experimental scope and lack of computational analysis. *Rebuttal:* The authors added high-dimensional tasks (HalfCheetah), baselines (SAC, clip-PPO), and runtime analysis to demonstrate scalability.

- **Multi-step methods:** Reviewer L5Ur requested extensions to multi-step methods. *Rebuttal:* The authors extended the framework to N-step TD and TD(λ).

In conclusion, the main critiques have been addressed.

**Reviewer Scores:**

- **Reviewer L5Ur Rating: 6 / Confidence: 3** The authors provided an extension to N-step TD and TD(λ) and relaxed the stationarity assumption. I would expect a confirmation or an increase in the rating.

- **Reviewer 4sex Rating: 6 / Confidence: 2** Their main concerns were limited experimental scope and missing baselines, these were addressed by adding a high-dimensional benchmark (HalfCheetah) and comparing against SAC and standard PPO. I would expect a confirmation or an increase in the rating.

- **Reviewer 4UFk Rating: 4 / Confidence: 2** This reviewer stated that they were "not the intended audience" and decided to wait for the Reviewers' discussion phase to update their score. I would have expected a score increase during the discussion (probably to 6), to reflect that the other reported issues have been resolved.

- **Reviewer eHMp Rating: 8 / Confidence: 3** This reviewer explicitly replied to the rebuttal, stating they appreciated the response and would "maintain [their] score".

---

### Decision · Program_Chairs · 2026-01-26

Accept (Poster)